# High-throughput cell and spheroid mechanics in virtual fluidic channels

Muzaffar H. Panhwar [1,2], Fabian Czerwinski [1], Venkata A. S. Dabbiru [1,2], Yesaswini Komaragiri[1,2], Bob Fregin [1,2], Doreen Biedenweg[3], Peter Nestler[1], Ricardo H. Pires[1,2] & Oliver Otto [1,2 ✉]

Microfluidics by soft lithography has proven to be of key importance for biophysics and life science research. While being based on replicating structures of a master mold using benchtop devices, design modifications are time consuming and require sophisticated cleanroom equipment. Here, we introduce virtual fluidic channels as a flexible and robust alternative to microfluidic devices made by soft lithography. Virtual channels are liquid-bound fluidic systems that can be created in glass cuvettes and tailored in three dimensions within seconds for rheological studies on a wide size range of biological samples. We demonstrate that the liquid-liquid interface imposes a hydrodynamic stress on confined samples, and the resulting strain can be used to calculate rheological parameters from simple linear models. In proof-of-principle experiments, we perform high-throughput rheology inside a flow cytometer cuvette and show the Young's modulus of isolated cells exceeds the one of the corresponding tissue by one order of magnitude.

[1] Zentrum für Innovationskompetenz: Humorale Immunreaktionen bei kardiovaskulären Erkrankungen, Universität Greifswald, Fleischmannstr. 42, 17489 Greifswald, Germany. [2] Deutsches Zentrum für Herz-Kreislauf-Forschung e.V., Standort Greifswald, Universitätsmedizin Greifswald, Fleischmannstr. 42, 17489 Greifswald, Germany. [3] Klinik für Innere Medizin B, Universitätsmedizin Greifswald, Fleischmannstr. 8, 17475 Greifswald, Germany. ✉email: oliver.otto@uni-greifswald.de

Phenotyping of cells utilizing intrinsic material properties, i.e., elasticity and viscosity, is increasingly being recognized to be of key relevance in basic and applied life science research[1–3]. Early developments have mainly been driven by technologies like atomic force microscopy[4], micropipette aspiration[5] as well as optical stretching[6,7] and focused on the fundamental interplay between cytoskeletal protein structure and viscoelastic cell properties to combine physics and biology on a molecular as well as cellular scale[8–10]. In this field, mechanosensitivity has been found as an essential mechanism of how cells respond to and communicate with their environment[11].

The great success of mechanical phenotyping to study cell function paved the way toward applications in translational research but required an elevated sample throughput that could not be reached by established methods. One answer to this challenge was found in microfluidic devices[12–14]. Already decades ago, Hochmuth et al. utilized glass capillaries and high-speed video microscopy to study red blood cell deformation in flow[15–17]. In their pioneering work they carried out an extensive characterization of erythrocyte shape, orientation and viscosity. Glass capillaries share the advantages of all microfluidic systems, i.e., high-throughput and serial sample processing capabilities, but lack the flexibility to design and implement sophisticated fluid flows.

The introduction of soft lithography as a rapid prototyping technology revolutionized the way of creating microfluidic structures[18,19]. Using replicas of elastomers enabled benchtop production of lab-on-a-chip devices to access the nano- and microscale for single molecule and single-cell experiments[20–23]. For example, Gossett et al. demonstrated high-speed mechanical phenotyping of various cell lines and showed for embryonic stem cells a discrimination of their pluripotent state with sufficient statistical power[24]. The idea of stretching cells in an extensional flow has been expanded by Dudani et al. toward a pinched-flow geometry, where hydraulic circuit design is used to deform cells in multiple stretching modes[25]. Recently, mechanical cell characterization has also been complemented with real-time data analysis[26,27] as well as fluorescent labeling to combine the sensitivity of intrinsic material properties with the specificity of molecular markers[28].

While lab-on-a-chip devices are nowadays routinely being used in translational research to assess viscoelastic properties, e.g., for studying the onset of infectious diseases[29] and for monitoring patients' status over time[30,31], the high-throughput analysis of this label-free biomarker is currently limited to single cells and excludes multicellular structures. However, a detailed understanding of how mechanics impacts on tissue development and homeostasis would be highly relevant to establish 3D cell culture models that incorporate material properties into the field of tissue engineering[32,33]. Despite some initial work on spheroids using atomic force microscopy and micropipette aspiration current approaches lack the potential of simultaneous high-throughput characterization of cells and tissues applying a single experimental method[34,35].

A suitable microfluidic system for high-speed mechanical phenotyping of spheroids is a glass cuvette. Having a diameter of several hundred of micrometers these cuvettes are the main entity in fluorescence-based flow cytometers, where translocating cells are analyzed by laser excitation and detection of the emitted as well as the scattered light[36]. While providing sufficient confinement for spheroids, commercial flow cytometers are not capable to analyze single cells by their mechanical properties. One reason is found in the large cross-sectional area of the cuvette, which is needed to reduce blocking by debris but also results in low hydrodynamic stress amplitudes. Since determination of elasticity and viscosity requires, first, exposure of cells to sufficient shear as well as normal stresses and, second, detection of the corresponding cellular deformation, the large cuvette diameter effectively inhibits mechanical measurements.

Here, we introduce virtual fluidic channels to create microfluidic constrictions of variable cross-sections inside glass cuvettes of commercial flow cytometers. Virtual channels are liquid-bound microfluidic devices and are formed by co-flowing aqueous polymer solutions at low Reynolds numbers where a stabilizing sheath confines a sample flow between immiscible liquid phases. We demonstrate that virtual fluidic channel dimensions can be adjusted within seconds in real-time and enable high-throughput rheological measurements of biological samples of both, cells and spheroids.

Taking advantage of a simple scaling law defining a channel diameter, which depends on flow rates and viscosity but not on polymer composition, we investigate suspended cells in different degrees of virtual confinement. In contrast to bullet-like cells found in Poiseuille flows our results reveal a characteristic shape resembling an ellipsoid. This steady-state can be explained by a hydrodynamic stress originating from the viscosity mismatch at the liquid–liquid interface and we show its utilization for simple creep-compliance experiments that allow for an extraction of material properties.

We develop an analytical model and compare the interfacial stress to typical shear and normal stress distributions inside Poiseuille flows of laminar systems. Our results verify that cell deformation of small objects embedded in virtual channels inside large geometries, e.g., single cells in cuvettes, is governed by interfacial stress, while large objects, e.g., spheroids as a tissue model system in glass cuvettes, deform dominantly by shear stress.

We combine our technology with real-time deformability cytometry (RT-DC) for high-throughput characterization of biological samples. In proof-of-principle experiments we show that virtual channels can be applied to perform mechanical cell assays and to identify cytoskeletal alterations label-free. Finally, we investigate the role of single-cell mechanics for tissue stability. Virtual channels allow for direct comparison between the rheological properties of single cells and the corresponding tissues. Our results indicate that the elastic modulus of cells exceeds the one of tissue by an order of magnitude, while tissue stiffness increases with size.

## Results

**Virtual fluidic channels in microfluidic systems**. Virtual channel formation is first demonstrated inside a polydimethylsiloxane (PDMS) microfluidic chip with a squared constriction of 30 μm × 30 μm cross-section and 300 μm length (Fig. 1a, upper half and Supplementary Fig. 1A). Confining a sample solution of 57 μM methylcellulose (MC) inside a sheath flow of 50 mM polyethylene glycol (PEG) 8000, both in PBS, leads to an interface, which is stable and immiscible on the experimental time scale (Supplementary Fig. 2 and Supplementary Movie 1). Inside the constriction both aqueous phases can be distinguished optically by a refractive index difference at the liquid–liquid interface that we use to define the width $w$ of the virtual channel (Fig. 1a, top inset, white dashed line and Supplementary Movie 2). Finite element method (FEM) simulations of the full microfluidic geometry assuming a two-phase Stokes flow reveal the same binary concentration distribution of MC and PEG (Fig. 1a, lower half).

While the greater dynamic viscosity of the PEG solution effectively impedes flow and acts as a liquid wall inside the PDMS geometry, the MC phase with its lower viscosity tunnels through this confinement (Fig. 1a, top inset). The velocity profile is described by the superposition of two parabolic functions, for the

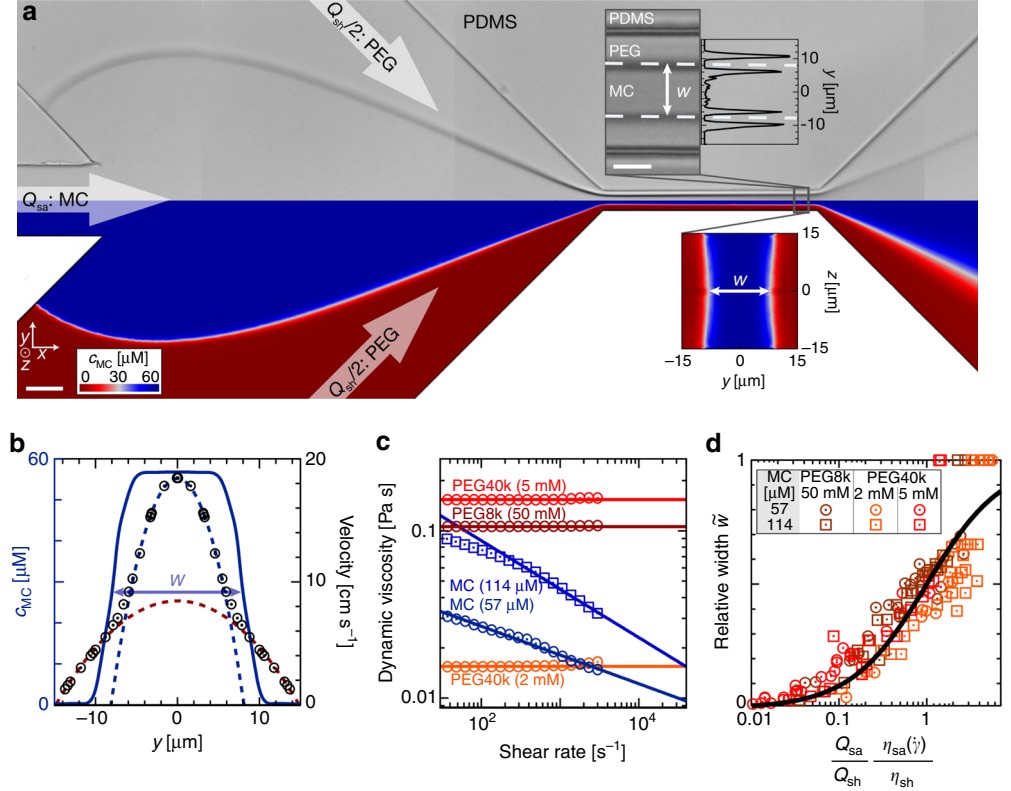

**Fig. 1 Virtual fluidic channel inside microfluidic chip. a** Microfluidic chip as stitched microscopy image (upper half) and as the concentration plot of a finite element method (FEM) simulation of the full geometry (lower half). Arrows indicate inflow of 57 µM methylcellulose in PBS (MC, flow rate $Q_{sa}$ = 48 nl s$^{-1}$) and 50 mM polyethylene glycol 8000 in PBS (PEG8000, flow rate $Q_{sh}$ = 8 nl s$^{-1}$) forming a stable virtual fluidic channel between two liquid–liquid interfaces represented in dark gray (upper half) and in white (lower half). Scale bar is 50 µm. Top inset shows a bright-field image of the central constriction and the projected squared intensity gradient (arb. units) across the full channel width. Virtual channel width corresponds to distance $w$ (white dashed lines) between the center of both intensity maxima. Scale bar is 10 µm. Bottom inset shows a cross-sectional view of the calculated (FEM) polymer concentration inside the channel. **b** Velocity profile (black circles) inside the center of the constriction derived from FEM simulations with the corresponding MC concentration distribution (blue solid line) used to identify the virtual channel width $w$. The red and the blue dashed lines indicate the parabolic flow profiles of the outer and inner aqueous phases. **c** Dynamic viscosities of sample and sheath solutions as a function of shear rate. Virtual channel formation is performed using MC as sample buffer (blue) and PEG as sheath buffer (dark red, orange, bright red). Data points are measured and for shear rates greater than 3000 s$^{-1}$ the shear-rate dependency is modeled as a power-law fluid (solid blue line) and as Newtonian fluid (solid orange and red lines). **d** Relative virtual channel width $\tilde{w}$ as a function of flow rate and viscosity ratios. The plot summarizes $n$ = 146 experiments using different concentrations of MC, PEG8000 (PEG8K), and PEG40000 (PEG40K) for sample and sheath solution. The black curve is a solution to Eq. (1).

inner and outer aqueous phase (Fig. 1b, blue and red dashed line), respectively[37]. The velocity gradient of the resulting piecewise-defined Poiseuille flow (Fig. 1b, black circles) is given by the viscosity mismatch of both polymer solutions (Fig. 1c) where the MC phase corresponds to an effective (virtual) channel size smaller than the cross-section of the PDMS chip.

We investigated the impact of flow rate and viscosity on virtual channel width as these parameters allow for a precise adjustment of the liquid–liquid interface. Analyzing different polymers of various chain lengths and concentrations we confirm a simple functional relationship reported earlier[38,39]:

$$\frac{Q_{sa}}{Q_{sh}} \frac{\eta_{sa}(\dot{\gamma})}{\eta_{sh}} = \frac{\tilde{w}}{1 - \tilde{w}}, \tag{1}$$

where $Q_{sa}$ and $Q_{sh}$ are the flow rates of sample as well as sheath, $\eta_{sa}$ and $\eta_{sh}$ are the corresponding viscosities and $\tilde{w}$ is the channel width relative to the diameter of the PDMS constriction (see Methods). The viscosity of sample solution is derived from a power law utilizing experimental shear rates $\dot{\gamma}$ while our sheath solution follows a Newtonian behavior (Fig. 1c, see Methods).

The fact, that the relative virtual channel width is only determined by the flow rate and viscosity ratios at the respective shear rates, qualifies well-defined flow conditions unconstrained

by polymer length, concentration and the microfluidic chip (Fig. 1d). Considering the non-linear rheological properties of MC revealing a pronounced shear-thinning component, this simple relationship is unexpected in a complex hydrodynamic environment of co-flowing aqueous phases.

**Cell mechanical phenotyping in virtual channels**. Next, we study the capability of virtual channels as a confining constriction for probing mechanical properties of suspended cells. Using the myeloid precursor cell line HL60, RT-DC is performed in a standard PDMS chip of 20 µm × 20 µm cross-section[26] and results are compared with measurements inside a virtual channel of 21 µm width formed in a larger 30 µm x 30 µm chip (see Methods). Mechanical phenotyping in both, plastic chip and virtual channel, reveals similar distributions in cell size and deformation (Fig. 2a, b), cells display the typical bullet shape (Fig. 2a, b, insets) and only slightly perturb the MC-PEG interface (Fig. 2c).

Sensitivity of our system toward cytoskeletal modifications has been assessed by exposing HL60 cells to 1 µM cytochalasin D (CytoD, see Methods). For cells inside a PDMS channel the depolymerization of filamentous actin reveals the expected

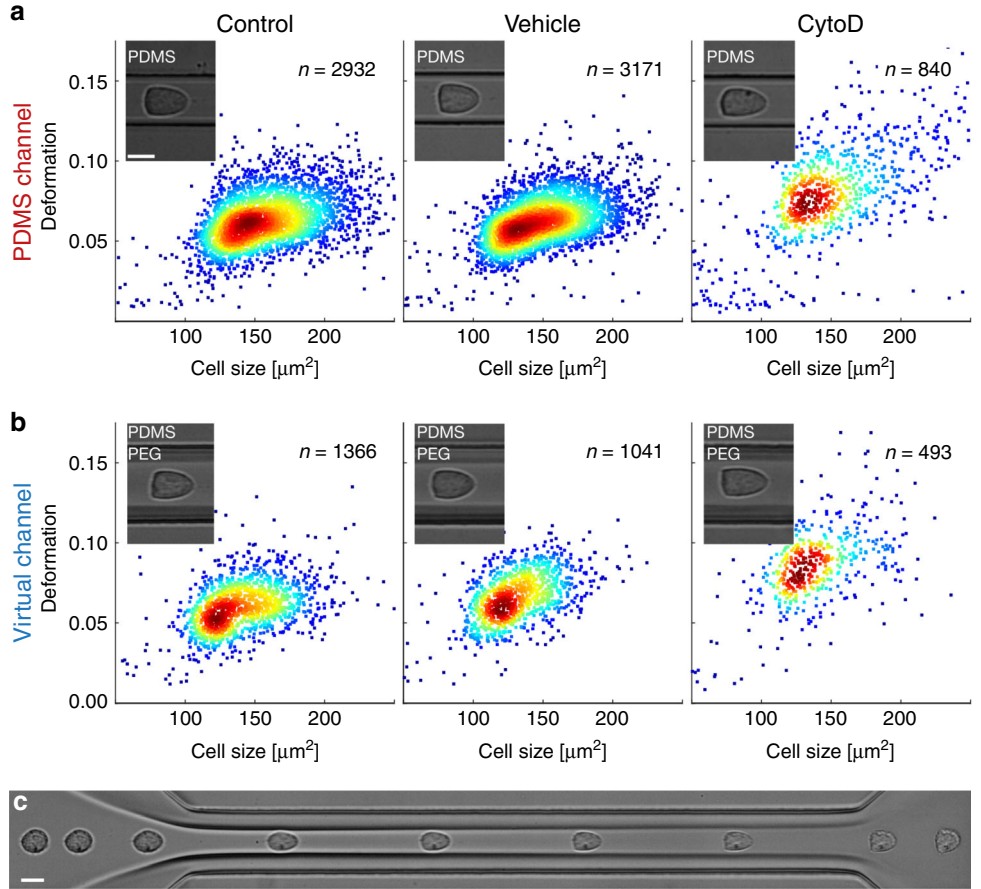

**Fig. 2 Cell deformation in PDMS chip and virtual fluidic channel. a** Real-time deformability cytometry (RT-DC) of HL60 cells in polydimethylsiloxane (PDMS) channel yielding scatter plots of deformation versus cell size for control cells (left), dimethyl sulfoxide (DMSO) vehicle control (0.25% (v/v), center) and 1 μM CytoD (right). Measurements have been done at a total flow rate of 40 nl s$^{-1}$ in a PDMS chip with a 300 μm long channel and 20 μm × 20 μm squared cross-section using 57 μM MC for sample and sheath buffer, respectively. **b** RT-DC of HL60 cells in a virtual channel of 21 μm width and 30 μm height for control cells (left), DMSO vehicle control (0.25% (v/v), center) and 1 μM CytoD (right). Virtual channel is formed inside a PDMS chip with a 300 μm long channel and 30 μm × 30 μm squared cross-section using 57 μM MC (sample) as well as 50 mM PEG8000 (sheath). Measurements are taken at indicated position (Fig. 1a, gray rectangle) and a total flow rate of 94 nl s$^{-1}$ ($Q_{sa}$ = 90 nl s$^{-1}$, $Q_{sh}$ = 4 nl s$^{-1}$). Insets show representative cell images. **c** Time series of single HL60 cell passing a virtual channel (sample 57 μM MC, $Q_{sa}$ = 72 nl s$^{-1}$, sheath 50 mM PEG8000, $Q_{sh}$ = 8 nl s$^{-1}$). Average cell velocity within the channel is ~20 cm s$^{-1}$. Scale bar is 10 μm. Recording of time series has been repeated five times. Color in scatter plots indicates a linear density scale from min (blue) to max (red).

increase in deformation $d$ from $d = 0.062 \pm 0.016$ (median ± std. dev.) for the control sample and $d = 0.062 \pm 0.015$ for the vehicle control of 0.25% (v/v) dimethyl sulfoxide (DMSO) to $d = 0.080 \pm 0.030$ for the 1 μM CytoD treatment (Fig. 2a). Virtual channel measurements agree with these results and yield $d = 0.059 \pm 0.015$ (control), $d = 0.065 \pm 0.015$ (vehicle control of 0.25% (v/v) DMSO) as well as $d = 0.087 \pm 0.023$ (1 μM CytoD) where flow rates have been adjusted to match the stress distribution on the cell surface inside the PDMS chips (Fig. 2b and Supplementary Fig. 3).

A statistical analysis of three experimental replicates summarizes more than 20,000 single-cell measurements and confirms in both systems the expected significant increase in cell deformation and decrease in Young's modulus $E$ relative to the vehicle control and control when cells are being exposed to 1 μM CytoD (Fig. 3, Supplementary Figs. 4 and 5). Importantly, we find no significant differences in deformation and Young's modulus $E$ comparing results in PDMS and virtual channels. In contrast, a significant decrease in cell size is found when cells are confined by a MC-PEG interface. This observation can be attributed to the geometry of our microfluidic chip having a sheath current from two sides, both connected to the same inlet. The corresponding

virtual channel is always aligned in the center of the constriction (Fig. 1a) confining the cells perpendicular to one direction of flow only and leaving the channel height of 30 μm unaltered (Fig. 1a, bottom inset).

Utilizing FEM simulations, we investigate the impact of a non-squared channel cross-section on the hydrodynamic stress distribution. While a 20 μm × 20 μm PDMS channel leads to a mean shear stress on the cell surface of 970 Pa (Supplementary Fig. 3, left), we find 908 Pa for a virtual channel having 21 μm width and a 30 μm height (Supplementary Fig. 3, right). With a difference below 10% our calculations suggest comparable microfluidic conditions, a conclusion which is supported by experimental results (Fig. 3).

**Cell mechanics in mesofluidic systems.** In a following step, we transfer our experimental framework to mesoscopic geometries on the centimeter scale. This approach would facilitate microfluidic assays controlled by flow rates and viscosities, fully unconstrained by (soft) lithography and other fabrication techniques. As a proof-of-principle experiment we use a standard flow cytometer glass cuvette of 2 cm length (Fig. 4a and Supplementary Fig. 1b). Although exceeding dimensions of standard

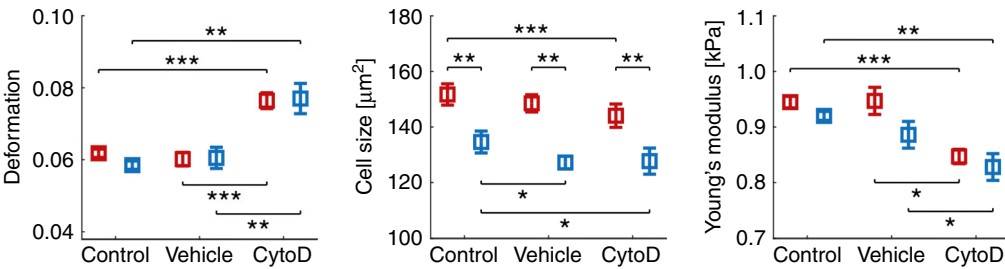

**Fig. 3 Analysis of cytoskeletal alterations using virtual fluidic channels.** Statistical analysis using linear mixed models of three independent biological replicates for cell deformation (left), cell size (center) and Young's modulus (right) inside a 20 μm × 20 μm PDMS chip (red) and 21 μm virtual fluidic channel inside a 30 μm × 30 μm PDMS chip (blue). Data compares control cells (n = 8024; PDMS and n = 4040; virtual channel), vehicle control (n = 8948; PDMS and n = 2778; virtual channel) as well as cells after treatment with 1 μM CytoD (n = 4148; PDMS and n = 1237; virtual channel). Data are presented as mean ± standard error of the mean (*p < 0.05; **p < 0.01; and ***p < 0.001).

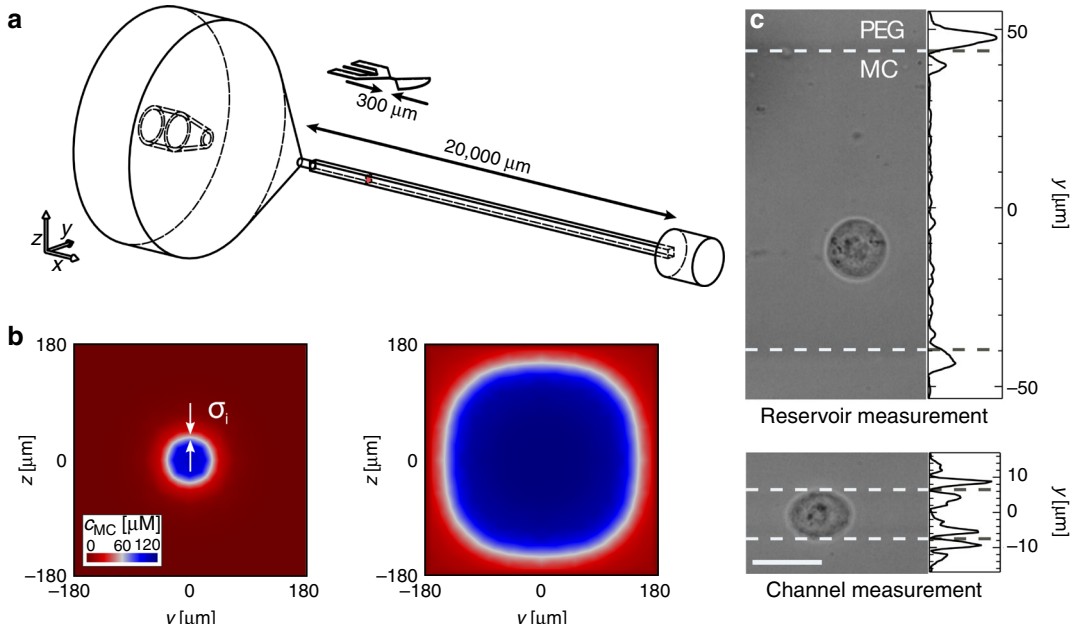

**Fig. 4 Virtual channel formation inside a flow cytometer glass cuvette. a** Technical drawing of glass cuvette and PDMS chip to scale for comparison. Arrows indicate length of constrictions of 2 cm for the cuvette and 300 μm for the PDMS device. **b** FEM simulations of concentration distribution for the cross-section of the cuvette performed for sample 114 μM MC at $Q_{sa} = 200$ nl s$^{-1}$ and sheath 5 mM PEG40000 at $Q_{sh} = 1000$ nl s$^{-1}$ yielding a 80 μm virtual channel (left). Arrows indicate the stress $\sigma_i$ due to the viscosity mismatch at the interface. Adjusting the flow rates to $Q_{sh} = 50$ nl s$^{-1}$ while keeping $Q_{sa}$ constant, enables increasing the diameter of the constriction to 260 μm (right). **c** Representative image of a HL60 cell in a virtual channel of 88 μm diameter (sample 114 μM MC at $Q_{sa} = 200$ nl s$^{-1}$, sheath 5 mM PEG40000 at $Q_{sh} = 1000$ nl s$^{-1}$, top) and inside a virtual channel of 14 μm diameter (sample 114 μM MC at $Q_{sa} = 15$ nl s$^{-1}$, sheath 5 mM PEG40000 at $Q_{sh} = 1000$ nl s$^{-1}$, bottom). Projected line plots at top and bottom indicate the squared intensity gradient (arb. units) perpendicular to the flow direction while center between the maxima identify virtual channel interfaces (white dashed lines). Recording of cells and spheroids inside virtual channels has been repeated six times where a fluctuation in virtual channel size of 10% has been observed.

microfluidic chips by almost two orders of magnitude, a stable virtual channel of circular cross-section can be formed inside a squared constriction of 360 μm side length (Supplementary Fig 6). The existence of two steady co-flowing aqueous solutions can be predicted from FEM simulations on the full flow profile (Fig. 4b) and has been confirmed by experimental data (Fig. 4c).

Adapting Eq. (1) to the cuvette geometry demonstrates that the virtual constriction can be controlled by sample and sheath flow rates as well as viscosities (see Methods):

$$\frac{Q_{sa}}{Q_{sh}}\frac{\eta_{sa}(\dot{\gamma})}{\eta_{sh}} = \frac{\frac{\pi}{4}\tilde{w}^2}{1-\frac{\pi}{4}\tilde{w}^2}, \qquad (2)$$

where the channel diameter w can be adjusted between a few and more than one hundred micrometers (Fig. 4b, c). Here, $\tilde{w}$ is determined relative to the side length of the cuvette cross-section.

Having a dynamic size range exceeding one order of magnitude our microfluidic system can be applied to mechanically characterize cells in their reference state, i.e., in large virtual channels at low hydrodynamic stress (Fig. 4c, top) and under a finite hydrodynamic stress in narrow virtual channels (Fig. 4c, bottom)[27]. Interestingly, the sheath inside the constriction of the glass cuvette fully encloses the sample and enables a true three-dimensional cell confinement (Fig. 4b and Supplementary Fig. 7) and a homogeneous stress distribution on the cell surface mimicking the axial symmetric velocity profile (Supplementary Fig. 7, left panel). This is in contrast to standard PDMS chips, where virtual channels are limited to 2.5 dimensions with fixed feature heights given by the planar geometry of the master mold (Fig. 1a, bottom inset and Supplementary Fig. 7, right panel; see Methods). The different sheath flow geometries in the cuvette and

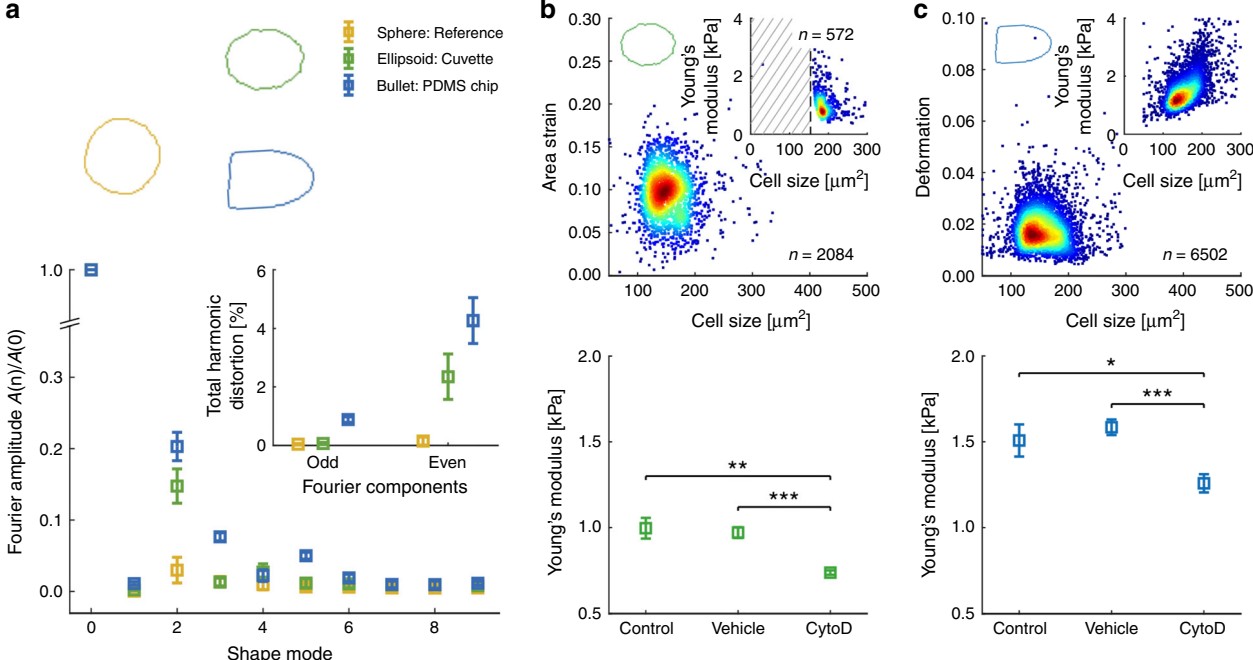

**Fig. 5 High-throughput cell mechanics inside a flow cytometer glass cuvette. a** Fourier amplitudes of cellular shape modes for $n = 25$ HL60 cells inside a $w = 88\ \mu m$ (yellow) and $w = 14\ \mu m$ (green) virtual channel of a glass cuvette as well as $n = 25$ HL60 cells inside the virtual channel of PDMS chip shown in Fig. 2 (blue). The inset represents the total harmonic distortion calculated for the cells inside the virtual channel of a cuvette (green), a PDMS chip (blue) and a reference measurement (yellow). Data are presented as mean ± standard deviation. **b** Area strain of HL60 cells inside a $w = 14\ \mu m$ virtual channel of a glass cuvette ($Q_{sa} = 15\ nl\ s^{-1}$ and $Q_{sh} = 1000\ nl\ s^{-1}$, top). Inset shows Young's modulus distribution of HL60 cells calculated from area strain with $\sigma_i = 78\ Pa$, shaded area indicates fraction of cells not confined by the virtual channel. Bottom graph compares the Young's modulus of three independent biological replicates of HL60 cells ($n = 2689$), a DMSO vehicle control (0.25% (v/v); $n = 3829$), and HL60 cells treated with $1\ \mu M$ CytoD ($n = 3497$). **c** Deformation of HL60 cells inside a $w = 20\ \mu m$ virtual channel of a glass cuvette ($Q_{sa} = 20\ nl\ s^{-1}$ and $Q_{sh} = 1000\ nl\ s^{-1}$, top). Inset shows Young's modulus distribution of HL60 cells calculated from a hydrodynamic model considering shear- and normal stress on cell surface in steady-state. Bottom graph compares Young's modulus of three independent biological replicates of HL60 cells ($n = 19{,}018$), a DMSO vehicle control (0.25% (v/v); $n = 23{,}612$) and HL60 cells treated with $1\ \mu M$ CytoD ($n = 15{,}245$). Scale bar is $20\ \mu m$. Color in scatter plots indicates a linear density scale from min (blue) to max (red). Statistical data analysis is performed using linear mixed models and data are presented as mean ± standard error of the mean ($*p < 0.05$; $**p < 0.01$; $***p < 0.001$).

PDMS system also explain the different scaling in relative virtual channel width $\tilde{w}$ of Eqs. (1) and (2).

Next, we ask if virtual channels inside a flow cytometer cuvette are suited to induce cell deformation. The answer is not obvious since cell surface stress scales with the velocity gradient between the maximum at channel center and zero velocity due to the no-slip boundary condition at the glass surface (Fig. 1b). Assuming typical experimental values for a cuvette of side length $360\ \mu m$ and $3\ cm\ s^{-1}$ cell velocity this results in shear rates in the order of $10^2\ s^{-1}$, which is in contrast to $\sim 10^4\ s^{-1}$ for cells inside a $20\ \mu m$ PDMS chip at a velocity of $10\ cm\ s^{-1}$.

Despite two orders in magnitude difference in shear rate we do not only find the majority of cells inside the virtual channel of a cuvette deforming but also observe an ellipsoidal steady-state shape (Fig. 4c, bottom and Fig. 5a, green contour), which clearly deviates from the bullet shape in PDMS systems (Fig. 2c and Fig. 5a, blue contour) and from the circular reference state (Fig. 4c, top and Fig. 5a, yellow contour). Aiming to understand the physical origin of the alternating cell shapes, we perform a Fourier decomposition of the cell contour analyzing the amplitude of the first ten Fourier components[40].

For cells inside a virtual channel of a glass cuvette the second shape mode dominates (Fig. 5a, green) while cells inside the virtual channel of a PDMS chip possess significant contributions of the third and fifth mode (Fig. 5a, blue). Looking at the total harmonic distortion of the first ten Fourier components we find

no contribution of the odd modes to ellipsoidal cell shape (Fig. 5a, inset), which implies a deformation inside the cuvette perpendicular to the direction of flow, only.

The fundamental difference in Fourier spectra points toward stress distributions on the cell surface that originate from different physical principles. We speculate that, while cell deformation in narrow channels, i.e., (virtual channels in) micrometer-sized systems, is mainly driven by shear, cells inside virtual channels of a cuvette are deformed by an interfacial stress $\sigma_i$ originating from the liquid–liquid interface (Fig. 4b, left and Fig. 4c, bottom).

We investigate this hypothesis by calculating the hydrodynamic shear stress and normal stress on a cell surface inside a glass cuvette using FEM simulations (see Methods). Here, we assume a total flow rate of $1.02\ \mu l\ s^{-1}$ ($Q_{sh} = 1000\ nl\ s^{-1}$, $Q_{sa} = 20\ nl\ s^{-1}$) and a constant viscosity of $11\ mPa\ s$. In geometries with a relative size $\lambda = r_{sphere}/R$, where $r_{sphere}$ is the cell radius and $R$ is the equivalent cuvette radius, peak stress increases with increasing $\lambda$ for a fixed flow rate (Supplementary Fig. 8). While this relationship is valid for any laminar microfluidic system with a central spherical obstacle, we can show that virtual channels also possess a constant stress contribution at the interface resulting from the viscosity mismatch between the two co-flowing aqueous polymer phases. This effect has been described previously for systems where an annular flow of low viscosity, surrounds a core of high viscosity, e.g., oil, to reduce friction at

the oil-wall interface for power-effective transportation in pipes[41]. Under these conditions, $\sigma_i$ can be derived from the simple expression:

$$\sigma_i = -\frac{8\,\eta_{sa}(\dot\gamma)}{w}(u_c - 2u_a), \qquad (3)$$

where $\eta_{sa}(\dot\gamma)$ is the viscosity of the sample flow at shear rate $\dot\gamma$, $w$ is the width of the virtual channel, $u_c$ is the cell velocity inside the virtual channel and $u_a$ the superficial velocity proportional to $\frac{Q_{sh}}{\pi R^2}$.

Utilizing experimental values for HL60 cells moving at a flow rate of $1.015\,\mu l\,s^{-1}$ ($Q_{sh} = 1000\,nl\,s^{-1}$, $Q_{sa} = 15\,nl\,s^{-1}$) inside a $w = 14\,\mu m$ virtual channel of a cuvette with a side length of $360\,\mu m$ we obtain a velocity of $u_c = 2.8\,cm\,s^{-1}$ and a stress at the liquid–liquid interface of $\sigma_i = 78\,Pa$. In comparison to the hydrodynamic shear and normal stress, $\sigma_i$ dominates (Supplementary Fig. 8) thus inducing cell deformation predominantly perpendicular to the direction of flow (Fig. 4c, bottom). For the calculation, viscosity values of $\eta_{sa}(\dot\gamma) = 11\,mPa\,s$ were taken from FEM simulations and independently confirmed using calibration beads with a fixed elastic modulus of $E = 1.5 \pm 0.5\,kPa$ (see Methods)[42].

Since Eq. (3) allows to consider the ellipsoidal cell shape as the steady-state response to a simple creep-compliance experiment with a constant interfacial stress $\sigma_i = E\,\varepsilon_c$, the Young's modulus can directly be calculated from the area strain $\varepsilon_c$ of the cell (see Methods). Effectively, the liquid–liquid interface acts as a high-frequency liquid cantilever for probing global cell rheological properties on a millisecond time scale such that material properties can be extracted from simple linear models.

As a proof-of-principle experiment we characterize HL60 cells inside the virtual channel of a glass cuvette utilizing the parameters above. Assuming volume conservation the area strain can be calculated (see Methods) where a mean value of $\bar\varepsilon_c = 0.10 \pm 0.03$ (mean ± std. dev.) is obtained analyzing more than two thousand cells (Fig. 5b, top). From the single strain values $\varepsilon_c$ the Young's modulus of each cell can be derived (Fig. 5b, top inset). Here, the shaded area defines an exclusion zone of cells being smaller than the virtual channel diameter, which are not probed by the interfacial stress and excluded for rheological analysis.

In a final step we perform measurements on HL60 cells after cytoskeletal modifications where we apply the liquid–liquid interface to induce cell deformation. Analyzing three biological replicates, we compare the Young's modulus of control cells ($n = 2689$), the corresponding DMSO control (0.25% (v/v); $n = 3829$), and cells after treatment with $1\,\mu M$ CytoD ($n = 3497$). A statistical analysis reveals no difference between control ($E = 1.00 \pm 0.06\,kPa$ (mean ± SEM)) and vehicle ($E = 0.97 \pm 0.03\,kPa$), but a significant reduction after treatment with CytoD ($E = 0.74 \pm 0.03\,kPa$) compared with the control ($p < 0.0065$) and the DMSO control ($p < 0.0006$; Fig. 5b, lower graph and Supplementary Fig. 9).

For verification of our results, we adjust the virtual channel diameter inside the glass cuvette to $w = 20\,\mu m$ utilizing a total flow rate $1.02\,\mu l\,s^{-1}$ ($Q_{sa} = 20\,nl\,s^{-1}$ and $Q_{sh} = 1000\,nl\,s^{-1}$). With a mean diameter of HL60 cells smaller than $w$, cellular stress is governed by hydrodynamic shear and normal forces and not by the liquid–liquid interface. Analyzing more than 6000 cells, we find a median cell deformation $d = 0.017 \pm 0.007$ (median ± std. dev.; Fig. 5c, top graph), which enables applying the analytical model published earlier to predict the Young's modulus by solving the full flow profile around the cell inside the microfluidic chip[27]. We obtain $E = 1.37 \pm 0.55\,kPa$ (Fig. 5c, top inset). A statistical analysis of three experimental replicates for control ($n = 19{,}018$ cells), DMSO vehicle (0.25% (v/v); $n = 23{,}612$), and $1\,\mu M$ CytoD ($n = 15{,}245$) reveals no difference between the control ($E = 1.5 \pm 0.1\,kPa$ (mean ± SEM)) and the

vehicle ($E = 1.58 \pm 0.05\,kPa$), but a significant reduction in the elastic modulus for cells treated with CytoD ($E = 1.25 \pm 0.06\,kPa$) compared with the control ($p < 0.04$) and the DMSO control ($p < 0.0004$; Fig. 5c, lower graph and Supplementary Fig. 10).

**High-throughput tissue mechanics.** We reveal the full potential of virtual fluidic channels by studying the impact of single-cell mechanics on tissue properties. Such a comparative analysis is of high importance, e.g., to understand emergent effects in cellular co-culture systems in regenerative medicine. This has so far only been possible using atomic force microscopy with its ability to bridge length scales from a few micrometers to a millimeter scale. Here, we aim to utilize the capability of virtual channels to modify the constriction diameter and to tailor the hydrodynamic stress distribution toward high-throughput characterization of HEK293T cells and spheroids inside the glass cuvette of a commercial flow cytometer.

Spheroids have been cultured for multiple days in agarose micro-molds with an initial seeding density of ~30 cells per well. We observe an average spheroid diameter of $82 \pm 23\,\mu m$ (mean ± std. dev.; $n = 5$) at day 1, which increases to $134 \pm 19\,\mu m$ ($n = 13$) at day 4 (Fig. 6a, left). Approximating the number of cells inside the spheroids from the volume allows to fit an exponential growth curve to our data, where we find a doubling time of 32 h (Fig. 6a, right).

For studying the contribution of single cells to tissue stiffness we first adjust the diameter of the virtual channel inside the cuvette to $w = 16\,\mu m$ to probe single HEK293T cells by interfacial stress (Fig. 6b, left panel; $114\,\mu M$ MC sample and $5\,mM$ PEG40000 sheath) yielding $\varepsilon_c = 0.09 \pm 0.03$ (median ± std. dev.; Fig. 6b, center panel and Supplementary Fig. 11). The cellular Young's modulus $E$ is calculated under steady-state conditions applying the interface model. Utilizing experimental parameters of $u_c = 2.8\,cm\,s^{-1}$ and $\eta_{sa} = 11\,mPa\,s$ yields an interfacial stress of $\sigma_i = 66\,Pa$, which exceeds the stress originating from Poiseuille flow (Supplementary Fig. 8). From the stress–strain relationship we obtain a mean Young's modulus of $E = 0.73 \pm 0.38\,kPa$ for single HEK293T cells (Fig. 6b, right panel).

Finally, virtual fluidic channels are applied for high-throughput mechanical characterization of HEK293T spheroids as a 3D-tissue model. Cells are cultured into spheroids with a diameter of ~$130\,\mu m$ using agarose micro-molds, where each spheroid contains an average of 650 cells (Fig. 6a, right, blue box plot; see Methods). Taking advantage of the simple relationship between $Q_{sa}$, $Q_{sh}$, $\eta_{sa}$, and $\eta_{sh}$, stable virtual channels of $190\,\mu m$ mean diameter are established inside the constriction of the cuvette (Eq. (2) and Fig. 6c, left panel). While suspended spheroids slightly perturb the liquid–liquid interface during translocation, the low Reynolds number regime ensures a fast recovery within less than two spheroid diameters. The spheroids show the typical bullet-like shape with a mean deformation of $0.038 \pm 0.027$ (mean ± std. dev.; Fig. 6c, center panel), which is significantly higher than the undeformed reference state at $260\,\mu m$ virtual channel width (Supplementary Fig. 12).

For estimating the elastic modulus of spheroids, we first calculate $\sigma_i$ from Eq. (3) utilizing experimental parameters $Q_{sh} = 330\,nl\,s^{-1}$, $u_c = 1.67\,cm\,s^{-1}$, and $w = 190\,\mu m$ as stated above. Assuming a viscosity $\eta_{sh} = 53\,mPas$ from FEM simulations of the full hydrodynamic geometry yields an interfacial stress of $\sigma_i = 20\,Pa$. With a magnitude much smaller than the hydrodynamic shear and normal stress, spheroid deformation is dominated by the parabolic flow profile and the elastic modulus is obtained from the analytical model published earlier[27,43]. This yields $E = 86 \pm 36\,Pa$ (mean ± std. dev.; Fig. 6c, right panel). Analyzing three biological replicates of more than 700 spheroids

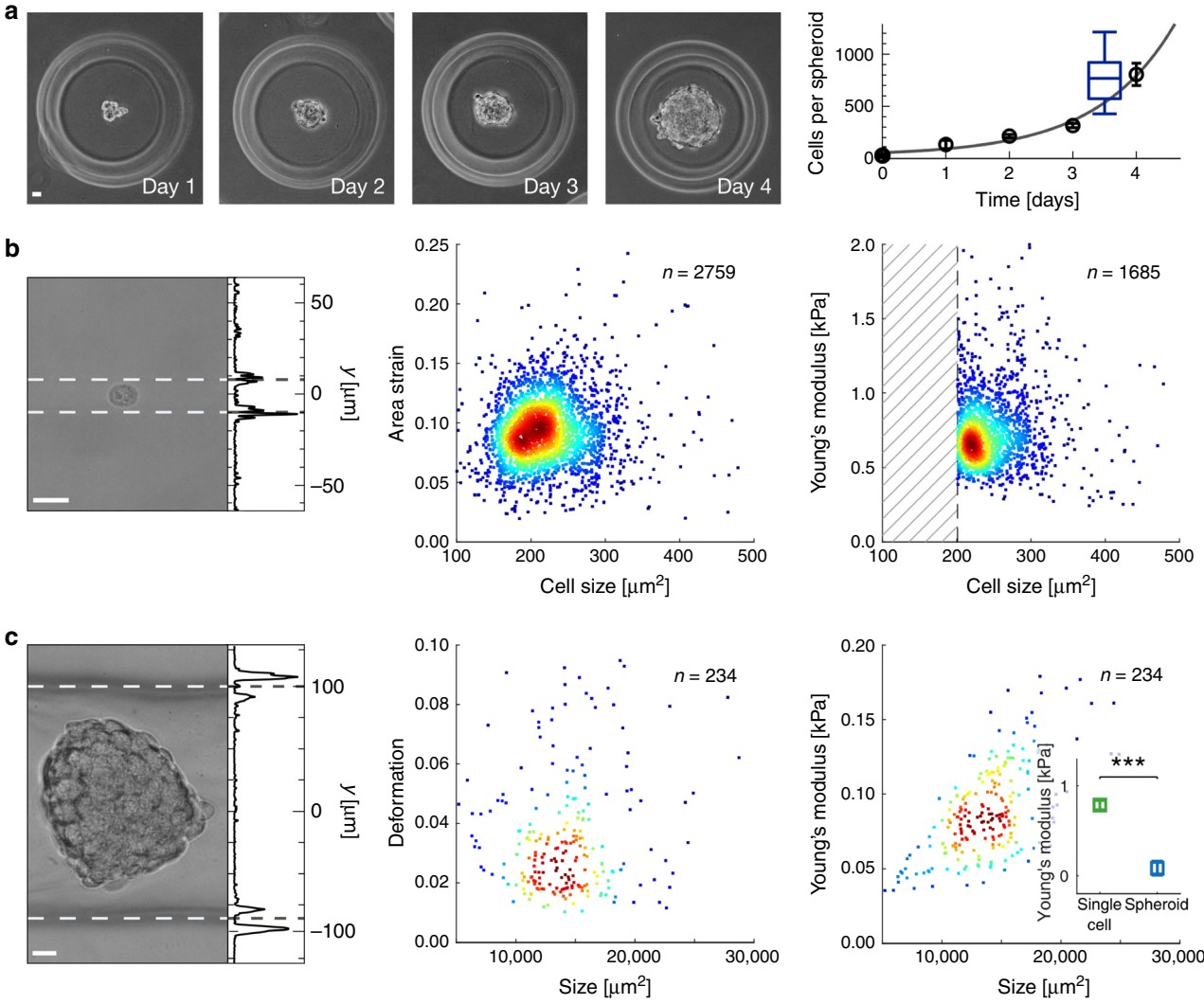

**Fig. 6 High-throughput cell and tissue mechanics inside glass cuvette. a** Bright-field images of HEK293T spheroids obtained at four subsequent days inside agarose mold. Initial seeding density is ~30 cells per micro well (left). Exponential fit (solid line) to the mean cell number (black circles) per spheroid yields a doubling time of ~32 h. Time series has been recorded twice. Distribution of cell number per spheroid indicated in (**c**, center panel) is represented in blue box plot showing median (center line), 25 and 75 percentiles (lower and upper box bound) as well as 10 and 90 percentiles (whiskers). **b** Representative image of a HEK293T cell in virtual channel of 16 μm diameter (sample 114 μM MC at $Q_{sa} = 20$ nl s$^{-1}$, sheath 5 mM PEG40000 at $Q_{sh} = 1000$ nl s$^{-1}$, left). Corresponding scatter plots of area strain (center) and Young's modulus (right) versus HEK293T cell size. Young's modulus is calculated from area strain assuming an interfacial stress of 66 Pa. Shaded area indicates cells not confined inside the virtual channel. **c** Representative image of HEK293T spheroid in virtual channel of 190 μm width (sample 114 μM MC at $Q_{sa} = 330$ nl s$^{-1}$, sheath 5 mM PEG40000 at $Q_{sh} = 200$ nl s$^{-1}$, left). Projected line plots in (**b**) and (**c**) indicate the squared intensity gradient (arb. units) perpendicular to the flow direction while centers of the intensity maxima identify the virtual channel interfaces (dashed white lines). Corresponding scatter plots of deformation (center) and Young's modulus (right) versus HEK293T spheroid size. Inset in right graph compares Young's modulus of single cells ($E = 0.79 ± 0.06$ kPa) and spheroids ($E = 84 ± 78$ Pa) from three independent biological replicates consisting of $n = 2433$ single-cell and $n = 762$ spheroid measurements ($p = 0.0003$). Scale bar is 20 μm. Color in scatter plots indicates a linear density scale from min (blue) to max (red). Statistical data analysis is performed using linear mixed models and data are represented as mean ± standard error of the mean (***$p < 0.001$).

and 2000 cells, we derive an apparent Young's modulus of 84 ± 78 Pa (mean ± SEM), which is a factor of ten smaller than single cells cultured in a 2D plastic culture dish with 0.79 ± 0.06 kPa (Fig. 6c, right panel inset).

## Discussion

Microfluidics has shown great impact in fundamental and applied biochemical, biomedical as well as biophysical research[21,30,44]. The possibility to control nanoliter sample volumes to perform quantitative molecular and cellular assays of highest sensitivity is one of the great promises of fluidic microtechnology and aims to replace bulk experimental assays[23]. Recent examples include

single molecule and protein sensing, chemical reactions for analyte diagnostics, fluorescence cytometry as well as mechanical cell characterization[21,24,45,46].

In the low Reynolds number regime laminar systems can be categorized into single and multiple phase flows[47–51]. While single aqueous phases have mainly been used to study material properties of molecules and cells[24–26,52], multiphase flows allow to tailor the total reaction volume to the analyte of interest, e.g., to control diffusive processes or for spatiotemporal alignment of particles and cells[21,38]. Interestingly, a combination of both approaches, i.e., modifying the system size and tailoring the hydrodynamic stress distribution to perform rheological

characterization of biological matter across various scales, has not been achieved before.

The latter would be of high importance for the growing field of 3D-tissue assays. In contrast to two-dimensional cell culture systems using plastic petri dishes, 3D models allow to mimic a physiological environment that can be used, e.g., for drug discovery[53], to study thrombi formation in vitro[54] as well as cardiac tissue regeneration[55]. While recent research was focusing on generation of 3D cell cultures, e.g., using hanging drop microplates or spheroid microplates with ultra-low attachment coating, high-throughput characterization of tissue function mainly focused on molecular labels or genomics as well as transcriptomics potentially altering the sample[56].

In contrast to standard assays in cell biology, methods utilizing intrinsic material properties of tissues provide a label-free access to biological function. Originally, these methods have been limited to single cells[6,8] and were mainly based on the interaction between an acoustic, optical or physical probe[57,58]. Recently, atomic force microscopy and micropipette aspiration have been extended toward 3D-tissue assays where they probe mechanical properties locally by integrating over a limited number of cells[59]. However, a high-throughput assay able to impose a homogeneous stress distribution on a tissue that integrates cell-cell and cell-matrix interactions into the stress–strain response remained elusive.

By introducing virtual fluidic channels, we establish a simple, fast and highly flexible method to perform microfluidic experiments independent of chip material and chip geometry. We demonstrate that only two parameters, volumetric flow rate and medium viscosity, are sufficient for on-the-fly modifications of channel cross-sections resulting in constrictions with fluid dynamics comparable to systems built from soft lithography.

Virtual fluidic channels can be formed inside classic microfluidic systems, e.g., PDMS chips, but also inside mesoscopic geometries of centimeter size. Using a glass cuvette from a commercial flow cytometer, we show the formation of a stable liquid-bound constriction with cross-sections that can be changed from a few microns to hundreds of micrometers within seconds.

As one potential application we performed mechanical characterization of HL60 cells in both systems. Proof-of-principle experiments in PDMS chips using real-time deformability cytometry revealed an elasticity of $0.95 \pm 0.01$ kPa (mean $\pm$ SEM) and $0.92 \pm 0.02$ kPa for solid and fluid confinement, respectively, showing no significant differences. Next, we repeated our assay using virtual channels created in a large glass cuvette where we obtained a slightly elevated Young's modulus of $1.5 \pm 0.1$ kPa. The observed differences potentially originate from different levels in deformation amplitudes present in both systems. In fact, the deformation inside the glass cuvette is so small that deviations of the initial cell shape from an ideal circle (circularity = 1) might impede application of the analytical model that requires finite cell shape changes in shear flow[27,43].

For solving this discrepancy, we demonstrate that the viscosity mismatch at the liquid–liquid interface leads to an interfacial stress resulting in a cellular strain that allows for extracting material properties from simple creep-compliance experiments. For HL60 cells probed by this liquid–liquid interface of a virtual channel inside a glass cuvette we obtain an elastic modulus of $0.97 \pm 0.03$ kPa, which is in good agreement not only with our PDMS results based on shear flow but also with data published earlier[27,40]. In fact, our proof-of-principle experiments demonstrate that a virtual channel acts like a liquid cantilever that exposes biological samples to a homogeneous 3D stress distribution for probing biological matter on a millisecond scale. In contrast to atomic force microscopy or microconstrictions virtual

fluidic channels enable a label-free analysis without any cell-wall interaction and at high-throughput.

We highlight further potential applications of virtual fluidic channels and provide the first high-throughput rheological characterization of spheroids inside the cuvette of a commercial flow cytometer. Spheroids have gained an increasing importance as in vitro 3D cell culture models in biology closely mimicking in vivo conditions during development[10,32]. In addition, the mechanical microenvironment in a tissue-like structure might also be of importance to understand immune-specific cell response to external stimuli, e.g., during inflammation[60]. While our data on this tissue model confirms previous studies of small sample size[34,35], virtual fluidic channels additionally enable comparative single-cell and tissue rheology of sufficient statistical power appropriate for large-scale sample screenings.

In summary, virtual fluidic channels allow for on-the-fly generation of nearly arbitrary hydrodynamic environments inside large fluidic geometries, e.g., commercial glass cuvettes. We demonstrate that our method provides simultaneous access to material properties of biological samples across various length scales from single cells to tissues, which could be highly relevant for a functional evaluation of engineered tissue grafts in regenerative applications. Ultimately, our approach renders microfluidic experiments independent of soft lithography and extends the parameter space of classical fluorescence-based flow cytometry toward an unbiased cell and tissue analysis utilizing intrinsic material properties.

## Methods

**Sample and sheath fluid.** Sample solution is used as cell as well as spheroid carrier and is made of methylcellulose (MC, approximate molecular weight 88 kDa, Sigma) in PBS−/− (without $Mg^{2+}$ and $Ca^{2+}$). Briefly, MC concentrations of 0.5% (w/v), respectively, 57 μM, and 1.0% (w/v), respectively, 114 μM, are dissolved at room temperature by agitation overnight. The osmolarity of MC solutions is measured to be stable at a physiological range between 275 and 295 mOsm (Fiske 210 Osmometer). The sheath fluid is used to establish a liquid confinement of the sample and is made of polyethylene glycol (PEG) in PBS−/− with a molecular weight of 8000 Da (AppliChem) or 40,000 Da (Serva Electrophoresis). Experiments have been carried out at concentrations of 50 mM (PEG8000) and 2 mM as well as 5 mM (PEG40000).

The shear-rate dependent viscosities of our buffers are obtained from a full rheological characterization using a rheometer in cone-plate geometry (MCR502, Anton Paar). MC solutions reveal a concentration-dependent shear-thinning behavior and the dynamic viscosities follow a simple power law (Fig. 1c):

$$\eta_{MC}(\dot{\gamma}) = n_{1|2} \cdot \left(\frac{\dot{\gamma}}{\dot{\gamma}_0}\right)^{(m_{1|2}-1)} \tag{4}$$

where $\eta_{MC}(\dot{\gamma})$ is the dynamic viscosity of 57 and 114 μM MC at the hydrodynamic shear rate $\dot{\gamma}$, respectively. Here, $n$ denotes a consistency parameter with $n_{57\,\mu M} = 0.078$ Pa s and $n_{114\,\mu M} = 0.60$ Pa s, $m$ is the flow behavior index with $m_{57\,\mu M} = 0.79$ and $m_{114\,\mu M} = 0.64$, while $\dot{\gamma}_0 = 1$ s$^{-1}$ for reference.

All PEG solutions can be described by a Newtonian behavior (Fig. 1c) with a constant viscosity of $\eta_{PEG8000\,(50\,mM)} = 0.106$ Pa s, $\eta_{PEG40000\,(2\,mM)} = 0.015$ Pa s and $\eta_{PEG40000\,(5\,mM)} = 0.154$ Pa s over the entire relevant shear-rate range from 100 to 50,000 s$^{-1}$.

All buffers are sterile filtered (pore size 0.2 μm, VWR) prior to experiments.

**Virtual channel formation.** We use two major geometries to establish virtual channels in fluidic devices: a microfluidic chip assembled from polydimethylsiloxane (PDMS) and sealed with a glass cover slide as well as a mesoscopic glass cuvette commonly used in commercial flow cytometers.

The microfluidic chip is produced using soft lithography (Zellmechanik Dresden) and consists of two inlets, which are connected to a syringe pump (NemeSys, Cetoni). An outlet is used for recollection of cells for potential downstream analysis. Prior to an experiment, two 1-ml syringes (Luer-Lock syringe, BD) with MC solution (sample) and PEG solution (sheath) are assembled on the syringe pump system, which is controlled by the ShapeIn2 software (version 2.0.5, Zellmechanik Dresden). The PDMS chip is first filled with the MC buffer through the sample inlet at 100 nl s$^{-1}$ and subsequently the PEG solution is added through the sheath inlet at 50 nl s$^{-1}$ (Supplementary Fig. 1A). The virtual channel forms immediately inside the squared central constriction of 300 μm length and 30 μm × 30 μm cross-section, can be adjusted by controlling flow rates as well as buffer viscosities and is stable throughout the entire duration of the experiment.

In a second set of experiments, a standard flow cytometry glass cuvette (Sysmex) is used. With a length of 2 cm and a quadratic 360 μm × 360 μm cross-section the constriction of the cuvette exceeds the dimensions of our microfluidic chip by almost a factor of 100. In contrast to the planar geometry in the PDMS system, the sheath flow completely surrounds the sample and results in a three-dimensional confinement (Fig. 4b, left panel). Both inlets are connected to a syringe pump (NemeSys, Cetoni) and 114 μM MC solution (1 ml Luer-Lock syringe, BD) as well as 5 mM PEG40000 solution (10 ml Luer-Lock syringe, BD) are used for sample and sheath, respectively (Supplementary Fig. 1B). For initiating experiments, the cuvette was simultaneously filled with MC at 200 nl s$^{-1}$ through the sample inlet and PEG at 1000 nl s$^{-1}$ through the sheath inlet. The virtual channel forms immediately upon contact and remains stable throughout the entire experiment. The sample flow is not in contact with the surface of the cuvette as it is completely surrounded and confined by sheath buffer.

**Finite element method simulations.** Numerical finite element method (FEM) simulations have been performed using Comsol Multiphysics (version 5.4, Comsol Group) and its computational fluid dynamics and transport packages. For the microfluidic chip the full layout is reconstructed, and one quadrant used for hydrodynamic simulations taking advantage of the two symmetry axes through the center line of the design (Fig. 1a and Supplementary Fig. 1A)[40]. The layout of the glass cuvette is reconstructed following the manufacturer's documentation, and one eighth of the entire structure is simulated utilizing the four symmetry axes through the center line of the design (Fig. 4a and Supplementary Fig. 1B).

FEM simulations are carried out using physics-controlled meshes of an extra fine element size defined by the meshing algorithm. For the microfluidic chip this leads to a meshed volume of ~$6 \times 10^6$ elements with an average element radius of 1.2 μm, for the glass cuvette to ~$1 \times 10^6$ elements with an average element radius of 14.3 μm. For all experiments, the fluid is set to be incompressible and solid walls are defined by a no-slip boundary condition. The fluid is modeled as a superposition of the respective MC and PEG solution material features, normalized to the local concentrations. While laminar inflow at sample and sheath inlet are assigned flow rates under constant pressure conditions, backflow is suppressed, reflecting experimental conditions. Parametric and material sweeps supporting the derivation of the scaling law in Eq. (1) are performed for all sample and sheath materials as well as flow rate combinations and channel geometries (Supplementary Table 1).

FEM simulations to determine the shear stress on a sphere's surface are carried out by simulating a steady-state condition minimizing the difference of the integrated shear stress and the pressure acting on the sphere following ref. [27]. The whole system is set to move by the velocity of the interface $u_i$ found in full-geometry simulations. Simulation of surface stresses as a function of relative cell size $\lambda$ is conducted in the very same manner using respective parameters (Supplementary Table 1).

**Virtual channel characterization.** In both major geometries, PDMS chip and glass cuvette, we use the squared grayscale intensity gradient to identify the interfaces between MC and PEG solutions in the focal plane of our bright-field images. Having a region-of-interest of 250 pixels in length and sufficient height, to capture the entire virtual channel, the mean pixel intensity is calculated for each pixel line. From this one-dimensional intensity distribution perpendicular to the flow direction the squared intensity gradient is derived yielding two local maxima. These are identified as the optical interface position (Fig. 1a, top inset) and are chosen to coincide with the half-maximum MC concentration found by FEM simulations (Fig. 1b) to describe the distinct phases accordingly:

$$c_{MC}(y) \geq \frac{c_{MC, max}}{2} \Leftrightarrow |y| \leq \tilde{w} y_{max} \tag{5}$$

or, equivalently:

$$c_{MC}(y) < \frac{c_{MC, max}}{2} \Leftrightarrow |y| > \tilde{w} y_{max} \tag{6}$$

with $-y_{max} \leq y \leq y_{max}$ and $0 \leq \tilde{w} \leq 1$, $c_{MC}(y)$ is the methylcellulose concentration distribution while $-y_{max}$ and $y_{max}$ are the lateral boundaries of the solid constriction inside the microfluidic chip or cuvette, respectively. The relative width $\tilde{w}$ of the virtual channel is thus defined by the half-maximum concentration $\frac{1}{2} c_{MC, max}$ (Fig. 1a, b).

A simple linear proportionality analysis allows to describe $\tilde{w}$ in our PDMS chip (Fig. 1a, top inset)[38]:

$$\frac{Q_{sa}}{Q_{sh}} = \frac{A_{sa}}{A_{sh}} \frac{u_{sa}}{u_{sh}} = \frac{\tilde{w}}{1-\tilde{w}} \frac{u_{sa}}{u_{sh}} \propto B \frac{\tilde{w}}{1-\tilde{w}} \frac{\eta_{sh}}{\eta_{sa}(\dot{\gamma})} \Rightarrow \frac{Q_{sa}}{Q_{sh}} \frac{\eta_{sa}(\dot{\gamma})}{\eta_{sh}} = B \frac{\tilde{w}}{1-\tilde{w}} \tag{7}$$

with the flow rates $Q_{sa}$ and $Q_{sh}$, the dynamic viscosities $\eta_{sa}$ and $\eta_{sh}$, the cross-sections of the aqueous phases $A_{sa}$ and $A_{sh}$, the mean velocities $u_{sa}$ and $u_{sh}$, and the linear proportionality factor $B$. We fit Eq. (7) to our experimental data (Fig. 1d) with an open parameter $B$ yielding $B = 0.994 \pm 0.047$ (99% confidence interval). Therefore, we safely set $B = 1$, which simplifies Eq. (7) to:

$$\frac{Q_{sa}}{Q_{sh}} \frac{\eta_{sa}(\dot{\gamma})}{\eta_{sh}} = \frac{\tilde{w}}{1-\tilde{w}}. \tag{1}$$

Interestingly, this simple equation is completely independent from the specific polymers, chain lengths, molecular weights, and concentrations.

An analysis of the entire flow rate and viscosity range for sample and sheath yields $0.05 < \frac{Q_{sa}}{Q_{sh}} \frac{\eta_{sa}}{\eta_{sh}} < 2$, where a stable interface can be observed. Beyond this range either sample or sheath flow may completely dominate.

In a similar way, a description can be derived for a circular virtual channel inside a rectangular cross-section:

$$\frac{Q_{sa}}{Q_{sh}} = \frac{A_{sa}}{A_{sh}} \frac{u_{sa}}{u_{sh}} = \frac{\frac{\pi \tilde{w}^2}{4}}{1 - \frac{\pi \tilde{w}^2}{4}} \frac{u_{sa}}{u_{sh}} \propto B \frac{\tilde{w}}{1-\tilde{w}} \frac{\eta_{sh}}{\eta_{sa}(\dot{\gamma})} \Rightarrow \frac{Q_{sa}}{Q_{sh}} \frac{\eta_{sa}(\dot{\gamma})}{\eta_{sh}} = B \frac{\frac{\pi \tilde{w}^2}{4}}{1 - \frac{\pi \tilde{w}^2}{4}}, \tag{8}$$

which simplifies to Eq. (2) for $B = 1$. Notably, Eqs. (2) and (8) show a quadratic behavior for the relative width due to the specific boundary conditions.

Characterization of interface stability has been done by analyzing the diffusion of both polymers, MC and PEG, in PBS. Diffusion coefficients of MC, PEG8000 and PEG40000 in PBS were measured by dynamic light scattering (DLS, Zetasizer Nano-ZS, Malvern Instruments). DLS analyzes the autocorrelation function of back-scattered intensity fluctuations over time. The diffusion coefficients are $D_{MC} = 8 \times 10^{-12}$ m$^2$ s$^{-1}$, $D_{PEG8000} = 2 \times 10^{-10}$ m$^2$ s$^{-1}$, and $D_{PEG40000} = 3 \times 10^{-11}$ m$^2$ s$^{-1}$, respectively. Measurements were carried out at 25 °C at a scatter angle of 173°. Average diffusive displacement of dissolved polymers can be calculated using the solution to Fick's second law $\Delta x = \sqrt{2 D t_{passage}}$ and is <1 μm for each polymer (using typical time to traverse a microfluidic chip constriction $t_{passage}$ = channel length/flow velocity ≈ 2 ms).

**Cell culture.** HL60 cells, a myeloid precursor cell line (courtesy of Dan and Ada Olins) are cultured in RPMI-1640 medium (BioWest) with 10% FCS (Gibco), 1% penicillin/streptomycin (BioWest), and 2 mM L-Glutamin (BioWest) in an incubator at 37 °C under 5% CO$_2$ and 95% ambient air. Every 48 h cells are centrifuged at $200 \times g$ for 5 min (Allegra X-15R, Beckman Coulter) and cells are resuspended to a concentration of ~$1.5 \times 10^5$ cells per milliliter. Experiments are carried out ~36 h after splitting during log-phase. Cells have been checked for Mycoplasma infection, and viability is assessed prior to experiments to ~95% using Trypan Blue.

HEK293T cells as a model system for adherent cells are cultured in DMEM high glucose medium (BioWest) with 10% FCS (Gibco) and 2 mM L-Glutamine (BioWest) in an incubator at 37 °C under 5% CO$_2$ and 95% ambient air. After reaching a confluency of 80%, cells are detached from T25 culture flasks enzymatically by exposure to 1× Trypsin (BioWest) for 2 min, centrifuged at $200 \times g$ for 5 min (Allegra X-15R, Beckmann Coulter), and the supernatant is discarded. Cells are resuspended in cell culture medium to a final amount of $0.8 \times 10^6$ cells per flask. Cells have been checked for Mycoplasma infection and viability is assessed prior to experiments to ~95% using Trypan Blue.

**Spheroid culture.** HEK293T spheroids as a tissue model system are cultured in agarose micro-molds following the manufacturer protocol (Microtissues). Briefly, after autoclaving the micro-molds, 2% (w/v) agarose (Biozym Scientific) is dissolved in 0.9% (w/v) sterile saline and boiled in a microwave oven. After cooling to 60–70 °C, 500 μl molten agarose is added into a 12-series micro-mold, removed after gelation and transferred to a 12-well culture plate and equilibrated with medium.

Cells are initially grown in a T75 culture flask with a seeding density of $2 \times 10^6$ cells. After reaching a confluency of 60–80%, cells are detached enzymatically by exposure to 1× Trypsin (BioWest) for 5 min, centrifuged at $200 \times g$ for 5 min (Allegra X-15R, Beckmann Coulter) and 7000 cells are resuspended in a total volume of 120 μl of cell culture medium inside the agarose micro-mold containing 256 wells. Cells are grown for 3 days reaching a mean spheroid diameter of ~130 μm inside the wells. Each spheroid consists of 650 single cells on average.

The agarose micro-molds are flipped upside down and the spheroids are collected in a 12-well plate submerged in PBS$-/-$ and centrifuged at $50 \times g$ for 5 min (Allegra X-15R, Beckmann Coulter). After collection in a falcon tube and a further centrifugation at $50 \times g$ for 5 min, the pellet is resuspended in MC buffer to a concentration of 1000 spheroids per 200 μl prior to experiments.

**Real-time deformability cytometry.** Real-time deformability cytometry (RT-DC) is used for high-throughput mechanical cell and spheroid characterization. The system is built on a commercial solution (AcCellerator, Zellmechanik Dresden) consisting of an inverted microscope (Axio Observer A.1, Zeiss), a CMOS camera (MC1362, Mikrotron), a microsecond-pulsed LED illumination (L1, Zellmechanik Dresden), and a dedicated syringe pump (NemeSys, Cetoni).

Either a microfluidic chip or a glass cuvette is assembled on the $xy$-stage of the microscope. Cells or spheroids translocate through a constriction and deform by hydrodynamic shear and normal stress[26,27]. Image acquisition and analysis are performed in real-time with a throughput exceeding 1000 cells or 10 spheroids per second (depending on initial concentration) and deformation is quantified using the circularity of each particle:

$$\text{Deformation} = 1 - \text{Circularity} = 1 - \frac{2\sqrt{\pi \text{Area}}}{\text{Perimeter}}. \tag{9}$$

Measurements on single cells are performed using a 40x objective (PDMS chip: Apochromat, Zeiss; glass cuvette: LD Plan Neofluar, Zeiss) with an optical resolution of 0.34 μm per pixel, and on spheroids using a 20x objective (LDC, Zeiss) with 0.68 μm per pixel.

**Measurement protocol**. Prior to RT-DC experiments using microfluidic chips, the entire fluidic system is first filled via the sample inlet using 57 µM MC solution at a flow rate of 100 nl s$^{-1}$. Next, 50 mM PEG8000 solution is added through the sheath inlet at a flow rate of 50 nl s$^{-1}$. Upon contact of both aqueous phases in the entrance region of the constriction, a distinctive virtual channel forms immediately, which can be identified by the optical phase difference at the interface (Fig. 1a). After stabilizing the fluid flow for 3 min, sample and sheath are adjusted to the experimental flow rates and equilibrated for 2 min.

In a typical RT-DC experiment, inlet tubing is detached after virtual channel formation and 100 µl of cell suspension in MC buffer at a concentration of ~10$^6$ cells per milliliter are drawn using the syringe pump. The tube is reconnected and the experimental flow rates are applied. RT-DC of single cells is performed as published[26]. Briefly, deformation and size of several thousand single cells inside the virtual channel are analyzed. Measurements are carried out at flow rates of $Q_{sa}$ = 10 nl s$^{-1}$ and $Q_{sh}$ = 30 nl s$^{-1}$ (sample and sheath MC (57 µM), 20 µm × 20 µm cross-section PDMS chip), $Q_{sa}$ = 90 nl s$^{-1}$ (sample MC (57 µM)) and $Q_{sh}$ = 4 nl s$^{-1}$ (sheath PEG8000 (50 mM), virtual channel of 21 µm × 30 µm cross-section in the constriction of PDMS chip of 30 µm × 30 µm cross-section). In addition, a reservoir measurement, i.e., before entering the (virtual) channel, is taken for reference.

Prior to experiments using the glass cuvette, the full fluidic system is cleaned manually by flushing both inlets with 10 ml of 70% isopropanol and de-ionized water. This procedure also ensures a complete degassing of the system, which is an essential requirement for stable virtual channels. Next, the glass cuvette is simultaneously filled with 114 µM MC at 200 nl s$^{-1}$ via the sample and 5 mM PEG40000 at 1000 nl s$^{-1}$ via sheath inlet. Upon virtual channel formation inside the central constriction, the system is stabilized for 10 min before experimental flow rates are applied and then equilibrated for additional 2 min.

In a typical RT-DC experiment, the sample is resuspended in 1 ml of 114 µM MC to a concentration of ~10$^6$ cells per milliliter or 5 × 10$^3$ spheroids per milliliter. Long-term virtual channel stability is achieved by preparing a biological sample solution in a separate syringe/tubing system and replacing the initial set manually. RT-DC measurement are carried out 3 mm inside the constriction of the cuvette and by controlling the flow rates to tune the virtual channel to a diameter of 20 µm using flow rates $Q_{sa}$ = 20 nl s$^{-1}$ and $Q_{sh}$ = 1000 nl s$^{-1}$ (cells) and to a diameter of 190 µm using flow rates $Q_{sa}$ = 330 nl s$^{-1}$ and $Q_{sh}$ = 200 nl s$^{-1}$ (spheroids). In addition, a reservoir measurement is conducted in a virtual channel of 88 µm width for single-cell samples and 260 µm width for spheroid samples as a reference. For each condition several thousand cells and hundreds of spheroids are analyzed, respectively.

Data acquisition, image analysis, and the control of the syringe pump are performed by using the ShapeIn2 software (version 2.0.5, Zellmechanik Dresden).

**Compound treatment of HL60 cells**. The HL60 cells were treated with cyto-chalasin D (CytoD) to inhibit filamentous actin polymerization (Sigma-Aldrich). Cell concentration was adjusted to 10$^6$ cells in 100 µl of 57 µM MC and CytoD dissolved in dimethyl sulfoxide (DMSO; Sigma-Aldrich) was added to a final concentration of 1 µM[61]. Prior to the experiment cells were incubated for 20 min at 37 °C (5% CO$_2$ and 95% ambient air). During measurement DMSO and CytoD concentration is maintained constant to avoid the reversibility of the effect[62].

**Analytical models for cell mechanical properties**. For determining the elastic properties of cells and spheroids inside the glass cuvette we model the full virtual channel inside a cross-section of 360 µm side length. The relative diameter $\bar{w}$ of the virtual channel is obtained from experimental data (Fig. 1a, top inset), the velocity at the liquid–liquid interface is extracted from FEM simulations assuming Stokes flow and used as a moving wall boundary condition. Placing a sphere of radius $r_{sphere}$, representing either a cell or a spheroid, in the centre of the virtual channel allows for extracting a mean surface shear rate for all experimental conditions. The dynamic viscosity of the MC solution is then determined by using Eq. (4).

For deriving the elastic modulus of cells and spheroids, respectively, we provide two different analytical models.

The first approach is based on an analytical model introduced earlier[27]. Briefly, for a moving object inside the (virtual) channel we assume a steady-state defined by the balance of shear and normal forces on its surface. Expansion of the Stokes equation in the appropriate geometry and coupling to linear elasticity theory allows to predict surface displacement of an initially round object. Our model shows for a given relative size $\lambda = r_{sphere}/R$, where $r_{sphere}$ is the object's radius and $R$ is the (equivalence) radius of the constriction, that the characteristic stress scales with the viscosity of the surrounding medium as well as with object velocity and is a function of the surface position. Since deformation is governed by shear and normal forces originating from the Poiseuille flow inside the microfluidic channel, the steady-state object shape resembles a bullet with finite front radius, which is best characterized by its circularity. Assuming objects of varying bulk elastic modulus $E$, cell deformation can be predicted for a given characteristic stress and be used to generate a lookup table to solve the inverse problem, i.e., to predict $E$ from cell deformation.

The second model is based on the reduction of viscous friction for material transport in pipes and is described by an interfacial stress $\sigma_i$ at the liquid–liquid interface of co-flowing solutions[41]. Here, we use $\sigma_i$ to impose a surface stress on

objects confined in a virtual channel and to induce a deformation with an amplitude that depends on the elastic modulus of the object, its velocity, the virtual channel radius and medium viscosity (Eq. (3)). In contrast to the above model utilizing hydrodynamic shear and normal forces where deformation is quantified by circularity, the interfacial stress acts perpendicular to the direction of flow and leads to an ellipsoidal deformation, that can be described by an area strain.

The area strain $\varepsilon_c$ is calculated from the cross-sectional area of the deformed cell $A(t)$. Assuming volume conservation and a spherical reference cell shape the area of the undeformed cell $A_0$ can be approximated and the strain obtained from:

$$\varepsilon_c = \frac{A(t) - A_0}{A_0}. \tag{10}$$

The one-dimensional elongation in the direction of flows allows to consider the shape as the steady-state strain in a Kelvin–Voigt model and the elastic modulus can be calculated from $\sigma_i = E\varepsilon_c$.

In principle both, the interfacial stress model and the Stokes equation in combination with linear elasticity theory, can be used to extract elastic properties from RT-DC measurements. Our results suggest that the applicability depends on the relative object size $\lambda$ inside the constriction. While for large $\lambda$, e.g., cells inside a PDMS channel, surface stress is dominated by the hydrodynamic shear and normal forces originating from Poiseuille flow, deformation in geometries with small $\lambda$ is governed by the interfacial stress only and shear stress can be neglected.

We investigate the dependency of hydrodynamic shear and interfacial stress on $\lambda$ by performing FEM simulations for different flow rates of 0.53 and 1.02 µl s$^{-1}$ and the corresponding viscosities of 53 and 11 mPa s for spheroid and cell measurements, respectively (Supplementary Fig. 8). While the interfacial stresses (dashed lines) do not depend on $\lambda$, we find an increasing peak stress for increasing relative object size. Here, $\sigma_i$ has been derived from Eq. (3) utilizing typical experimental parameters of $w$ = 14 µm, $u_c$ = 2.8 cm s$^{-1}$, and $u_a$ = 0.8 cm s$^{-1}$ for single HL60 cells (blue dotted line) and $w$ = 190 µm, $u_c$ = 1.67 cm s$^{-1}$, and $u_a$ = 0.38 cm s$^{-1}$ for spheroids (black dotted line), respectively. For single cells, we find that $\sigma_s > \sigma_i$ for $\lambda > 0.9$, where $\sigma_s$ is the hydrodynamic shear stress, while for spheroids we obtain $\sigma_s > \sigma_i$ for $\lambda > 0.5$.

For choosing a relative size ratio where $\sigma_i$ dominates, we postulate that the interfacial stress has to exceed the hydrodynamic stress by a factor of 10 or higher. Applying the experimental conditions stated above this results into a threshold of $\lambda < 0.4$ for single cells while spheroid deformation is always governed by shear stress.

**Viscosity calibration**. Viscosity calibration was performed by using beads of fixed elastic modulus $E$ = 1.5 ± 0.5 kPa[42]. Measuring cell deformation inside a virtual channel of $w$ = 16 µm and utilizing the stress–strain relationship $\sigma_i = E\varepsilon_c$ allows to calculate $\eta_{sa}$ from Eq. (3).

**Statistical data analysis**. Statistical significance is calculated based on independent experimental replicates using linear mixed models[63]. Here, a pairwise comparison between two groups is utilized to attribute differences in an observable to random and fixed effects, respectively. Random effects represent systematic and stochastic measurement bias, e.g., alterations in virtual channel width, while fixed effects account for the actual effect size, i.e., the increase or decrease in an experimental quantity. Statistical significance is obtained assuming two models, one with and one without the fixed effect and the maximum likelihoods are calculated. Using the likelihood ratio and applying Wilks' theorem, stating that the distribution of the test statistic −2 log (likelihood ratio) approaches a $\chi^2$ distribution with a number of degrees of freedom corresponding to the dimensionality difference of the two models, the corresponding $p$-values are determined[64].

Sample pairing is performed by utilizing experimental replicates from different measurement days or, if carried out at the same day, linear mixed model analysis is combined with a permutation test of all samples where the $p$-value with the lowest significance is reported. Multiple comparisons are not corrected for as parameters are not analyzed in parallel.

All experiments have been repeated at least three times independently with a sample size of $n > 100$ for spheroids and $n > 1000$ for cell measurements. The corresponding $p$-values are reported in Supplementary Table 2.

**Reporting summary**. Further information on research design is available in the Nature Research Reporting Summary linked to this article.

# Data availability

The datasets generated in this study are available from the corresponding author upon reasonable request. The file formats of the raw rata are AVI for videos, TDMS and RTDC (HDF5) for RT-DC data, and MPH for Comsol Multiphysics simulations.

# Code availability

Statistical data analysis was performed using the software ShapeOut (version 0.8.7, Zellmechanik Dresden, https://github.com/ZELLMECHANIK-DRESDEN/ShapeOut) and Matlab (version R2017a, Mathworks). The source code for FEM simulations and analysis (Comsol Multiphysics, version 5.4 and Matlab version R2017a, Mathworks) is available from the corresponding author upon reasonable request.

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

## Acknowledgements

We thank Paula Büttner, Sophie Schilling and Patrick Bohn for experimental support and are grateful for valuable discussions with Mihaela Delcea. We kindly thank Jochen

Guck for the glass cuvette loan. We gratefully acknowledge financial support from the German Federal Ministry of Research and Education (ZIK grant to O.O. under grant agreement 03Z22CN11) and the German Center for Cardiovascular Research (postdoc startup grant 81X3400107 to O.O.). This work has also been funded by the Deutsche Forschungsgemeinschaft (DFG, German Research Foundation) 231396381/GRK1947.

## Author contributions

O.O. conceived the project. F.C. developed the experimental setup. F.C. performed finite element method simulations. F.C. and M.H.P. carried out rheological buffer characterization. P.N. analyzed interface stability. F.C., M.H.P., B.F. and D.B. designed and performed experiments on HL60 and HEK293T cells. V.A.D. and R.H.P. developed the spheroid assay. F.C. and M.H.P. carried out cell and spheroid experiments. Y.K. performed compound treatment. F.C., M.H.P., B.F. and O.O. performed statistical data analysis. O.O. supervised the project. F.C. and O.O. wrote the manuscript. All authors reviewed the manuscript.

## Competing interests

M.H.P., P.N., V.A.D., D.B., R.H.P., and Y.K. declare no competing interest. O.O. is shareholder of Zellmechanik Dresden GmbH distributing real-time deformability cytometry. F.C., B.F., and O.O. filed a patent application for virtual fluidic channels (Applicant: University Greifswald; inventors: Fabian Czerwinski, Bob Fregin, and Oliver Otto; PCT Application; application number: PCT/EP2018/075605).
