## [Peer Review File · Nature Communications]

Reviewers' comments:

Reviewer #1 (Remarks to the Author):

The authors present the use of viscous co-flowing sheath streams as a method to adjust the shear forces being applied to cells to measure their mechanical properties. Given this main focus the paper does not feel properly motivated or scoped. There are large claims of how virtual channels can drastically improve fabrication of microfluidic devices, but this is only shown using the RT-DC technique. With this application only in hand it is hard to claim this is an improvement that will span across all of microfluidics. The findings they obtain from the flexible 'virtual' RT-DC system might be interesting to the Nature Communication community, but I'm not sure that the generalization of virtual microfluidic walls is properly supported to be compelling yet to a broader set of readers.

Specific notes:

A key important point of the work is the potential ability to tune the dynamic range for different size/stiffness of cells in order to evaluate some perturbation to cell mechanical properties. However, the work only shows that the overall deformability of cells changes with different outer sheath flows. How does this change in lumen size change the sensitivity before/after a mechanical perturbation to a cells, e.g. from a cytoskeletal inhibitor. Alternatively, for two cell types that have different mechanical properties, how does the tuning of the virtual channels affect the ability to differentiate these two cell types? These are key experiments that appear to be missing for the virtual microfluidic system.

The authors claim in the manuscript that they can tailor their hydrodynamic stress in seconds, but in the Methods, they mention running the flows for minutes in order to stabilize flow.

On line 113 and elsewhere, the authors make a note that the cells do not perturb the interface. Is this an issue? What are the consequences?

The authors make a leap relating hydrodynamic stress and channel width that they have established in previous papers. New readers would benefit from a simple explanation in this paper.

Figure 2A would be improved if more similar cells were selected for comparison. It appears that the middle image, which is supposed to be the virtual equivalent of the PDMS device, is not deformed as well. But the image on the right shows a deformed cell that matches the PDMS device, although the virtual channel is narrower.

Additionally, in Figure 2A, the right data with the narrowest virtual channel shows deformation patterns with cell size that is a nonlinear transform of the original deformation profile with the PDMS device. Is there a PDMS device of similar width for comparison to illustrate that the authors are getting equivalent results?

Figure 2B shows a change in the interface between streams along the length of the channel. Are the streams diffusing, therefore altering the virtual channels, and ability to deform the cells?

The authors end the first section with an incomplete discussion on the fact that the sample stream does not have virtual walls in the z direction or is not sufficiently confined (line 121). What is the implication of this? How does the ability to deform cells change as a result? This seems to be a transition to the next section where a cuvette is used, but it is not motivated and discussed well such that the reader could benefit from the authors' learnings.

On page 360, the derivation of Equation 1 is discussed. The equation is fit, including "an open parameter B yielding $B=0.994$. Therefore, we safely set $B=1$." What is the confidence interval for

B? Does it include 1? As it stands, the authors should say something like, "Therefore, we set $B=1$ for convenience", since they have not shown that it is a safe equivalence.

On line 77, the authors should just separate the variables Q_{sa} and Q_{sh} which is confusing at first. They should just separate them as they do in the rest of the paper: Q_{sa} and Q_{sh} etc.

Reviewer #2 (Remarks to the Author):

This manuscript describes the use of fluid-flow induced shear to measure cell deformability. I appreciate the authors' approach in this paper as it not only demonstrates utility in a microfluidic system but also shows how the concept can be integrated within a commercial type device. Where I struggle is the novelty as much of what is shown has been studied extensively in the past. RT-DC, hydrodynamic focusing, the influence of viscosity on hydrodynamic focusing, hydrodynamic focusing for stretching (of both polymers and cells) have all been previously investigated. Perhaps the advance is in the integration of methods and ideas but I do not think the authors make a strong case along these lines. I believe that, until the authors clarify the novelty of the work presented here, the manuscript is unsuitable for publication in Nature Communications.

For example, how does the concept of "virtual fluidic channels" differ from hydrodynamic focusing, a field that goes back many years [Knight, J. et al. (1998). *Physical Review Letters*, 80(17), 3863–3866; Wu, Z., et al. (2005). *Sensors and Actuators B: Chemical*, 107(2), 965–974]? The authors suggest the streams are "immiscible"; however the fluids here are both aqueous solutions and, though diffusion is slowed down with high molecular weight polymer solutions, I would expect that streams should merge over some undescribed length scale. If it is the issue of differing viscosity between sample and sheath streams, this has been both previously observed experimentally and modeled [Wu, Z. et al. (2005). *Sensors and Actuators B: Chemical*, 107(2), 965–974; Brown, et al. (2006), 88(13), 134109–134109], previous work that includes the authors' scaling relationship presented as Eq. 1.

With regards to stretching and hydrodynamic focusing, I believe the authors need to provide some additional background on the difference between their approach and previous methods. Using hydrodynamic focusing for stretching has been demonstrated for DNA [Wong, P. et al. (2003). *Journal of Fluid Mechanics*, 497, 55–65] and cells [Dudani, J. S. et al. (2013). *Lab on a Chip*, 13(18), 3728–7] and the novelty of using a focusing geometry here for stretching is not clear. Extension of focusing methods to flow cytometers has also been investigated [Lee, G. B., et al. (2001). *Transactions of the ASME*, 123, 672–679.]; however, I am not aware of studies using differing viscosities in larger geometries. Given measurable deformations and other limitations, can the authors comment on the range of cell sizes and stiffnesses that can be studied with this approach?

The idea that using flow focusing as a means of dynamically altering the channel to avoid repeated rapid prototyping is not strongly supported here. The authors note that "system sizes...need to be tailored to the analyte of interest, which is of specific importance in single-molecule and single-cell studies."; however, the concept seems underdeveloped and is used sparingly with the data collected. In Fig 2, the authors do test two different fluidic widths, see a different cell shape, and then attribute this to "...a hydrodynamic stress contribution from the effective surface tension at the liquid-liquid interphase.". Is there anything to support this suggestion? Presumably there would be different velocity and fluid shear stress profiles at the different widths which would deform the cells differently. Was this calculated?

How is imaging performed in the capillary? Does the refractive index contrast between sheath and core streams affect imaging in any significant way?

Reviewer #3 (Remarks to the Author):

An interesting manuscript that describes the first use of multiphase co-flow to quantify cell mechanics. Compared with single-phase micro channel method, "virtual channel" allows real-time adjustment of the fluidic dynamics, which may increase the efficiency in assay.

Concerns include: 1) The authors put too much emphasis on the fabrication of the microfluidic devices as labor-intensive and time-consuming. I can understand where is their coming from but I don't think the discussion is fully accurate or hits the major issue - as it really distracts from the main story. Actually, the two major drawbacks from single-phase micro channels in study of single-cell mechanics are: First, it's hard to decouple cell-wall interaction or adhesion from deformation on the influence of transit time. Second, clogging of the channels from cells with a volume larger than the upper limit from a design. I would suggest reframing this section to focus more accurately on aspects important to this work. (2) Microfluidic channel is only one of the available techniques in this area. It would be helpful to provide a bit more context for the general reader as to how this method fits with the broader slate of techniques, e.g., like electrical, optical or acoustic methods. (3) Finally there is a large body of literature in the area of multiphase laminar flow in microfluidics (and associated simulation) and single cell mechanics in microfluidics. Again would be helpful to better put this work into context.

In summary, while interesting work, it seems more appropriate for a more specialized journal.

Reviewers' comments:

Reviewer #1 (Remarks to the Author):

The authors present the use of viscous co-flowing sheath streams as a method to adjust the shear forces being applied to cells to measure their mechanical properties. Given this main focus the paper does not feel properly motivated or scoped. There are large claims of how virtual channels can drastically improve fabrication of microfluidic devices, but this is only shown using the RT-DC technique. With this application only in hand it is hard to claim this is an improvement that will span across all of microfluidics. The findings they obtain from the flexible 'virtual' RT-DC system might be interesting to the Nature Communication community, but I'm not sure that the generalization of virtual microfluidic walls is properly supported to be compelling yet to a broader set of readers.

Specific notes:

Comment 1

Reviewer:

A key important point of the work is the potential ability to tune the dynamic range for different size/stiffness of cells in order to evaluate some perturbation to cell mechanical properties. However, the work only shows that the overall deformability of cells changes with different outer sheath flows. How does this change in lumen size change the sensitivity before/after a mechanical perturbation to a cells, e.g. from a cytoskeletal inhibitor.

Alternatively, for two cell types that have different mechanical properties, how does the tuning of the virtual channels affect the ability to differentiate these two cell types? These are key experiments that appear to be missing for the virtual microfluidic system.

Author response:

We thank the reviewer for this excellent suggestion. We have been performing experiments on HL60 cells with and without exposure to 1 μ M cytochalasin D. Our results show that the observed changes in cell mechanical properties after depolymerization of filamentous actin do not depend on the experimental system, i.e. a standard PDMS channel or a virtual channel. While deformation and Young's modulus are unaltered, we observe a reduction in cell size for virtual channel measurements. This can be explained by the difference in cross-section. The PDMS chip in our experiments has a cross-section of 20 μ m x 20 μ m while the virtual channel is confined to a width of 21 μ m and a height of 30 μ m. However, our simulations show that the mean surface stress differs by less than 10%, which leads to the same deformation.

In addition we performed a statistical analysis on experimental replicates confirming the observed trend.

We replaced Figure 2 with an updated version and also modified the text:

"Next, we study the capability of virtual channels as a confining constriction for probing mechanical properties of suspended cells. Using the myeloid precursor cell line HL60, RT-DC is performed in a standard PDMS chip of 20 μ m x 20 μ m cross-section (Fig. 2A, left panel) and compared to a virtual channel of 21 μ m width (Fig. 2A, center panel) inside the constriction of a 30 μ m x 30 μ m chip ¹.

While cells display the typical bullet shape and do not perturb the MC-PEG interface (Fig. 2A, insets), mechanical phenotyping in both, plastic chip and virtual channel, reveals distributions of

cell size and deformation which differ in their median values by 2 % or less (Supplementary Fig. 2). Actively increasing the hydrodynamic stress by decreasing the virtual channel width to 14 μm , we observe an increased median deformation (Fig. 2A, right panel) suggesting a hydrodynamic stress contribution from the effective surface tension at the liquid-liquid interphase. Tracking single cells through the full virtual constriction, we find the cells adopt the simulated peak velocity (Fig. 2B). Notably, the geometry of our microfluidic chip having a sheath current from two sides, both connected to the same inlet, always results in virtual channels that is aligned in the centre of the constriction (Fig. 1A and Supplementary Fig. 1A) and confines the cells perpendicular to one direction of flow only, leaving the channel height of 30 μm unaltered (Fig. 1A, bottom inset)."

by:

"Next, we study the capability of virtual channels as a confining constriction for probing mechanical properties of suspended cells. Using the myeloid precursor cell line HL60, RT-DC is performed in a standard PDMS chip of 20 μm x 20 μm cross-section¹² and results are compared to measurements inside a virtual channel of 21 μm width formed in a 30 μm x 30 μm chip (see Methods). Mechanical phenotyping in both, plastic chip and virtual channel, reveals similar distributions in cell size and deformation (Fig. 2A and 2B), cells display the typical bullet shape (Fig. 2A and B, insets) and only slightly perturb the MC-PEG interface (Fig. 2C, Supplementary Fig. 2 and Supplementary Video 2).

Sensitivity of our system towards cytoskeletal modifications has been assessed by exposing HL60 cells to 1 μM cytochalasin D (CytoD, see Methods). For cells inside a PDMS channel the depolymerization of filamentous actin reveals the expected increase in deformation d from $d = 0.062 \pm 0.016$ (median \pm std. dev.) for the control sample and $d = 0.062 \pm 0.015$ for the vehicle control of 0.25% (v/v) DMSO to $d = 0.080 \pm 0.030$ for the 1 μM CytoD treatment (Fig. 2A). Virtual channel measurements agree with these results and yield $d = 0.059 \pm 0.015$ (control), $d = 0.065 \pm 0.015$ (vehicle control of 0.25% (v/v) DMSO) as well as $d = 0.087 \pm 0.023$ (1 μM CytoD) where flow rates have been adjusted to match the stress distribution on the cell surface inside the PDMS chips (Fig. 2B and Supplementary Fig. 3).

A statistical analysis of three experimental replicates summarizes more than 20,000 single cell measurements and confirms in both systems the expected significant increase in cell deformation and decrease in Young's modulus E relative to the vehicle control and wildtype when cells are being exposed to 1 μM CytoD (Fig. 2D, Supplementary Fig. 4 and Supplementary Fig. 5).

Importantly, we find no significant differences in deformation and Young's modulus E comparing results in PDMS and virtual channels. In contrast, a significant decrease in cell size is found when cells are confined by a MC-PEG interface. This observation can be attributed to the geometry of our microfluidic chip having a sheath current from two sides, both connected to the same inlet. The corresponding virtual channel is always aligned in the center of the constriction (Fig. 1A and Supplementary Fig. 1A) confining the cells perpendicular to one direction of flow only and leaving the channel height of 30 μm unaltered (Fig. 1A, bottom inset).

Utilizing FEM simulations, we investigate the impact of a non-squared channel cross-section on the hydrodynamic stress distribution. While a 20 μm x 20 μm PDMS channel leads to a mean shear stress on the cell surface of 970 Pa (Supplementary Fig. 3, left), we find 908 Pa for a virtual channel having 21 μm width and a 30 μm height (Supplementary Fig. 3, right). With a difference below 10% our calculations suggest comparable microfluidic conditions, a conclusion which is also supported by the experimental results (Fig. 2D)."

We also modified the Figure as follows:

and the caption:

Figure 2: Cell deformation in virtual fluidic channel. A Real-time deformability cytometry (RT-DC) on HL60 cells yielding scatter plots of deformation versus cell size. Measurements are taken at indicated position (Fig. 1A grey rectangle). Left scatter plot shows RT-DC data from a standard PDMS chip of 20 μm x 20 μm cross-section using 57 μM MC (sample and sheath solution, flow rates $Q_{sa} = 80 \text{ nl s}^{-1}$, $Q_{sh} = 4 \text{ nl s}^{-1}$). Centre and right scatter plots show RT-DC data from a PDMS chip of 30 μm x 30 μm cross-section with a virtual channel width of 21 μm (centre panel) and 14 μm (right panel) using sample 57 μM MC at $Q_{sa} = 80 \text{ nl s}^{-1}$ and sheath 50 mM PEG8000 at $Q_{sh} = 4 \text{ nl s}^{-1}$ (centre panel) and at $Q_{sh} = 8 \text{ nl s}^{-1}$ (right panel). Insets show representative images. B Time series of a single HL60 cell passing a virtual channel (sample 57 μM MC, $Q_{sa} = 72 \text{ nl s}^{-1}$, sheath 50 mM PEG8000, $Q_{sh} = 8 \text{ nl s}^{-1}$). Average cell velocity within the channel is approximately 20 cm s^{-1} . Scale bar is 10 μm ."

has been modified and replaced by:

Figure 2: Cell deformation in PDMS chip and virtual fluidic channel. A Real-time deformability cytometry (RT-DC) on HL60 cells in PDMS channel yielding scatter plots of deformation versus cell size for control cells (left), DMSO vehicle control (0.25% (v/v), center) and 1 μM CytoD (right). Measurements have been done at a total flow rate of 40 nl s^{-1} in a PDMS chip with a 300 μm long channel and a 20 μm x 20 μm squared cross-section using 57 μM MC for sample and sheath buffer, respectively. **B** RT-DC on HL60 cells in a virtual channel of 21 μm width and 30 μm height yielding scatter plots of deformation versus cell size for control cells (left), DMSO vehicle control (0.25% (v/v), center) and 1 μM CytoD (right). Virtual channel has been established inside a PDMS chip with a 300 μm long channel and a 30 μm x 30 μm squared cross-section using 57 μM MC (sample buffer) as well as 50 mM PEG8000 (sheath buffer). Measurements are taken at indicated position (Fig. 1A, grey rectangle) and flow rates have been adjusted to a total flow rate of 94 nl s^{-1} ($Q_{sa} = 90 \text{ nl s}^{-1}$, $Q_{sh} = 4 \text{ nl s}^{-1}$). Insets show representative cell images. **C** Time series of a single HL60 cell passing a virtual channel (sample 57 μM MC, $Q_{sa} = 72 \text{ nl s}^{-1}$, sheath 50 mM PEG8000, $Q_{sh} = 8 \text{ nl s}^{-1}$). Average cell velocity within the channel is approximately 20 cm s^{-1} . **D** Statistical analysis using linear mixed models of three biological replicates for cell deformation (left), cell size (center) and Young's modulus (right) inside a 20 μm x 20 μm PDMS chip (red) and 21 μm virtual fluidic channel inside a 30 μm x 30 μm PDMS chip (blue). Data compares control cells ($n = 8,024$; PDMS and $n = 4,040$; virtual channel), vehicle control ($n = 8,948$; PDMS and $n = 2,778$; virtual channel) as well as cells after treatment with 1 μM CytoD ($n = 4,148$; PDMS and $n = 1,237$; virtual channel). Scale bar is 10 μm . Error bars represent the standard error of the mean (*, $p < 0.05$; **, $p < 0.01$; and ***, $p < 0.001$)."

Comment 2

Reviewer:

The authors claim in the manuscript that they can tailor their hydrodynamic stress in seconds, but in the Methods, they mention running the flows for minutes in order to stabilize flow.

Author response:

We gratefully acknowledge the question by the reviewer pointing to an inaccuracy in our manuscript. The reviewer is right that we state in the manuscript that we stabilize the flow for three minutes and wait further two minutes after adjusting the sample or sheath flow. This statement refers to measurement protocol for RT-DC where we need stable flow conditions from our syringe pump. Especially, at low flow rates the system of syringe pump, chip and tubing requires some minutes to equilibrate independently if standard PDMS chips or virtual channels are being used.

However, once the flow is established, in fact, we can tailor the width of the virtual fluidic channel and by that the surface stress within seconds. For a better explanation we now changed the first paragraph in the Measurement Protocol section on page 18 from:

“Prior to experiments using microfluidic chips, the entire fluidic system is first filled via the sample inlet using 57 μM MC solution at a flow rate of 100 nl s^{-1} .”

to:

“Prior to RT-DC experiments using microfluidic chips, the entire fluidic system is first filled via the sample inlet using 57 μM MC solution at a flow rate of 100 nl s^{-1} .”

We also added a supplementary video to confirm this statement and adapted the following sentence in the first paragraph of the results section on page 2 from:

“Inside the constriction both aqueous phases can be distinguished optically by a refractive index difference that we use to define the width w of the virtual channel (Fig. 1A, top inset, blue dashed line).”

to:

“Inside the constriction both aqueous phases can be distinguished optically by a refractive index difference that we use to define the width w of the virtual channel (Fig. 1A, top inset, white dashed line and Supplementary Video 1).”

Comment 3

Reviewer:

On line 113 and elsewhere, the authors make a note that the cells do not perturb the interface. Is this an issue? What are the consequences?

Author response:

We gratefully acknowledge the comment and question from the reviewer. The finding that the cells do not perturb the interface allows for a completely new approach to extract cell mechanical properties from microfluidic measurements. This fact has not been discussed in the initial submission of the manuscript. Now we have explored that possibility in detail and highlight our findings in the new Figure 3.

Figure 3: High-throughput cell mechanical measurements inside a flow cytometer glass cuvette. **A** Technical drawing of glass cuvette and PDMS chip to scale for comparison. Arrows indicate length of constrictions of 2 cm for the cuvette and 300 μm for the PDMS device. **B** FEM simulations of concentration distribution for the cross-section of the cuvette performed for sample 114 μM MC at $Q_{\text{sa}} = 200 \text{ nl s}^{-1}$ and sheath 5 mM PEG40000 at $Q_{\text{sh}} = 1,000 \text{ nl s}^{-1}$ yielding a 80 μm virtual channel (left). Arrows indicate the stress σ_i due to the viscosity mismatch at the interface. Adjusting the flow rates to $Q_{\text{sh}} = 50 \text{ nl s}^{-1}$ while keeping Q_{sa} constant, enables increasing the diameter of the constriction to 260 μm (right). **C** Representative image of a HL60 cell in a virtual channel of 88 μm diameter (sample 114 μM MC at $Q_{\text{sa}} = 200 \text{ nl s}^{-1}$, sheath 5 mM PEG40000 at $Q_{\text{sh}} = 1,000 \text{ nl s}^{-1}$, top) and inside a virtual channel of 14 μm diameter (sample 114 μM MC at $Q_{\text{sa}} = 15 \text{ nl s}^{-1}$, sheath 5 mM PEG40000 at $Q_{\text{sh}} = 1,000 \text{ nl s}^{-1}$, bottom). Line plots at top and bottom indicate the squared intensity gradient (arb. units) perpendicular to the flow direction while center between the maxima identify virtual channel interfaces (white dashed lines). **D** Fourier amplitudes of cellular shape modes for $n = 25$ HL60 cells inside a $w = 188 \mu\text{m}$ (yellow) and $w = 14 \mu\text{m}$ (green) virtual channel of a flow cuvette as well as $n = 25$ HL60 cells inside the virtual channel of a PDMS chip shown in Fig. 2 (blue). The inset represents the total harmonic distortion calculated for the cells inside the virtual channel of a cuvette (green), a PDMS chip (blue) and a reference measurement (yellow). **E** Area strain of HL60 cells inside a $w = 14 \mu\text{m}$ virtual channel of a glass cuvette ($Q_{\text{sa}} = 15 \text{ nl s}^{-1}$ and $Q_{\text{sh}} = 1,000 \text{ nl s}^{-1}$, top). Inset shows Young's modulus distribution of HL60 cells calculated from area strain with $\sigma_i = 78 \text{ Pa}$, shaded area indicates fraction of cells not confined by the virtual channel. Bottom graph compares the Young's modulus of three experimental replicates of HL60 cells, a DMSO vehicle control (0.25% (v/v)) and HL60 cells treated with 1 μM CytoD. **F** Deformation of HL60 cells inside a $w = 20 \mu\text{m}$ virtual channel of a glass cuvette ($Q_{\text{sa}} = 20 \text{ nl s}^{-1}$ and $Q_{\text{sh}} = 1,000 \text{ nl s}^{-1}$, top). Inset shows Young's modulus distribution of HL60 cells calculated from a hydrodynamic model considering shear- and normal stress on cell surface in steady-state. Bottom graph compares the Young's modulus of three experimental replicates of HL60 cells, a DMSO vehicle control (0.25% (v/v)) and HL60 cells treated with 1 μM CytoD. Scale bar is 20 μm . Statistical data analysis is performed using linear mixed models and error bars represent the standard error of the mean (*, $p < 0.05$; **, $p < 0.01$; ***, $p < 0.001$).

In our cuvette measurements we demonstrate that the interface exerts a stress on the cell, which leads to a strain response. The fact that the viscosity mismatch between two aqueous phases leads to an interfacial stress has been described before². Here, we utilize this stress to induce a cellular strain and to extract the Young's modulus assuming a steady-state response in a Kelvin-Voigt model. To the best of our knowledge that has not been done before. The strength of this approach is found in its simplicity. In contrast to most hydrodynamic models,

which require solving the full hydrodynamic flow profile around a moving cell to extract material properties, our method only utilizes the linear relationship between stress and strain. We show the validity of our approach by performing experiments on PAA calibration beads with a fixed elastic modulus of $1.5 \pm 0.5 \text{ kPa}$ ³. Utilizing a theory published earlier, which includes the sample viscosity η_{sa} at shear rate $\dot{\gamma}$, the virtual channel with w , the cell velocity u_c and the superficial velocity u_a allows to calculate σ_i as the interfacial stress²:

$$\sigma_i = -\frac{8\eta_{sa}(\dot{\gamma})}{w}(u_c - 2u_a) \quad \#(1)$$

Applying the steady-state relation $\sigma_i = E \varepsilon_c$ enables to extract the interfacial stress from experimental values and subsequently the sample viscosity, which can be compared to our FEM simulations. Here we obtain a viscosity of 11 mPa s, which agrees with the FEM simulations yielding 10.7 mPa s (Response Fig. 1).

Response Figure 1: Mechanical measurements on calibration beads inside glass cuvette. Measurements have been carried out inside a $15 \mu\text{m}$ virtual channel in a glass cuvette of $360 \mu\text{m} \times 360 \mu\text{m}$ cross-section at a flow rate of $1.015 \mu\text{l s}^{-1}$.

Comment 4

Reviewer:

The authors make a leap relating hydrodynamic stress and channel width that they have established in previous papers. New readers would benefit from a simple explanation in this paper.

Author response:

We thank the reviewer for his suggestion. We expanded the explanation of the analytical model. In addition, we now make a comparison between the analytical model based on the Stokes equation and linear elasticity theory and the interfacial stress and the liquid-liquid interface of the virtual channel. The latter was not included in the initial submission of the manuscript. Since we modified Figure 3 and show mechanical measurements utilizing the interfacial stress of virtual channel (see answer above) we also included a discussion of this model in the Methods section in page 17. We changed the paragraph from:

“For determining the elastic properties of cells and spheroids inside the glass cuvette we model the full virtual channel inside a cross-section of $360 \mu\text{m}$ side length. The diameter w of the virtual channel is obtained from experimental data (Fig. 1A, top inset), the velocity at the liquid-liquid

interface is extracted from FEM simulations assuming Stokes flow and used as a moving wall boundary condition. Placing a sphere of radius r_{sphere} , representing either a cell or a spheroid, in the centre of the virtual channel allows to extract a mean surface shear rate for all experimental conditions. The dynamic viscosity of the MC solution is then determined by using Equation 2 and the apparent Young's modulus under steady-state conditions is extracted using an analytical model ⁴. Briefly, assuming steady-state conditions of equal shear and normal forces on the surface of an object translocating a constriction, the hydrodynamic flow field can be coupled to a linear elastic model to derive the Young's modulus."

to

" For determining the elastic properties of cells and spheroids inside the glass cuvette we model the full virtual channel inside a cross-section of 360 μm side length. The relative diameter w of the virtual channel is obtained from experimental data (Fig. 1A, top inset), the velocity at the liquid-liquid interface is extracted from FEM simulations assuming Stokes flow and used as a moving wall boundary condition. Placing a sphere of radius r_{sphere} , representing either a cell or a spheroid, in the centre of the virtual channel allows for extracting a mean surface shear rate for all experimental conditions. The dynamic viscosity of the MC solution is then determined by using Equation 24.

For deriving the elastic modulus of cells and spheroids, respectively, we provide two different analytical models.

Stokes equation and linear elasticity theory

The first approach is based on an analytical model introduced earlier ³¹. Briefly, for a moving object inside the (virtual) channel we assume a steady-state defined by the balance of shear and normal forces on its surface. Expansion of the Stokes equation in the appropriate geometry and coupling to linear elasticity theory allows to predict surface displacement of an initially round object. Our model shows for a given relative size $\lambda = r_{\text{sphere}}/R$, where r_{sphere} is the object's radius and R is the (equivalence) radius of the constriction, that the characteristic stress scales with the viscosity of the surrounding medium as well as with object velocity and is a function of the surface position. Since deformation is governed by shear and normal forces originating from the Poiseuille flow inside the microfluidic channel, the steady-state object shape resembles a bullet with finite front radius, which is best characterized by its circularity. Assuming objects of varying bulk elastic modulus E , cell deformation can be predicted for a given characteristic stress and be used to generate a lookup table to solve the inverse problem, i.e., to predict E from cell deformation.

Interfacial stress at liquid-liquid interface

Earlier work on the reduction of viscous friction for material transport in pipes described an interfacial stress σ_i at the liquid-liquid interface of co-flowing aqueous solutions ³². Here, we use σ_i to impose a surface stress on objects confined in a virtual channel and to induce a deformation with an amplitude that depends on the elastic modulus of the object, its velocity, the virtual channel radius and medium viscosity (Equation 3). In contrast to the above model utilizing hydrodynamic shear and normal forces where deformation is quantified by circularity, the

interfacial stress acts perpendicular to the direction of flow and leads to an ellipsoidal deformation, that can be described by an area strain. The one-dimensional elongation in the direction of flows allows to consider the shape as the steady-state strain in a Kelvin-Voigt-Model and the elastic modulus can be calculated from $\sigma_i = E \varepsilon_c$.

In principle both, the interfacial stress model and the Stokes equation in combination with linear elasticity theory, can be used to extract elastic properties from RT-DC measurements. Our results suggest that the applicability depends on the relative object size λ inside the constriction. While for large λ , e.g. cells inside a PDMS channel, surface stress is dominated by the hydrodynamic shear and normal forces originating from Poiseuille flow, deformation in geometries with small λ is governed by the interfacial stress only as shear stress can be neglected.

We investigate the dependency of hydrodynamic shear and interfacial stress on λ by performing FEM simulations for different flow rates of $0.53 \mu\text{l s}^{-1}$ and $1.02 \mu\text{l s}^{-1}$ and the corresponding viscosities of 53 mPa s and 11 mPa s for spheroid and cell measurements, respectively (Supplementary Fig. 7). While the interfacial stresses (dashed lines) do not depend on λ , we find an increasing peak stress for increasing relative object size. Here, σ_i has been derived from Equation 3 utilizing typical experimental parameters of $w = 14 \mu\text{m}$, $u_c = 2.8 \text{ cm s}^{-1}$ and $u_a = 0.8 \text{ cm s}^{-1}$ for single HL60 cells (blue dotted line) and $w = 190 \mu\text{m}$, $u_c = 1.67 \text{ cm s}^{-1}$, and $u_a = 0.38 \text{ cm s}^{-1}$ for spheroids (black dotted line), respectively. For single cells, we find that $\sigma_s > \sigma_i$ for $\lambda > 0.9$, where σ_s is the hydrodynamic shear stress, while for spheroids we obtain $\sigma_s > \sigma_i$ for $\lambda > 0.5$.

For defining a relative size ratio interval for dominating σ_i , we define that the interfacial stress has to exceed the hydrodynamic stress by a factor of 10 or higher. Applying the experimental conditions stated above this results into a threshold of $\lambda < 0.4$ for single cells while spheroid deformation is always governed by shear stress."

and on page 18:

"Viscosity calibration was performed by using beads of fixed elastic modulus $E = 1.5 \pm 0.5 \text{ kPa}$ ³³. Measuring cell deformation inside a virtual channel of $w = 16 \mu\text{m}$ and utilizing the stress-strain relationship $\sigma_i = E \varepsilon_c$ allows to calculate η_{sa} from Equation 3."

Comment 5

Reviewer:

Figure 2A would be improved if more similar cells were selected for comparison. It appears that the middle image, which is supposed to be the virtual equivalent of the PDMS device, is not deformed as well. But the image on the right shows a deformed cell that matches the PDMS device, although the virtual channel is narrower.

Author response:

We thank the reviewer for this comment and agree that the choice of example cells has not been made well. Following the first suggestion of the reviewer we have now repeated all measurements relevant for Figure 2, expanded our assay and demonstrate that virtual fluidic

channels are also appropriate to quantify cytoskeletal changes in cells. Therefore, we choose representatives from the new dataset.

Comment 6

Reviewer:

Additionally, in Figure 2A, the right data with the narrowest virtual channel shows deformation patterns with cell size that is a nonlinear transform of the original deformation profile with the PDMS device. Is there a PDMS device of similar width for comparison to illustrate that the authors are getting equivalent results?

Author response:

The reviewer is correct pointing out that the deformation of a confined cell is different to one deformation by shear flow. Since we found that point important we investigated the effect of the interface in detail and the results are now found in Figure 3 in the main manuscript. We would also like to refer to the Comment 3 of the reviewer. Here, we present data on deformation of cells inside virtual channels, where the stress originates from the interface between both aqueous phases. The data is presented on page 7 – 10 of the main manuscript:

“In a following step, we transfer our experimental framework to mesoscopic geometries on the centimeter scale. This approach would facilitate microfluidic assays controlled by flow rates and viscosities, fully unconstrained by (soft) lithography and other fabrication techniques. As a proof-of-principle experiment we use a standard flow cytometer glass cuvette of 2 cm length (Fig. 3A and Supplementary Fig. 1B). Although exceeding dimensions of standard microfluidic chips by almost two orders of magnitude, a stable virtual channel of circular cross-section can be formed inside a squared constriction of 360 μm side length. The existence of two steady co-flowing aqueous solutions can be predicted from FEM simulations on the full flow profile (Fig. 3B) and have been confirmed by experimental data (Fig. 3C).

Adapting Equation 1 to the cuvette geometry demonstrates that the virtual constriction can be controlled by sample and sheath flow rates as well as viscosity (see Methods):

$$\frac{Q_{sa} \eta_{sa}(\dot{\gamma})}{Q_{sh} \eta_{sh}} = \frac{\frac{\pi}{4} w^2}{1 - \frac{\pi}{4} w^2} . \#(2)$$

Here, the channel diameter w can be adjusted between a few and more than one hundred micrometers. Having a dynamic size range exceeding one order of magnitude our microfluidic system can be applied to mechanically characterize cells in their reference state, i.e. in large virtual channels at low hydrodynamic stress (Fig. 3C, top) and under a finite hydrodynamic stress in narrow virtual channels (Fig. 3C, bottom)³¹. Interestingly, the sheath inside the constriction of the glass cuvette fully encloses the sample and enables a true three-dimensional cell confinement (Fig. 3B) and a homogeneous stress distribution on the cell surface mimicking the axial symmetric velocity profile (Supplementary Fig. 6, left panel). This is in contrast to standard PDMS chips, where virtual channels are limited to 2.5 dimensions with fixed feature heights given by the planar geometry of the master mold (Fig. 1A, bottom inset and Supplementary Fig 6, right panel; see Methods). The different sheath flow geometries in the cuvette and PDMS system also explain the different scaling in relative virtual channel width w of Equations 1 and 2.

Next, we ask if virtual channels inside a flow cytometer cuvette, i.e., a mesofluidic system, are suited to induce cell deformation. The answer is not obvious since cell surface stress scales with the velocity gradient between the maximum at channel center and zero velocity due to the no-slip boundary condition at the glass surface (Fig. 1B). Assuming typical experimental values for a cuvette of side length 360 μm and 3 cm s^{-1} cell velocity this results in shear rates in the order of 10^2 s^{-1} , which is in contrast to approximately 10^4 s^{-1} for cells inside a 20 μm PDMS chip at a velocity of 10 cm s^{-1} .

Despite two orders in magnitude difference in shear rate we do not only find the majority of cells inside the virtual channel of a cuvette deforming but also observe an ellipsoidal steady-state shape (Fig 3C, bottom and Fig. 3D, green contour), which clearly deviates from the bullet shape in PDMS systems (Fig. 2D and Fig 3D, blue contour) and from the circular reference state (Fig. 3C, top and Fig. 3D, yellow contour). Aiming to understand the physical origin of the alternating cell shapes, we perform a Fourier decomposition of the cell contour analysing the amplitude of the first ten Fourier components ²⁷.

For cells inside a virtual channel of a glass cuvette the second shape mode dominates (Fig. 3D, green) while cells inside the virtual channel of a PDMS chip possess significant contributions of the third and fifth mode (Fig. 3D, blue). Looking at the total harmonic distortion of the first ten Fourier components we find no contribution of the odd modes to ellipsoidal cells (Fig. 3D, inset), which implies a deformation inside the cuvette perpendicular to the direction of flow, only.

The fundamental difference in Fourier spectra points towards stress distributions on the cell surface that originate from different physical principles. We speculate that, while cell deformation in narrow channels, i.e. (virtual channels in) micrometer-sized systems, is mainly driven by shear, cells inside virtual channels of a cuvette are deformed by an interfacial stress σ_i originating from the liquid-liquid interface (Fig. 3B, left and Fig. 3C, bottom).

We investigate this hypothesis by calculating the hydrodynamic shear stress and normal stress on a cell surface inside a glass cuvette using FEM simulations (see Methods). Here, we assume an experimental flow rate of 1.02 $\mu\text{l s}^{-1}$ and a constant viscosity of 11 mPa s . In geometries with a relative cell size $\lambda = r_{\text{sphere}}/R$, where r_{sphere} is the cell radius and R is the equivalent cuvette radius, respectively, peak stress increases with increasing λ for a fixed flow rate (Supplementary Fig. 7). While this relationship is valid for any laminar microfluidic system with a central spherical obstacle, virtual channels also possess a constant stress contribution at the interface resulting from the viscosity mismatch between the two co-flowing aqueous polymer phases. This effect has been described previously for systems where an annular flow of low viscosity, surrounds a core flow of high viscosity, e.g., oil, to reduce friction at the oil-wall interface for power-effective transportation in pipes ³². Under these conditions, σ_i can be derived from the simple expression:

$$\sigma_i = -\frac{8 \eta_{\text{sa}}(\dot{\gamma})}{w} (u_c - 2u_a) , \quad \#(3)$$

where $\eta_{\text{sa}}(\dot{\gamma})$ is the viscosity of the sample flow at shear rate $\dot{\gamma}$, w is the width of the virtual channel, u_c is the cell velocity inside the virtual channel and u_a the superficial velocity proportional to $\frac{Q_{\text{sh}}}{\pi R^2}$.

Utilizing experimental values for HL60 cells at a flow rate of $1.015 \mu\text{l s}^{-1}$ inside a $w = 14 \mu\text{m}$ virtual channel of a cuvette with side length $360 \mu\text{m}$ and $u_c = 2.8 \text{ cm s}^{-1}$ yield an interfacial stress at the liquid-liquid interface of $\sigma_i = 78 \text{ Pa}$ and induces cell deformation perpendicular to the direction of flow (Fig 3C, bottom). Viscosity values of $\eta_{\text{sa}}(\dot{\gamma}) = 11 \text{ mPa s}$ was calculated from FEM simulations and independently confirmed using calibration beads with a fixed elastic modulus of $E = 1.5 \pm 0.5 \text{ kPa}$ ³³ (see Methods). Since Equation 3 allows to consider the ellipsoidal cell shape as the steady-state response to a simple creep-compliance experiment with a constant interfacial stress $\sigma_i = E \varepsilon_c$, the Young's modulus can directly be calculated from the area strain ε_c of the cell (see Methods). Effectively, the liquid-liquid interface acts as a high-frequency liquid cantilever for probing global cell rheological properties on a millisecond timescale such that material properties can be extracted from simple linear models."

Following the suggestion of the reviewer we also performed measurements of single cells in small PDMS chips with a cross-section of $15 \mu\text{m} \times 15 \mu\text{m}$ to investigate if the deformation pattern is comparable to narrow virtual channels. However, the fact that cells touch the channel walls lead to clogging of the device and no data acquisition for a sufficient number of cells had been possible. In fact, we see one of the advantages of virtual channels that interactions between cell membrane and solid walls of the plastic chip are of no relevance.

Comment 7

Reviewer:

Figure 2B shows a change in the interface between streams along the length of the channel. Are the streams diffusing, therefore altering the virtual channels, and ability to deform the cells?

Author response:

We thank the reviewer for raising this important question. We agree that diffusion might play an important role. We approached this question by observing the interface in PDMS chips long channels. We added a stitched image into the Supplementary Material (Supplementary Fig. 2). Here, for a channel length of 1.7 mm we find the change in contrast comparable to Fig. 2C in the first $200 \mu\text{m}$. After $200 \mu\text{m}$ the interface seems to stabilize and does not change until the end of the channel. We attribute the change in contrast to elastic instabilities originating at the liquid-liquid interphase at the channel entrance where polymer are stretched in the direction of flow⁶. After relaxation into a steady-state defined by the Poiseuille flow profile the interface remains stable. This conclusion is also supported by cuvette measurements where we acquire our data approximately 3 mm away from the inlet and observe an interface.

Supplementary Figure 2: Virtual channel in a long PDMS constriction. Stitched image of virtual channel formed by co-flowing sample ($57 \mu\text{M MC}$, $Q_{sa} = 120 \text{ nl s}^{-1}$) and sheath (5 mM PEG40000 , $Q_{sh} = 70 \text{ nl s}^{-1}$) inside a PDMS constriction of 1.75 mm length. Video was acquired at $256 \text{ frames per second}$ and consists of 1280 frames in total. The very wide field of view required refocusing in height. Scale bar $100 \mu\text{m}$.

Comment 8

Reviewer:

The authors end the first section with an incomplete discussion on the fact that the sample stream does not have virtual walls in the z direction or is not sufficiently confined (line 121). What is the implication of this? How does the ability to deform cells change as a result? This seems to be a transition to the next section where a cuvette is used, but it is not motivated and discussed well such that the reader could benefit from the authors' learnings.

Author response:

We greatly acknowledge the comment from the reviewer and the suggestion to explain in detail the confinement in 2D vs. 3D, i.e. in PDMS chips vs. in a glass cuvette. With the restructuring of the manuscript to include cytoskeletal modifications (Figure 2) and a detailed discussion of the interfacial stress (Figure 3), we approach this important point at different positions within the manuscript.

Using FEM simulations, we can show that the stress distribution on the cells does not depend on whether the cell is confined in z -direction or not. We added the following paragraph in the results section on page 5:

“Utilizing FEM simulations, we investigate the impact of a non-squared channel cross-section on the hydrodynamic stress distribution. While a $20 \mu\text{m} \times 20 \mu\text{m}$ PDMS channel leads to a mean shear stress on the cell surface of 970 Pa (Supplementary Fig. 3, left), we find 908 Pa for a virtual channel having $21 \mu\text{m}$ width and a $30 \mu\text{m}$ height (Supplementary Fig. 3, right). With a difference

below 10% our calculations suggest comparable microfluidic conditions, a conclusion which is also supported by the experimental results (Fig. 2D)."

This statement is supported by the experimental comparison of HL60 cells in standard PDMS chips and virtual channels inside the chips. We summarize our results in the new Figure 2 of the manuscript. Here, we show that the distributions of deformation and cell size are similar. In our discussion we also refer to Supplementary Figure 3. Here, we analyse the shear stress distribution on a cell surface in PDMS and a virtual fluidic channel. For the experimental conditions in Figure 2 we obtain a slightly different mean shear stress of 970 Pa (PDMS chip) and 907 Pa (virtual channel) while the stress distribution is comparable.

Supplementary Figure 3: Shear stress on cell surface in PDMS and virtual fluidic channel. Finite element method simulations were performed on full geometries of a 20 μm x 20 μm PDMS chip (left) and a virtual channel of 21 μm width and 30 μm height inside a 30 μm x 30 μm PDMS chip (right). Sample solution was MC (114 μM) and cells were modelled as spheres with a radius of 6.6 μm . Flow rates have been adjusted to $Q_{\text{sa}} = 40 \text{ nl s}^{-1}$ (PDMS channel) yielding a mean shear stress on cell surface of 970 Pa and $Q_{\text{sa}} = 90 \text{ nl s}^{-1}$ (virtual channel) yielding a mean shear stress of 907 Pa.

We further discuss the impact of a 2D vs. 3D confinement when introducing virtual channels inside a commercial flow cytometer cuvette. The design of this mesofluidic device leads to a complete cell confinement where the sheath fully encloses the sample flow. Using FEM simulations of the full geometry and experimental data we demonstrate that the diameter of this virtual channel can be tuned dynamically (Fig. 3B and C). We describe the difference between 2D and 3D confinement on page 6 of the main manuscript, where we added the following paragraph:

"Interestingly, the sheath inside the constriction of the glass cuvette fully encloses the sample and enables a true three-dimensional cell confinement (Fig. 3B) and a homogeneous stress distribution on the cell surface mimicking the axial symmetric velocity profile (Supplementary Fig. 6, left panel). This is in contrast to standard PDMS chips, where virtual channels are limited to 2.5 dimensions with fixed feature heights given by the planar geometry of the master mold (Fig. 1A, bottom inset and Supplementary Fig 6, right panel; see Methods). The different sheath flow geometries in the cuvette and PDMS system also explain the different scaling in relative virtual channel width w of Equations 1 and 2."

Reducing the virtual channel to a diameter smaller than the mean cell size, we observe a stable and by the presence of the cells unaltered interface while cells deforming into an ellipsoidal shape. We investigate this by performing a Fourier decomposition of the cell shape, which highlights a stress perpendicular to the direction of flow and originating from the interface. Utilizing a theory described previously in the literature, we can quantify the magnitude of this stress and use the cellular strain to extract an elastic modulus from a Kelvin-Voigt model². To the best of our knowledge it has not been shown before that a liquid-liquid interface can be used to perform a cell mechanical measurement. The interface acts like

a virtual cantilever probing several hundred of cells per second. With the advantages of applying a simple linear model to extract rheological parameters, the high-throughput, the scalability of the method from single cells to tissues, as well as the non-existing risk of channel blocking we believe that our approach can have an added value for the research community.

For introducing the idea of utilizing a liquid-liquid interface for mechanical measurements we added the following paragraphs on page 8 of the results section:

“Despite two orders in magnitude difference in shear rate we do not only find the majority of cells inside the virtual channel of a cuvette deforming but also observe an ellipsoidal steady-state shape (Fig 3C, bottom and Fig. 3D, green contour), which clearly deviates from the bullet shape in PDMS systems (Fig. 2D and Fig 3D, blue contour) and from the circular reference state (Fig. 3C, top and Fig. 3D, yellow contour). Aiming to understand the physical origin of the alternating cell shapes, we perform a Fourier decomposition of the cell contour analysing the amplitude of the first ten Fourier components ²⁷.

For cells inside a virtual channel of a glass cuvette the second shape mode dominates (Fig. 3D, green) while cells inside the virtual channel of a PDMS chip possess significant contributions of the third and fifth mode (Fig. 3D, blue). Looking at the total harmonic distortion of the first ten Fourier components we find no contribution of the odd modes to ellipsoidal cells (Fig. 3D, inset), which implies a deformation inside the cuvette perpendicular to the direction of flow, only.

The fundamental difference in Fourier spectra points towards stress distributions on the cell surface that originate from different physical principles. We speculate that, while cell deformation in narrow channels, i.e. (virtual channels in) micrometer-sized systems, is mainly driven by shear, cells inside virtual channels of a cuvette are deformed by an interfacial stress σ_i originating from the liquid-liquid interface (Fig. 3B, left and Fig. 3C, bottom).

We investigate this hypothesis by calculating the hydrodynamic shear stress and normal stress on a cell surface inside a glass cuvette using FEM simulations (see Methods). Here, we assume an experimental flow rate of $1.02 \mu\text{l s}^{-1}$ and a constant viscosity of 11 mPa s . In geometries with a relative cell size $\lambda = r_{\text{sphere}}/R$, where r_{sphere} is the cell radius and R is the equivalent cuvette radius, respectively, peak stress increases with increasing λ for a fixed flow rate (Supplementary Fig. 7). While this relationship is valid for any laminar microfluidic system with a central spherical obstacle, virtual channels also possess a constant stress contribution at the interface resulting from the viscosity mismatch between the two co-flowing aqueous polymer phases. This effect has been described previously for systems where an annular flow of low viscosity, surrounds a core flow of high viscosity, e.g., oil, to reduce friction at the oil-wall interface for power-effective transportation in pipes ³². Under these conditions, σ_i can be derived from the simple expression:

$$\sigma_i = -\frac{8 \eta_{\text{sa}}(\dot{\gamma})}{w} (u_c - 2u_a) , \quad \#(3)$$

where $\eta_{\text{sa}}(\dot{\gamma})$ is the viscosity of the sample flow at shear rate $\dot{\gamma}$, w is the width of the virtual channel, u_c is the cell velocity inside the virtual channel and u_a the superficial velocity proportional to $\frac{Q_{\text{sh}}}{\pi R^2}$.

Utilizing experimental values for HL60 cells at a flow rate of $1.015 \mu\text{l s}^{-1}$ inside a $w = 14 \mu\text{m}$ virtual channel of a cuvette with side length $360 \mu\text{m}$ and $u_c = 2.8 \text{ cm s}^{-1}$ yield an interfacial stress at the

liquid-liquid interface of $\sigma_i = 78$ Pa and induces cell deformation perpendicular to the direction of flow (Fig 3C, bottom). Viscosity values of $\eta_{sa}(\dot{\gamma}) = 11$ mPa s was calculated from FEM simulations and independently confirmed using calibration beads with a fixed elastic modulus of $E = 1.5 \pm 0.5$ kPa³³ (see Methods). Since Equation 3 allows to consider the ellipsoidal cell shape as the steady-state response to a simple creep-compliance experiment with a constant interfacial stress $\sigma_i = E \varepsilon_c$, the Young's modulus can directly be calculated from the area strain ε_c of the cell (see Methods). Effectively, the liquid-liquid interface acts as a high-frequency liquid cantilever for probing global cell rheological properties on a millisecond timescale such that material properties can be extracted from simple linear models."

Comment 9

Reviewer:

On page 360, the derivation of Equation 1 is discussed. The equation is fit, including "an open parameter B yielding B=0.994. Therefore, we safely set B=1." What is the confidence interval for B? Does it include 1? As it stands, the authors should say something like, "Therefore, we set B=1 for convenience", since they have not shown that it is a safe equivalence.

Author response:

We thank the reviewer for pointing out that incorrect discussion on the fit. We now state $B = 0.994 \pm 0.047$ with a 99% confidence interval. In addition, we have fixed $B = 1.000$ (black dashed line) and $B = 0.994$ (blue dashed line) and can demonstrate that both corresponding fits cannot be distinguished by eye (Response Figure 2). Having a confidence interval that includes 1.0 we would like to keep the sentence on page 14: "Therefore, we safely set $B = 1 \dots$ "

Response Figure 2: Comparison of fit of Equation 1 to data for B=1.000 (black) and B=0.994 (blue).

Comment 10

Reviewer:

On line 77, the authors should just separate the variables $Q_{sa}|_{sh}$ which is confusing at first. They should just separate them as they do in the rest of the paper: Q_{sa} and Q_{sh} etc.

Author response:

We thank the reviewer for highlighting the important point. We have changed the sentence on page 2 from:

“Analysing different polymers of various chain lengths and concentrations we find a simple functional relationship (Fig. 1D):

$$\frac{Q_{sa} \eta_{sa}(\dot{\gamma})}{Q_{sh} \eta_{sh}} = \frac{w}{1-w}, \#(1)$$

where $Q_{sa|sh}$ are the flow rates of sample respectively sheath flow and $\eta_{sa|sh}$ are the corresponding viscosities (see Methods).”

to:

“We investigated the impact of flow rate and viscosity on virtual channel width as these parameters allow for a precise adjustment of the liquid-liquid interface. Analysing different polymers of various chain lengths and concentrations we find a simple functional relationship reported earlier^{9,22}:

$$\frac{Q_{sa} \eta_{sa}(\dot{\gamma})}{Q_{sh} \eta_{sh}} = \frac{w}{1-w}, \#(1)$$

where Q_{sa} and Q_{sh} are the flow rates of sample as well as sheath flow and η_{sa} and η_{sh} are the corresponding viscosities (see Methods). The viscosity of the sample flow is derived from a power law utilizing experimental shear rates while our sheath solution follows a Newtonian behavior (Fig. 1C, see Methods).”

Reviewer #2 (Remarks to the Author):

Reviewer:

This manuscript describes the use of fluid-flow induced shear to measure cell deformability. I appreciate the authors’ approach in this paper as it not only demonstrates utility in a microfluidic system but also shows how the concept can be integrated within a commercial type device. Where I struggle is the novelty as much of what is shown has been studied extensively in the past. RT-DC, hydrodynamic focusing, the influence of viscosity on hydrodynamic focusing, hydrodynamic focusing for stretching (of both polymers and cells) have all been previously investigated. Perhaps the advance is in the integration of methods and ideas but I do not think the authors make a strong case along these lines. I believe that, until the authors clarify the novelty of the work presented here, the manuscript is unsuitable for publication in Nature Communications.

Author response:

We appreciate the overall positive feedback of the reviewer and agree that the first manuscript did not properly emphasize our work with respect to the existing state-of-the-art. In fact, we not only show for the first time the integration of cell mechanical measurements into a commercial flow cytometer glass cuvette, but also demonstrate in the revised version of the

manuscript that the liquid-liquid interface can be applied to probe cells utilizing the interfacial stress.

Reviewer:

For example, how does the concept of “virtual fluidic channels” differ from hydrodynamic focusing, a field that goes back many years [Knight, J. et al. (1998). *Physical Review Letters*, 80(17), 3863–3866; Wu, Z., et al. (2005). *Sensors and Actuators B: Chemical*, 107(2), 965–974]? The authors suggest the streams are “immiscible”; however the fluids here are both aqueous solutions and, though diffusion is slowed down with high molecular weight polymer solutions, I would expect that streams should merge over some undescribed length scale. If it is the issue of differing viscosity between sample and sheath streams, this has been both previously observed experimentally and modeled [Wu, Z. et al. (2005). *Sensors and Actuators B: Chemical*, 107(2), 965–974; Brown, et al. (2006), 88(13), 134109–134109], previous work that includes the authors’ scaling relationship presented as Eq. 1.

Author response:

We thank the reviewer for highlighting these important references. The reviewer is right, when stating that the above publications all have introduced the concept of co-flowing aqueous solution and the generation of virtual channels. We also appreciate for pointing out that Equation 1 in our manuscript has already been introduced by Knight *et al.*, Wu *et al.* and Brown *et al.*

However, we would like to highlight some important differences that distinguish our work from the existing literature. Knight *et al.* using hydrodynamic focusing of a sample flow into a narrow channel to enhance microfluidic mixing by reduction of diffusion times. There device resembles Fig. 1 in our manuscript of a PDMS device. Here, the confinement is essentially limited to 2.5D, where the height of the virtual channel is given by the mold used for soft lithography or hot embossing. In our manuscript we show that the concept of virtual channels can be transferred to a 3D confinement found in commercial flow cytometry cuvettes where the sheath flow fully encloses the sample flow. We demonstrate that Equation 1 can be adapted to describe the scaling of the virtual channel width:

$$\frac{Q_{sa} \eta_{sa}(\dot{\gamma})}{Q_{sh} \eta_{sh}} = \frac{\frac{\pi}{4} w^2}{1 - \frac{\pi}{4} w^2} . \#(2)$$

In our manuscript we emphasize that finding on page 6 of the Results section as follows:

“Here, the channel diameter w can be adjusted between a few and more than one hundred micrometers. Having a dynamic size range exceeding one order of magnitude our microfluidic system can be applied to mechanically characterize cells in their reference state, i.e. in large virtual channels at low hydrodynamic stress (Fig. 3C, top) and under a finite hydrodynamic stress in narrow virtual channels (Fig. 3C, bottom) ³¹. Interestingly, the sheath inside the constriction of the glass cuvette fully encloses the sample and enables a true three-dimensional cell confinement (Fig. 3B) and a homogeneous stress distribution on the cell surface mimicking the axial symmetric velocity profile (Supplementary Fig. 6, left panel). This is in contrast to standard PDMS chips, where virtual channels are limited to 2.5 dimensions with fixed feature heights given by the planar geometry of the master mold (Fig. 1A, bottom inset and Supplementary Fig 6, right panel; see Methods). The different sheath flow geometries in the cuvette and PDMS system also explain the different scaling in relative virtual channel width w of Equations 1 and 2.”

In addition, the work of Knight *et al.* aims to reduce the diffusion time and enable mixing on a microsecond timescale. Using our combination of polymers, we show that the interface between the two aqueous phases is stable over the entire length of the microfluidic PDMS chip of 300 μm . Given a mean cell velocity of 10 cm s^{-1} this corresponds to a millisecond timescale. In our Supplementary Material (Supplementary Fig. 2) we also demonstrate the generation of stable virtual channels on a millimeter scale inside a PDMS chip. The possibility to generate steady-state liquid-liquid interfaces is essential to transfer the idea of virtual channels from PDMS chips to a commercial flow cytometry geometry.

The work of Wu *et al.* also focuses on micromixing and includes the idea of using different viscosities to control the width of the virtual channel. They introduce an analytical model as well as numerical simulation and provide experimental data. While results show different flow profiles for different experimental conditions the discussion is also limited to 2.5D of devices built by soft lithography or similar approaches. In contrast, Brown *et al.* apply the concept of hydrodynamic focusing to generate an optical wave guide and discuss the importance of refractive indices for performing fluorescence-based assay. As stated above, our work overcomes the limitations of planar microfluidic devices and expands the applications of virtual channels away from microfluidics towards cell-based assays that integrate mechanical properties into flow cytometry.

Reviewer:

With regards to stretching and hydrodynamic focusing, I believe the authors need to provide some additional background on the difference between their approach and previous methods. Using hydrodynamic focusing for stretching has been demonstrated for DNA [Wong, P. et al. (2003). *Journal of Fluid Mechanics*, 497, 55–65] and cells [Dudani, J. S. et al. (2013). *Lab on a Chip*, 13(18), 3728–7] and the novelty of using a focusing geometry here for stretching is not clear. Extension of focusing methods to flow cytometers has also been investigated [Lee, G. B., et al. (2001). *Transactions of the ASME*, 123, 672–679.]; however, I am not aware of studies using differing viscosities in larger geometries. Given measurable deformations and other limitations, can the authors comment on the range of cell sizes and stiffnesses that can be studied with this approach?

Author response:

We gratefully acknowledge the question from the reviewer and appreciate highlighting the important literature. In fact, while we are aware of especially the work of Dudani *et al.* we neglected their results in the discussion of our manuscript. The unifying idea of Wong et al. and Dudani et al. is the utilization of an extensional flow to stretch DNA molecules and cells respectively. This approach beautifully allows to compare e.g. cells under different conditions based on deformation and strain but does not allow to extract material properties, e.g. the elastic modulus.

We follow a different approach. In addition of using virtual channels as a liquid confinement that inhibits cell-wall interaction as well as clogging we also measure cell deformation under steady-state conditions at the end of the channel or several millimeters away from the inlet of the glass cuvette, only. This strategy has the advantage that analytical models can be applied to extract material properties from our data, which is required, e.g., to compare results between different technologies for mechanical cell assays⁴. Most remarkable, we also show that the liquid-liquid interface imposes a stress on the cells under confinement (Fig. 3). We calculate the stress, which originates from the viscosity mismatch between the aqueous phases

and demonstrate that the resulting cellular strain can be used to extract material properties from a simple Kelvin-Voigt model. We verify our results by using calibration beads of fixed elastic modulus of $E = 1.5 \pm 0.5 \text{ kPa}$ ³.

To the best of our knowledge it has never been shown that this interface can be used as a liquid cantilever probing cells on a millisecond timescale with a throughput of several hundred cells per second. A successful transfer of our technology into a commercial flow cytometer device would enable performing mechanical cell assays without the necessity to calculate the full hydrodynamic flow profile of the cell. We integrated our findings into the results section on page 6:

"In a following step, we transfer our experimental framework to mesoscopic geometries on the centimeter scale. This approach would facilitate microfluidic assays controlled by flow rates and viscosities, fully unconstrained by (soft) lithography and other fabrication techniques. As a proof-of-principle experiment we use a standard flow cytometer glass cuvette of 2 cm length (Fig. 3A and Supplementary Fig. 1B). Although exceeding dimensions of standard microfluidic chips by almost two orders of magnitude, a stable virtual channel of circular cross-section can be formed inside a squared constriction of 360 μm side length. The existence of two steady co-flowing aqueous solutions can be predicted from FEM simulations on the full flow profile (Fig. 3B) and have been confirmed by experimental data (Fig. 3C).

Adapting Equation 1 to the cuvette geometry demonstrates that the virtual constriction can be controlled by sample and sheath flow rates as well as viscosity (see Methods):

$$\frac{Q_{sa} \eta_{sa}(\dot{\gamma})}{Q_{sh} \eta_{sh}} = \frac{\frac{\pi}{4} w^2}{1 - \frac{\pi}{4} w^2} \quad \#(2)$$

Here, the channel diameter w can be adjusted between a few and more than one hundred micrometers. Having a dynamic size range exceeding one order of magnitude our microfluidic system can be applied to mechanically characterize cells in their reference state, i.e. in large virtual channels at low hydrodynamic stress (Fig. 3C, top) and under a finite hydrodynamic stress in narrow virtual channels (Fig. 3C, bottom)³¹. Interestingly, the sheath inside the constriction of the glass cuvette fully encloses the sample and enables a true three-dimensional cell confinement (Fig. 3B) and a homogeneous stress distribution on the cell surface mimicking the axial symmetric velocity profile (Supplementary Fig. 6, left panel). This is in contrast to standard PDMS chips, where virtual channels are limited to 2.5 dimensions with fixed feature heights given by the planar geometry of the master mold (Fig. 1A, bottom inset and Supplementary Fig 6, right panel; see Methods). The different sheath flow geometries in the cuvette and PDMS system also explain the different scaling in relative virtual channel width w of Equations 1 and 2.

Next, we ask if virtual channels inside a flow cytometer cuvette, i.e., a mesofluidic system, are suited to induce cell deformation. The answer is not obvious since cell surface stress scales with the velocity gradient between the maximum at channel center and zero velocity due to the no-slip boundary condition at the glass surface (Fig. 1B). Assuming typical experimental values for a cuvette of side length 360 μm and 3 cm s^{-1} cell velocity this results in shear rates in the order of 10^2 s^{-1} , which is in contrast to approximately 10^4 s^{-1} for cells inside a 20 μm PDMS chip at a velocity of 10 cm s^{-1} .

Despite two orders in magnitude difference in shear rate we do not only find the majority of cells inside the virtual channel of a cuvette deforming but also observe an ellipsoidal steady-state shape (Fig 3C, bottom and Fig. 3D, green contour), which clearly deviates from the bullet shape in PDMS systems (Fig. 2D and Fig 3D, blue contour) and from the circular reference state (Fig. 3C, top and Fig. 3D, yellow contour). Aiming to understand the physical origin of the alternating cell shapes, we perform a Fourier decomposition of the cell contour analysing the amplitude of the first ten Fourier components ²⁷.

For cells inside a virtual channel of a glass cuvette the second shape mode dominates (Fig. 3D, green) while cells inside the virtual channel of a PDMS chip possess significant contributions of the third and fifth mode (Fig. 3D, blue). Looking at the total harmonic distortion of the first ten Fourier components we find no contribution of the odd modes to ellipsoidal cells (Fig. 3D, inset), which implies a deformation inside the cuvette perpendicular to the direction of flow, only.

The fundamental difference in Fourier spectra points towards stress distributions on the cell surface that originate from different physical principles. We speculate that, while cell deformation in narrow channels, i.e. (virtual channels in) micrometer-sized systems, is mainly driven by shear, cells inside virtual channels of a cuvette are deformed by an interfacial stress σ_i originating from the liquid-liquid interface (Fig. 3B, left and Fig. 3C, bottom).

We investigate this hypothesis by calculating the hydrodynamic shear stress and normal stress on a cell surface inside a glass cuvette using FEM simulations (see Methods). Here, we assume an experimental flow rate of $1.02 \mu\text{l s}^{-1}$ and a constant viscosity of 11 mPa s . In geometries with a relative cell size $\lambda = r_{\text{sphere}}/R$, where r_{sphere} is the cell radius and R is the equivalent cuvette radius, respectively, peak stress increases with increasing λ for a fixed flow rate (Supplementary Fig. 7). While this relationship is valid for any laminar microfluidic system with a central spherical obstacle, virtual channels also possess a constant stress contribution at the interface resulting from the viscosity mismatch between the two co-flowing aqueous polymer phases. This effect has been described previously for systems where an annular flow of low viscosity, surrounds a core flow of high viscosity, e.g., oil, to reduce friction at the oil-wall interface for power-effective transportation in pipes ³². Under these conditions, σ_i can be derived from the simple expression:

$$\sigma_i = -\frac{8 \eta_{\text{sa}}(\dot{\gamma})}{w} (u_c - 2u_a) , \quad \#(3)$$

where $\eta_{\text{sa}}(\dot{\gamma})$ is the viscosity of the sample flow at shear rate $\dot{\gamma}$, w is the width of the virtual channel, u_c is the cell velocity inside the virtual channel and u_a the superficial velocity proportional to $\frac{Q_{\text{sh}}}{\pi R^2}$.

Utilizing experimental values for HL60 cells at a flow rate of $1.015 \mu\text{l s}^{-1}$ inside a $w = 14 \mu\text{m}$ virtual channel of a cuvette with side length $360 \mu\text{m}$ and $u_c = 2.8 \text{ cm s}^{-1}$ yield an interfacial stress at the liquid-liquid interface of $\sigma_i = 78 \text{ Pa}$ and induces cell deformation perpendicular to the direction of flow (Fig 3C, bottom). Viscosity values of $\eta_{\text{sa}}(\dot{\gamma}) = 11 \text{ mPa s}$ was calculated from FEM simulations and independently confirmed using calibration beads with a fixed elastic modulus of $E = 1.5 \pm 0.5 \text{ kPa}$ ³³ (see Methods). Since Equation 3 allows to consider the ellipsoidal cell shape as the steady-state response to a simple creep-compliance experiment with a constant interfacial stress $\sigma_i = E \varepsilon_c$, the Young's modulus can directly be calculated from the area strain ε_c of the cell (see

Methods). Effectively, the liquid-liquid interface acts as a high-frequency liquid cantilever for probing global cell rheological properties on a millisecond timescale such that material properties can be extracted from simple linear models.”

Reviewer:

The idea that using flow focusing as a means of dynamically altering the channel to avoid repeated rapid prototyping is not strongly supported here. The authors note that “system sizes...need to be tailored to the analyte of interest, which is of specific importance in single-molecule and single-cell studies.”; however, the concept seems underdeveloped and is used sparingly with the data collected. In Fig 2, the authors do test two different fluidic widths, see a different cell shape, and then attribute this to “...a hydrodynamic stress contribution from the effective surface tension at the liquid-liquid interphase.” Is there anything to support this suggestion? Presumably there would be different velocity and fluid shear stress profiles at the different widths which would deform the cells differently. Was this calculated?

Author response:

We thank the reviewer for pointing out that the idea of utilizing the interfacial stress was not very well presented. We would like to refer to the answer of the above question and additionally state that the interfacial stress depends on the cell velocity, the viscosity, and the width of the virtual channel as correctly assumed by the reviewer. We also compared the magnitude of hydrodynamic stress inside the (virtual) channel resulting from the Poiseuille flow profile and the magnitude of the interfacial stress (Supplementary Fig. 7).

Supplementary Figure 7: Peak shear stress on cell surface as function of cross-sectional coverage in a cylindrical fluidic channel. Finite element method simulations were performed on full geometries of a cylinder (radius R) filled with a Newtonian media of viscosities 11 mPa s (blue marker) or 53 mPa s (black marker). A sphere with radius r_{sphere} is centred in the cross-section of a circular channel with a relative radius of λ describing the coverage. The dashed lines show the respective levels of interfacial stress.

Here, we define two different ranges of dominating stress regimes. For small relative cell sizes $\lambda = r_{\text{sphere}}/R$, where r_{sphere} is the cell radius and R is the radius of the channel, deformation is dominated by the interfacial stress, while large λ leads to a deformation by hydrodynamic shear forces. We added a detailed explanation in the Methods section on page 17:

“For determining the elastic properties of cells and spheroids inside the glass cuvette we model the full virtual channel inside a cross-section of 360 μm side length. The relative diameter w of the virtual channel is obtained from experimental data (Fig. 1A, top inset), the velocity at the liquid-liquid interface is extracted from FEM simulations assuming Stokes flow and used as a moving wall boundary condition. Placing a sphere of radius r_{sphere} , representing either a cell or a spheroid, in the centre of the virtual channel allows for extracting a mean surface shear rate for all experimental conditions. The dynamic viscosity of the MC solution is then determined by using Equation 24.

For deriving the elastic modulus of cells and spheroids, respectively, we provide two different analytical models.

Stokes equation and linear elasticity theory

The first approach is based on an analytical model introduced earlier³¹. Briefly, for a moving object inside the (virtual) channel we assume a steady-state defined by the balance of shear and normal forces on its surface. Expansion of the Stokes equation in the appropriate geometry and coupling to linear elasticity theory allows to predict surface displacement of an initially round object. Our model shows for a given relative size $\lambda = r_{\text{sphere}}/R$, where r_{sphere} is the object's radius and R is the (equivalence) radius of the constriction, that the characteristic stress scales with the viscosity of the surrounding medium as well as with object velocity and is a function of the surface position. Since deformation is governed by shear and normal forces originating from the Poiseuille flow inside the microfluidic channel, the steady-state object shape resembles a bullet with finite front radius, which is best characterized by its circularity. Assuming objects of varying bulk elastic modulus E , cell deformation can be predicted for a given characteristic stress and be used to generate a lookup table to solve the inverse problem, i.e., to predict E from cell deformation.

Interfacial stress at liquid-liquid interface

Earlier work on the reduction of viscous friction for material transport in pipes described an interfacial stress σ_i at the liquid-liquid interface of co-flowing aqueous solutions³². Here, we use σ_i to impose a surface stress on objects confined in a virtual channel and to induce a deformation with an amplitude that depends on the elastic modulus of the object, its velocity, the virtual channel radius and medium viscosity (Equation 3). In contrast to the above model utilizing hydrodynamic shear and normal forces where deformation is quantified by circularity, the

interfacial stress acts perpendicular to the direction of flow and leads to an ellipsoidal deformation, that can be described by an area strain. The one-dimensional elongation in the direction of flows allows to consider the shape as the steady-state strain in a Kelvin-Voigt-Model and the elastic modulus can be calculated from $\sigma_i = E \varepsilon_c$.

In principle both, the interfacial stress model and the Stokes equation in combination with linear elasticity theory, can be used to extract elastic properties from RT-DC measurements. Our results suggest that the applicability depends on the relative object size λ inside the constriction. While for large λ , e.g. cells inside a PDMS channel, surface stress is dominated by the hydrodynamic shear and normal forces originating from Poiseuille flow, deformation in geometries with small λ is governed by the interfacial stress only as shear stress can be neglected.

We investigate the dependency of hydrodynamic shear and interfacial stress on λ by performing FEM simulations for different flow rates of $0.53 \mu\text{l s}^{-1}$ and $1.02 \mu\text{l s}^{-1}$ and the corresponding viscosities of 53 mPa s and 11 mPa s for spheroid and cell measurements, respectively (Supplementary Fig. 7). While the interfacial stresses (dashed lines) do not depend on λ , we find an increasing peak stress for increasing relative object size. Here, σ_i has been derived from Equation 3 utilizing typical experimental parameters of $w = 14 \mu\text{m}$, $u_c = 2.8 \text{ cm s}^{-1}$ and $u_a = 0.8 \text{ cm s}^{-1}$ for single HL60 cells (blue dotted line) and $w = 190 \mu\text{m}$, $u_c = 1.67 \text{ cm s}^{-1}$, and $u_a = 0.38 \text{ cm s}^{-1}$ for spheroids (black dotted line), respectively. For single cells, we find that $\sigma_s > \sigma_i$ for $\lambda > 0.9$, where σ_s is the hydrodynamic shear stress, while for spheroids we obtain $\sigma_s > \sigma_i$ for $\lambda > 0.5$.

For defining a relative size ratio interval for dominating σ_i , we define that the interfacial stress has to exceed the hydrodynamic stress by a factor of 10 or higher. Applying the experimental conditions stated above this results into a threshold of $\lambda < 0.4$ for single cells while spheroid deformation is always governed by shear stress."

and on page 18:

"Viscosity calibration was performed by using beads of fixed elastic modulus $E = 1.5 \pm 0.5 \text{ kPa}$ ³³. Measuring cell deformation inside a virtual channel of $w = 16 \mu\text{m}$ and utilizing the stress-strain relationship $\sigma_i = E \varepsilon_c$ allows to calculate η_{sa} from Equation 3."

Reviewer:

How is imaging performed in the capillary? Does the refractive index contrast between sheath and core streams affect imaging in any significant way?

Author response:

For imaging in the capillary we use a 40x long distance objective (LD Plan Neofluar, Zeiss). Since projected cell images do not reveal any differences in size distribution in cuvette virtual channel and PDMS chip (Figure 2 and Figure 3) we assume no influence from the refractive index mismatches at the interfaces.

Reviewer #3 (Remarks to the Author):

Reviewer:

An interesting manuscript that describes the first use of multiphase co-flow to quantify cell mechanics. Compared with single-phase micro channel method, "virtual channel" allows real-time adjustment of the fluidic dynamics, which may increase the efficiency in assay.

Author response:

We thank the reviewer for the positive assessment of our manuscript.

Reviewer:

Concerns include: 1)The authors put too much emphasis on the fabrication of the microfluidic devices as labor-intensive and time-consuming. I can understand where is their coming from but I don't think the discussion is fully accurate or hits the major issue - as it really distracts from the main story. Actually, the two major drawbacks from single-phase micro channels in study of single-cell mechanics are: First, it's hard to decouple cell-wall interaction or adhesion from deformation on the influence of transit time. Second, clogging of the channels from cells with a volume larger than the upper limit from a design. I would suggest reframing this section to focus more accurately on aspects important to this work.

Author response:

We appreciate the comment of the reviewer. It is correct that the main drawbacks of single-phase micro channels are cell-wall interaction and clogging of the device. We believe that the revised version of the manuscript emphasizes both points. We included the completely new Fig. 3 where we describe the utilization of the liquid-liquid interface as a force transducer inducing a cellular strain. We show that derivation of interfacial stress and measuring cell strain allows to calculate an elastic modulus in a simple way. To the best of our knowledge this has not been done before and allows not only for transferring cell mechanical measurement from microfluidic devices into large hydrodynamic geometries following the idea, one size fits all, but also for enabling an integration into a commercial device. We highlight that by demonstrating our results in a flow cytometer cuvette. We now adapted this part and added the following part in the results section on page 7:

"In a following step, we transfer our experimental framework to mesoscopic geometries on the centimeter scale. This approach would facilitate microfluidic assays controlled by flow rates and viscosities, fully unconstrained by (soft) lithography and other fabrication techniques. As a proof-of-principle experiment we use a standard flow cytometer glass cuvette of 2 cm length (Fig. 3A and Supplementary Fig. 1B). Although exceeding dimensions of standard microfluidic chips by almost two orders of magnitude, a stable virtual channel of circular cross-section can be formed inside a squared constriction of 360 μm side length. The existence of two steady co-flowing aqueous solutions can be predicted from FEM simulations on the full flow profile (Fig. 3B) and have been confirmed by experimental data (Fig. 3C).

Adapting Equation 1 to the cuvette geometry demonstrates that the virtual constriction can be controlled by sample and sheath flow rates as well as viscosity (see Methods):

$$\frac{Q_{sa} \eta_{sa}(\dot{\gamma})}{Q_{sh} \eta_{sh}} = \frac{\frac{\pi}{4} w^2}{1 - \frac{\pi}{4} w^2} \quad \#(2)$$

Here, the channel diameter w can be adjusted between a few and more than one hundred micrometers. Having a dynamic size range exceeding one order of magnitude our microfluidic system can be applied to mechanically characterize cells in their reference state, i.e. in large virtual channels at low hydrodynamic stress (Fig. 3C, top) and under a finite hydrodynamic stress in narrow virtual channels (Fig. 3C, bottom)³¹. Interestingly, the sheath inside the constriction of the glass cuvette fully encloses the sample and enables a true three-dimensional cell confinement (Fig. 3B) and a homogeneous stress distribution on the cell surface mimicking the axial symmetric velocity profile (Supplementary Fig. 6, left panel). This is in contrast to standard PDMS chips, where virtual channels are limited to 2.5 dimensions with fixed feature heights given by the planar geometry of the master mold (Fig. 1A, bottom inset and Supplementary Fig 6, right panel; see Methods). The different sheath flow geometries in the cuvette and PDMS system also explain the different scaling in relative virtual channel width w of Equations 1 and 2.

Next, we ask if virtual channels inside a flow cytometer cuvette, i.e., a mesofluidic system, are suited to induce cell deformation. The answer is not obvious since cell surface stress scales with the velocity gradient between the maximum at channel center and zero velocity due to the no-slip boundary condition at the glass surface (Fig. 1B). Assuming typical experimental values for a cuvette of side length 360 μm and 3 cm s^{-1} cell velocity this results in shear rates in the order of 10^2 s^{-1} , which is in contrast to approximately 10^4 s^{-1} for cells inside a 20 μm PDMS chip at a velocity of 10 cm s^{-1} .

Despite two orders in magnitude difference in shear rate we do not only find the majority of cells inside the virtual channel of a cuvette deforming but also observe an ellipsoidal steady-state shape (Fig 3C, bottom and Fig. 3D, green contour), which clearly deviates from the bullet shape in PDMS systems (Fig. 2D and Fig 3D, blue contour) and from the circular reference state (Fig. 3C, top and Fig. 3D, yellow contour). Aiming to understand the physical origin of the alternating cell shapes, we perform a Fourier decomposition of the cell contour analysing the amplitude of the first ten Fourier components²⁷.

For cells inside a virtual channel of a glass cuvette the second shape mode dominates (Fig. 3D, green) while cells inside the virtual channel of a PDMS chip possess significant contributions of the third and fifth mode (Fig. 3D, blue). Looking at the total harmonic distortion of the first ten Fourier components we find no contribution of the odd modes to ellipsoidal cells (Fig. 3D, inset), which implies a deformation inside the cuvette perpendicular to the direction of flow, only.

The fundamental difference in Fourier spectra points towards stress distributions on the cell surface that originate from different physical principles. We speculate that, while cell deformation in narrow channels, i.e. (virtual channels in) micrometer-sized systems, is mainly driven by shear, cells inside virtual channels of a cuvette are deformed by an interfacial stress σ_i originating from the liquid-liquid interface (Fig. 3B, left and Fig. 3C, bottom).

We investigate this hypothesis by calculating the hydrodynamic shear stress and normal stress on a cell surface inside a glass cuvette using FEM simulations (see Methods). Here, we assume an experimental flow rate of 1.02 $\mu\text{l s}^{-1}$ and a constant viscosity of 11 mPa s . In geometries with a

relative cell size $\lambda = r_{\text{sphere}}/R$, where r_{sphere} is the cell radius and R is the equivalent cuvette radius, respectively, peak stress increases with increasing λ for a fixed flow rate (Supplementary Fig. 7). While this relationship is valid for any laminar microfluidic system with a central spherical obstacle, virtual channels also possess a constant stress contribution at the interface resulting from the viscosity mismatch between the two co-flowing aqueous polymer phases. This effect has been described previously for systems where an annular flow of low viscosity, surrounds a core flow of high viscosity, e.g., oil, to reduce friction at the oil-wall interface for power-effective transportation in pipes³². Under these conditions, σ_i can be derived from the simple expression:

$$\sigma_i = -\frac{8 \eta_{\text{sa}}(\dot{\gamma})}{w} (u_c - 2u_a) , \quad \#(3)$$

where $\eta_{\text{sa}}(\dot{\gamma})$ is the viscosity of the sample flow at shear rate $\dot{\gamma}$, w is the width of the virtual channel, u_c is the cell velocity inside the virtual channel and u_a the superficial velocity proportional to $\frac{Q_{\text{sh}}}{\pi R^2}$.

Utilizing experimental values for HL60 cells at a flow rate of $1.015 \mu\text{l s}^{-1}$ inside a $w = 14 \mu\text{m}$ virtual channel of a cuvette with side length $360 \mu\text{m}$ and $u_c = 2.8 \text{ cm s}^{-1}$ yield an interfacial stress at the liquid-liquid interface of $\sigma_i = 78 \text{ Pa}$ and induces cell deformation perpendicular to the direction of flow (Fig 3C, bottom). Viscosity values of $\eta_{\text{sa}}(\dot{\gamma}) = 11 \text{ mPa s}$ was calculated from FEM simulations and independently confirmed using calibration beads with a fixed elastic modulus of $E = 1.5 \pm 0.5 \text{ kPa}$ ³³ (see Methods). Since Equation 3 allows to consider the ellipsoidal cell shape as the steady-state response to a simple creep-compliance experiment with a constant interfacial stress $\sigma_i = E \varepsilon_c$, the Young's modulus can directly be calculated from the area strain ε_c of the cell (see Methods). Effectively, the liquid-liquid interface acts as a high-frequency liquid cantilever for probing global cell rheological properties on a millisecond timescale such that material properties can be extracted from simple linear models."

Reviewer:

(2) Microfluidic channel is only one of the available techniques in this area. It would be helpful to provide a bit more context for the general reader as to how this method fits with the broader slate of techniques, e.g., like electrical, optical or acoustic methods.

Author response:

We appreciate the suggestion of the reviewer and agree that it is important to provide more context for the general reader. In principle our technology of virtual fluidic channels can be combined with any detection method like electrical, optical and acoustic but we think that optical is most suited. Electrical detection would require impedance measurements. However, the virtual confinement does not change the channel resistance as the physical diameter does not change. The same is true for acoustic measurement. In a broader context, e.g. acoustic microscopy or optical stretching for studying cell mechanical properties we can perform a comparison with virtual channels, where our method provides the higher throughput. The discussion in the initial manuscript mainly focused on atomic force microscopy and micropipette aspiration to analyse the mechanical properties of tissue but we have now expanded that to opto-acoustic microscopy and have modified the discussion on page 12 accordingly:

"In the low Reynolds number regime laminar systems can be discriminated by single and multiple phase flows ³⁶⁻⁴⁰. While single aqueous phases have mainly been used to study material properties of molecules and cells ^{10-12,24}, multiphase flows allow to tailor the total reaction volume to the analyte of interest, e.g. to control diffusive processes or for spatiotemporal alignment of particles and cells with reagents ^{3,9}. Interestingly, a combination of both approaches, i.e. modifying the system size to tailor the hydrodynamic stress distribution to perform rheological characterization of biological matter across various scales, has not been achieved before.

The latter would be of high importance for the growing field of 3D tissue assays. In contrast to two-dimensional cell culture systems using plastic petri dishes, 3D models allow to mimic a physiological environment that can be used, e.g. for drug discovery ⁴¹, to study thrombi formation *in-vitro* ⁴² as well as cardiac tissue regeneration ⁴³. While recent research was focusing on generation of 3D cell cultures, e.g. using hanging drop microplates or spheroid microplates with ultra-low attachment coating, high-throughput characterization of tissue function mainly focused on molecular labels or genomics as well as transcriptomics potentially altering the sample ⁴⁴. In contrast to standard assays in cell biology, methods utilizing intrinsic material properties of tissues provide a label-free access to biological function. Originally, these methods have been limited to single cells ^{45,46} and were mainly based on the interaction between an acoustic, optical or physical probe ^{47,48}. Recently, atomic force microscopy and micropipette aspiration have been extended towards 3D tissue assays where they probe mechanical properties locally by integrating over a limited number of cells ⁴⁹. However, a high-throughput assay able to impose a homogeneous stress distribution on a tissue that integrates cell-cell and cell-matrix interactions into the stress-strain response remained elusive."

Reviewer:

(3) Finally there is a large body of literature in the area of multiphase laminar flow in microfluidics (and associated simulation) and single cell mechanics in microfluidics. Again would be helpful to better put this work into context.

Author response:

We agree with the reviewer that there is a significant amount of literature focusing either on multiphase laminar flow, extensional flow and single cell mechanics using microfluidic methods. Knight *et al.* and Wu *et al.* describe a hydrodynamic focusing device to reduce the volume of the sample flow to a size that facilitates mixing on a microsecond timescales ^{7,8}. They use a co-phase flow of different viscosities and demonstrate that sample flow diameter scales not only with flow rate but also with viscosity. Brown *et al.* expand on that idea by utilizing co-flowing aqueous solution of different refractive indices as an optical waveguide ⁹. In contrast to the above literature on multiphase flow, single molecule or single cell mechanics utilize single phase flow either as an extensional flow or in steady state. For example, Shaqfeh summarizes previous studies on stretching single DNA molecules to investigate their relaxation dynamics where they focus on complex flows and flow-induced reactions ¹⁰. Specifically for cell mechanics, Dudani *et al.* and Gossett *et al.* demonstrate the application of extensional flows to measure cell mechanical properties ^{11,12}. While they can identify sub-populations and compare samples of different origin, the complex flow pattern does not allow for the derivation of cellular material properties. In contrast, earlier work of

Hochmuth *et al.* showed that the Poiseuille flow can be used to induce steady-state cell deformation. While this approach enables the calculation of elastic properties from simple models, the application was limited to erythrocytes due to limited shear stress¹³⁻¹⁵. Some years ago, our research group introduced real-time deformability cytometry to characterize cells of almost arbitrary stiffness in a microfluidic device in real-time and high-throughput¹.

In the current manuscript we consequently built upon our previous experience and ask the question of scalability. This includes two aspects: the independency of the microfluidic chip platform as pointed out by the reviewer in the first question as well as the extension of the applicability of the mechanical assay from single cells to whole tissues. We agree, that the combination of both concepts was not well emphasized in the previous version of the manuscript. In the revised version, we first demonstrate that the liquid-liquid interface imposes a stress of the cells, which can be used to extract cell material properties. We provide a simple analytical model, which we compare to existing approaches of calculating the full profile around a cell moving inside a microfluidic channel and confirm our results using calibration beads of fixed elastic modulus³.

We added the following explanation on page 17 of the Methods section:

“For determining the elastic properties of cells and spheroids inside the glass cuvette we model the full virtual channel inside a cross-section of 360 μm side length. The relative diameter w of the virtual channel is obtained from experimental data (Fig. 1A, top inset), the velocity at the liquid-liquid interface is extracted from FEM simulations assuming Stokes flow and used as a moving wall boundary condition. Placing a sphere of radius r_{sphere} , representing either a cell or a spheroid, in the centre of the virtual channel allows for extracting a mean surface shear rate for all experimental conditions. The dynamic viscosity of the MC solution is then determined by using Equation 24.

For deriving the elastic modulus of cells and spheroids, respectively, we provide two different analytical models.

Stokes equation and linear elasticity theory

The first approach is based on an analytical model introduced earlier³¹. Briefly, for a moving object inside the (virtual) channel we assume a steady-state defined by the balance of shear and normal forces on its surface. Expansion of the Stokes equation in the appropriate geometry and coupling to linear elasticity theory allows to predict surface displacement of an initially round object. Our model shows for a given relative size $\lambda = r_{\text{sphere}}/R$, where r_{sphere} is the object's radius and R is the (equivalence) radius of the constriction, that the characteristic stress scales with the viscosity of the surrounding medium as well as with object velocity and is a function of the surface position. Since deformation is governed by shear and normal forces originating from the Poiseuille flow inside the microfluidic channel, the steady-state object shape resembles a bullet with finite front radius, which is best characterized by its circularity. Assuming objects of varying bulk elastic modulus E , cell deformation can be predicted for a given characteristic stress and be used to generate a lookup table to solve the inverse problem, i.e., to predict E from cell deformation.

Interfacial stress at liquid-liquid interface

Earlier work on the reduction of viscous friction for material transport in pipes described an interfacial stress σ_i at the liquid-liquid interface of co-flowing aqueous solutions³². Here, we use σ_i to impose a surface stress on objects confined in a virtual channel and to induce a deformation

with an amplitude that depends on the elastic modulus of the object, its velocity, the virtual channel radius and medium viscosity (Equation 3). In contrast to the above model utilizing hydrodynamic shear and normal forces where deformation is quantified by circularity, the interfacial stress acts perpendicular to the direction of flow and leads to an ellipsoidal deformation, that can be described by an area strain. The one-dimensional elongation in the direction of flows allows to consider the shape as the steady-state strain in a Kelvin-Voigt-Model and the elastic modulus can be calculated from $\sigma_i = E \varepsilon_c$.

In principle both, the interfacial stress model and the Stokes equation in combination with linear elasticity theory, can be used to extract elastic properties from RT-DC measurements. Our results suggest that the applicability depends on the relative object size λ inside the constriction. While for large λ , e.g. cells inside a PDMS channel, surface stress is dominated by the hydrodynamic shear and normal forces originating from Poiseuille flow, deformation in geometries with small λ is governed by the interfacial stress only as shear stress can be neglected.

We investigate the dependency of hydrodynamic shear and interfacial stress on λ by performing FEM simulations for different flow rates of $0.53 \mu\text{l s}^{-1}$ and $1.02 \mu\text{l s}^{-1}$ and the corresponding viscosities of 53 mPa s and 11 mPa s for spheroid and cell measurements, respectively (Supplementary Fig. 7). While the interfacial stresses (dashed lines) do not depend on λ , we find an increasing peak stress for increasing relative object size. Here, σ_i has been derived from Equation 3 utilizing typical experimental parameters of $w = 14 \mu\text{m}$, $u_c = 2.8 \text{ cm s}^{-1}$ and $u_a = 0.8 \text{ cm s}^{-1}$ for single HL60 cells (blue dotted line) and $w = 190 \mu\text{m}$, $u_c = 1.67 \text{ cm s}^{-1}$, and $u_a = 0.38 \text{ cm s}^{-1}$ for spheroids (black dotted line), respectively. For single cells, we find that $\sigma_s > \sigma_i$ for $\lambda > 0.9$, where σ_s is the hydrodynamic shear stress, while for spheroids we obtain $\sigma_s > \sigma_i$ for $\lambda > 0.5$.

For defining a relative size ratio interval for dominating σ_i , we define that the interfacial stress has to exceed the hydrodynamic stress by a factor of 10 or higher. Applying the experimental conditions stated above this results into a threshold of $\lambda < 0.4$ for single cells while spheroid deformation is always governed by shear stress."

and on page 18:

"Viscosity calibration was performed by using beads of fixed elastic modulus $E = 1.5 \pm 0.5 \text{ kPa}$ ³³. Measuring cell deformation inside a virtual channel of $w = 16 \mu\text{m}$ and utilizing the stress-strain relationship $\sigma_i = E\varepsilon_c$ allows to calculate η_{sa} from Equation 3."

Secondly, we use our system for a first high-throughput comparison between single cells and entire tissues. This part was not very well described before and we now improved Fig. 4 and added the following part on page 10 of the results section:

"We reveal the full potential of virtual fluidic channels by studying the impact of single cell mechanics on tissue properties using our hydrodynamic system. Such a comparative analysis is of high importance, e.g. to understand emergent effects in cellular co-culture systems in regenerative medicine. This has so far only been possible using atomic force microscopy with its ability to

bridge length scales from a few micrometers to a millimeter scale. Here, we aim to utilize the capability of virtual channels to modify the constriction diameter and hydrodynamic stress distribution over several orders of magnitude and perform a high-throughput characterization of single HEK293 cells and spheroids as a tissue model using a single experimental system.

Spheroids have been cultured for multiple days in agarose micro-molds with an initial seeding density of approximately 30 cells per well. We observe an average spheroid diameter of $82 \pm 23 \mu\text{m}$ (mean \pm std. dev.) on day one, which increases to $134 \pm 19 \mu\text{m}$ on day four (Fig. 4A, left). Approximating the number of cells inside the spheroids from the volume allows to fit an exponential growth curve to our data, where we find a doubling time of 32 hours (Fig. 4A, right).

For studying the contribution of single cells to tissue stiffness we first adjust the diameter of the virtual channel inside the cuvette to $w = 16 \mu\text{m}$ to probe single HEK293 cells by interfacial stress (Fig. 4B, left panel; 114 μM MC sample and 5 mM PEG40000 sheath) yielding $\varepsilon_c = 0.09 \pm 0.03$ (Fig. 4B, center panel and Supplementary Fig. 10). The cellular Young's modulus E is calculated under steady-state conditions applying the interface model. Utilizing experimental parameters of $u_c = 2.8 \text{ cm s}^{-1}$ and $\eta_{sa} = 11 \text{ mPa s}$ yields an interfacial stress of $\sigma_i = 66 \text{ Pa}$, which exceeds the stress originating from Poiseuille flow (Supplementary Fig. 7). This allows together with the mean area strain to obtain a Young's modulus $E = 0.78 \pm 0.38 \text{ kPa}$ for single HEK293 cells (Fig. 4B, right panel).

Finally, virtual fluidic channels are applied for high-throughput mechanical characterization of HEK293T spheroids as a 3D-tissue model. Cells are cultured into spheroids with a diameter of approximately $130 \mu\text{m}$ using agarose micro-molds, where each spheroid contains an average of 650 cells (Fig. 4A, right, blue box plot; see Methods). Taking advantage of the simple relationship between Q_{sa} , Q_{shr} , η_{sa} and η_{shr} stable virtual channels of $190 \mu\text{m}$ mean diameter are established inside the constriction of the cuvette (Equation 2 and Fig. 4C, left panel). While suspended spheroids slightly perturb the liquid-liquid interface during translocation, the low Reynolds number regime ensures a fast recovery within less than two spheroid diameters. The spheroids show the typical bullet-like shape comparable to single cells with a mean deformation of 0.040 ± 0.002 (Fig. 4C, center panel), which is significantly higher than the undeformed reference state at $260 \mu\text{m}$ virtual channel width (Fig. 4C, center panel inset and Supplementary Fig. 11).

For estimating the elastic modulus of spheroids we first calculate σ_i from Equation 3 utilizing experimental parameters $Q_{sh} = 330 \text{ nl s}^{-1}$, $u_c = 1.67 \text{ cm s}^{-1}$ and $w = 190 \mu\text{m}$ as stated above. Assuming a viscosity $\eta_{sh} = 53 \text{ mPa s}$ from FEM simulations considering the full hydrodynamic geometry yields an interfacial stress of $\sigma_i = 20 \text{ Pa}$. With a magnitude much smaller than the hydrodynamic shear and normal stress, spheroid deformation is dominated by the parabolic flow profile and the elastic modulus is extracted using the analytical model published earlier^{31,34}. This yield $E = 86 \pm 36 \text{ Pa}$ (mean \pm std. dev.; Fig. 4C, right panel). Analysing three experimental

replicates of more than 200 spheroids and 1,000 cells, respectively, we derive an apparent Young's modulus of 84 ± 9 Pa (mean \pm SEM), which is a factor of ten smaller than single cells cultured in a 2D plastic culture dish with 0.85 ± 0.07 kPa (Fig 4C, right panel inset)."

Figure 4: High-throughput tissue mechanics inside glass cuvette. **A** Bright-field images of HEK293T spheroids obtained at four subsequent days inside agarose mold. Initial seeding density is 30 cells per micro well (left). Exponential fit (solid line) to the mean cell number per spheroid yields a doubling time of approximately 32 hours. Blue box plot represents distribution of cell number per spheroid from the data indicated in (C, center panel) with a mean cell number of 650. **B** Representative image of a HEK293T cell in virtual channel of 16 μm diameter (sample 114 μM MC at $Q_{sa} = 20$ nl s⁻¹, sheath 5 mM PEG40000 at $Q_{sh} = 1,000$ nl s⁻¹, left). Corresponding scatter plots of area strain (center) and Young's modulus (right) versus HEK293T cell size. Young's modulus is calculated from area strain assuming an interfacial stress of 66 Pa. Shaded area indicates cells that are not confined inside the virtual channel. **C** Representative image of HEK293T spheroid in virtual channel of 190 μm width (sample 114 μM MC at $Q_{sa} = 330$ nl s⁻¹, sheath 5 mM PEG40000 at $Q_{sh} = 200$ nl s⁻¹, left). Line plots in (B) and (C) indicate the squared intensity gradient (arb. units) perpendicular to the flow direction while centres of the intensity maxima identify the virtual channel interfaces (dashed white lines). Corresponding scatter plots of deformation (center) and Young's modulus (right) versus HEK293T spheroid size. Inset in right graph compares Young's modulus of single cells ($E = 0.85 \pm 0.07$ kPa) and spheroids ($E = 84 \pm 9$ Pa) from three experimental replicates each consisting of more than 1,000 single cell and 200 spheroid measurements. Scale bar is 20 μm. Statistical data analysis is performed using linear mixed models and error bars represent the standard error of the mean (**, p < 0.01).

Reviewer:

In summary, while interesting work, it seems more appropriate for a more specialized journal.

Author response:

We agree with the reviewer, that the initial submission lacked the emphasis on the aspects that distinguish our work from the state-of-the-art. We are confident that the two new aspects of using a liquid-liquid interface to probe cell mechanical properties as well as the possibilities to perform single cell and tissue assays highlight the novelty of the presented approach. In fact, virtual channels allow for exactly this kind of comparative studies across different size scales.

Reviewers' comments:

Reviewer #1 (Remarks to the Author):

The authors have made a significant effort with additional experiments, figures, and explanation which have largely addressed my concerns. The ability to use the virtual channels to observe the same differences in deformability is helpful to validate the technique. Also, more clearly showing the results in a flow cytometer flow cell with individual cells and discussing the origin of the deformation of single cells in this geometry based on interfacial normal stress is quite interesting and introduces a novel concept.

My only remaining concern is how the work is introduced in the context of improving microfluidics channel tuning / fabrication generally. The manuscript focuses on the co-flow systems in the context of cell mechanics measurements, and I see high-throughput cell mechanics instruments and approaches as a better set of background literature to introduce the work, as it is still unclear (and not a focus of the work) how the viscous co-flow systems would be used in other microfluidic applications where channel dimensions need to be tuned on the fly.

Reviewer #2 (Remarks to the Author):

The authors resubmit a manuscript describing the measurement of cell mechanical properties in pinched flow microfluidic systems. While I remain positive about the goals and ideas this work describes and recognize the improved incorporation of relevant literature, it remains weakly focused and could better introduce the concept of the fluidic channels they are working here to promote. For example, the authors' case that the fluidic channels are an improvement on soft lithography detracts from this manuscript which would better focus on the innovations which appear to be:

1) co-flowing aqueous streams of differing viscosities can be used to shear cells at high throughput. As presented, I continue to struggle with the use of the term "virtual fluidic channels" as a newly introduced concept. Questions still remain about what fundamental difference exist with Dudani et al.. In that work, co-flowing fluidic, aqueous streams were used to deform cells with shear. In this revised manuscript, having streams of differing viscosity allows enhanced forces to be applied that cannot be done with identical fluids. Perhaps this issue can be fixed with a clearer Introduction, first describing previous work and then identifying exactly how their work builds upon it. Related to this issue is that I still do not believe that these solutions are "immiscible". This question was posed previously and wasn't answered in the reviewer response. Simply put, these aqueous mixtures have differing viscosities and relatively low diffusion coefficients but do they form droplets at other flow rates? If no, the interface evolution could perhaps be modeled with a simple diffusion profile over time down the channel length. This presumably would impact the relationships used to model forces on cells and the measures of elasticity extracted from their experiments.

2) the approach allows for scale up to commercial cuvettes. Here the innovation is the use of co-flowing aqueous streams in larger scale cuvettes that can be used in commercial devices. Related to this question is the physical origin of how a "3D" channel is created in the cuvette. Is it a wetting issue because these are truly immiscible phases or is it strictly related to shear? Along these lines and while the simulations of Fig 3B are clear, is there experimental verification (say, a confocal image) that this indeed does happen?

Specific comments:

Pg. 2: "here, the accessible volume for the analyte is defined by the microfluidic chip geometry only". Relative flow rates, and hence accessible volume, could certainly be altered in the work of Dudani, et al. (ref 24) so I'm sure I understand this statement.

References need to be double checked. I didn't go through these in detail but ref 32 in the Methods section appears to be ref 31 in the References list for example.

Reviewers' comments:

Reviewer #1 (Remarks to the Author):

Comment 1

Reviewer:

The authors have made a significant effort with additional experiments, figures, and explanation which have largely addressed my concerns. The ability to use the virtual channels to observe the same differences in deformability is helpful to validate the technique. Also, more clearly showing the results in a flow cytometer flow cell with individual cells and discussing the origin of the deformation of single cells in this geometry based on interfacial normal stress is quite interesting and introduces a novel concept.

Author response:

We thank the reviewer for the positive assessment of our revised manuscript.

Comment 2

Reviewer:

My only remaining concern is how the work is introduced in the context of improving microfluidics channel tuning / fabrication generally. The manuscript focuses on the co-flow systems in the context of cell mechanics measurements, and I see high-throughput cell mechanics instruments and approaches as a better set of background literature to introduce the work, as it is still unclear (and not a focus of the work) how the viscous co-flow systems would be used in other microfluidic applications where channel dimensions need to be tuned on the fly.

Author response:

We agree with the judgement of the reviewer that the introduction was too broadly written while the manuscript focusses only on cell and tissue mechanics. We have rewritten the first and final part of the introduction where we specifically highlight the potential of the method to perform high-throughput cell and tissue measurements and also emphasize the capability to integrate the technology into a commercial flow cytometer.

On page 1 we have replaced the following part of the introduction:

“The introduction of soft lithography as a rapid prototyping technology in microfluidics had an impact in life science research similar to photolithography in solid-state physics¹⁸. Using replicas of elastomers not only changed the way of creating small features, e.g., for lab-on-chip devices in single-molecule and single-cell applications, but also provided nearly unlimited access to the nano- and microscale straight from the bench^{21–23,37}.

While microfluidic devices have initially been used to study hydrodynamics at the transition between laminar and turbulent flows^{12,38–40}, today's applications span the entire range from single molecules and cells to screening assays based on droplet systems including sorting^{20,26,41–44}. The scalability across various length scales requires feature dimensions matching the sample size – a challenge that is solved by fabricating customized master molds or more recently by using emergent technologies, e.g., direct laser writing⁴⁵, glass capillaries⁴⁶, paper-based microfluidics⁴⁷ or reshaping of free-standing liquids on a surface^{48–50}.

One specific example for dynamic control of the total reaction volume are multiphase flows, which have been used for diffusive mixing^{40,51}, as optical waveguides⁵² or for mediating reaction times on a millisecond scale²¹. Here, aqueous solutions of polymers with different viscosities are used to either control particle confinement or modify light propagation inside liquid media.

In contrast to laminar systems with multiple phases, single phase flows have been used to stretch DNA molecules or cells in an extensional geometry to study their mechanical properties^{25,41,42}. Here, the accessible volume for the analyte is defined by the microfluidic chip geometry, only. While allowing for high-throughput screening and discrimination of different populations, the complex hydrodynamic environment does not enable extracting of material properties. More recently, methods based on micro-constrictions or Poiseuille flows allowed to obtain a cellular Young's modulus and viscosity^{26,43,53,54} but still suffered from cell-wall interactions or the requirement to adjust the physical environment to the analyte of interest^{10,35}. On-the-fly control of microfluidic dimensions, which would be essential first to make mechanical measurements accessible using standard bench-top equipment, e.g. flow cytometry, and second to probe soft matter on various length scales, e.g. to disentangle the role of single cells mechanics for tissue rheology, remained elusive.”

by:

“Phenotyping of cells utilizing intrinsic material properties, i.e. elasticity and viscosity, is increasingly being recognized to be of key relevance in basic and applied life science research¹⁻³. Early developments have mainly been driven by technologies like atomic force microscopy⁴, micropipette aspiration⁵ as well as optical stretching^{6,7} and focused on the fundamental interplay between cytoskeletal protein structure and viscoelastic cell properties to combine physics and biology on a molecular as well as cellular scale⁸⁻¹⁰. In this field, mechanosensitivity has been found as an essential mechanism of how cells respond to and communicate with their environment¹¹.

The great success of mechanical phenotyping to study cell function paved the way towards applications in translational research but required an elevated sample throughput that could not be reached by established methods. One answer to this challenge was found in microfluidic devices¹²⁻¹⁴. Already decades ago, Hochmuth *et al.* utilized glass capillaries and high-speed video microscopy to study red blood cell deformation in flow¹⁵⁻¹⁷. In their pioneering work they carried out an extensive characterization of erythrocyte shape, orientation and viscosity. Glass capillaries share the advantages of all microfluidic systems, i.e. high-throughput and serial sample processing capabilities, but lack the flexibility to design and implement sophisticated geometries.

The introduction of soft lithography as a rapid prototyping technology completely revolutionized the way of creating microfluidic structures^{18,19}. Using replicas of elastomers enabled benchtop production of lab-on-a-chip devices to access the nano- and microscale for single molecule and single cell experiments²⁰⁻²³. For example, Gossett *et al.* demonstrated high-speed mechanical phenotyping of various cell lines and showed for embryonic stem cells a discrimination of their pluripotent state with sufficient statistical power²⁴. The idea of stretching cells in an extensional flow has been expanded by Dudani *et al.* towards a pinched-flow geometry, where hydraulic circuit design is used to deform cells in multiple stretching modes²⁵. Recently, mechanical cell characterization has also been complemented with real-time data analysis^{26,27} as well as fluorescent labelling to combine the sensitivity of intrinsic material properties with the specificity of molecular markers²⁸.

While lab-on-a-chip devices are nowadays routinely being used in translational research to assess viscoelastic properties, e.g., for studying the onset of infectious diseases²⁹ and for monitoring patients' status over time^{30,31}, the high-throughput analysis of this label-free biomarker is currently limited to single cells and excludes multicellular structures. However, a detailed understanding of how mechanics impacts on tissue development and homeostasis would be highly relevant to establish 3D cell culture models that incorporate material properties into the field of tissue engineering^{32,33}. Despite some initial work on spheroids using atomic force microscopy and micropipette aspiration current approaches lack the potential of simultaneous high-throughput characterization of cells and tissues applying a single experimental method^{34,35}.

A suitable microfluidic system for high-speed mechanical phenotyping of spheroids is a standard glass cuvette. Having an equivalent diameter of several hundred of micrometers these cuvettes are the main entity in fluorescence-based flow cytometers, where translocating cells are analysed by laser excitation and detection of the emitted as well as the scattered light signal³⁶. While providing sufficient confinement for spheroids, commercial flow cytometers are not capable to analyze single cells by their mechanical properties. One reason is found in the large cross-sectional area of the cuvette, which is needed to reduce blocking by debris but also results in low hydrodynamic stress amplitudes. Since determination of elasticity and viscosity requires, first, exposure of cells to sufficient shear as well as normal stresses and, second, detection of the corresponding cellular deformation, the large cuvette diameter effectively inhibits mechanical measurements.

Here, we introduce virtual fluidic channels to create microfluidic constrictions of variable cross-section inside glass cuvettes from commercial flow cytometers. This scalability allows for adjusting the hydrodynamic stress range sufficiently to deform cells and spheroids within one experimental assay. Virtual channels are liquid-bound microfluidic devices and are formed by co-flowing aqueous polymer solutions at low Reynolds numbers where a stabilizing sheath confines a sample flow between immiscible liquid phases. We demonstrate that virtual fluidic channel dimensions can be adjusted within seconds in real-time and enable high-throughput rheological measurements of biological samples across various size ranges.”

To link the introduction also to the discussion and to pick up the focus on cell as well as tissue mechanics we added / modified the following paragraphs in the discussion on page 14:

“As one potential application we performed mechanical characterization of HL60 cells in both systems. Proof-of-principle experiments in PDMS chips using real-time deformability cytometry revealed an elasticity of 0.95 ± 0.01 kPa (median \pm SEM) and 0.92 ± 0.02 kPa for a solid and fluid confinement, respectively, showing no significant differences. Next, we repeated our assay using virtual channels created in a large glass cuvette where we obtained a slightly elevated Young’s modulus of 1.5 ± 0.1 kPa. The observed differences potentially originate from different levels in deformation amplitudes present in both systems. In fact, the deformation inside the glass cuvette is so small that deviations of the initial cell shape from an ideal circle (circularity = 1) might impeded application of an analytical model published earlier that requires finite cell shape changes in shear flow^{27,43}.

For solving this discrepancy, we demonstrate that the viscosity mismatch at the liquid-liquid interface leads to an interfacial stress resulting in a cellular strain that allows for extracting material properties from simple creep compliance experiments. For HL60 cells probed by this liquid-liquid interface of a virtual channel inside a glass cuvette we obtain an elastic modulus of 0.97 ± 0.03 kPa, which is in good agreement not only with our PDMS results based on shear flow but also with data published earlier^{27,40}. In fact, our proof-of-principle experiments demonstrate that a virtual channel acts like a liquid cantilever that exposes biological samples to a homogeneous 3D stress distribution for probing biological matter on a millisecond scale. In contrast to atomic force microscopy or microconstrictions virtual fluidic channels enable a label-free analysis without any cell-wall interaction and at high-throughput.”

To link the introduction also to the discussion we added the following paragraph to page 14:

“In summary, virtual fluidic channels allow for on-the-fly generation of nearly arbitrary hydrodynamic environments inside large fluidic geometries, e.g. commercial glass cuvettes. We demonstrate that our method provides simultaneous access to material properties of biological samples across various length scales reaching from single cells to tissues, which could be highly relevant for a functional evaluation of engineered tissue grafts in regenerative applications. Ultimately, our approach renders microfluidic experiments independent of soft lithography extending the parameter space of classical fluorescence flow cytometry towards an unbiased cell and tissue analysis utilizing intrinsic material properties.”

Reviewer #2 (Remarks to the Author):

Comment 1

Reviewer:

The authors resubmit a manuscript describing the measurement of cell mechanical properties in pinched flow microfluidic systems. While I remain positive about the goals and ideas this work describes and recognize the improved incorporation of relevant literature, it remains weakly focused and could better introduce the concept of the fluidic channels they are working here to promote. For example, the authors' case that the fluidic channels are an improvement on soft lithography detracts from this manuscript which would better focus on the innovations which appear to be:

Author response:

We thank the reviewer for the overall positive evaluation of our manuscript and refer to the specific answers to the points raised.

Reviewer:

1) co-flowing aqueous streams of differing viscosities can be used to shear cells at high throughput. As presented, I continue to struggle with the use of the term “virtual fluidic channels” as a newly introduced concept. Questions still remain about what fundamental difference exist with Dudani *et al.*. In that work, co-flowing fluidic, aqueous streams were used to deform cells with shear. In this revised manuscript, having streams of differing viscosity allows enhanced forces to be applied that cannot be done with identical fluids. Perhaps this issue can be fixed with a clearer Introduction, first describing previous work and then identifying exactly how their work builds upon it.

Author response:

We gratefully acknowledge the comment of the reviewer and fully agree that the main idea of the previous version of our manuscript was not very well motivated. We now specifically cite previous literature (Gossett *et al.*, Dudani *et al.*, Otto *et al.* and Rosendahl *et al.*) on page 2 of the introduction as follows:

“For example, Gossett *et al.* demonstrated high-speed mechanical phenotyping of various cell lines and showed for embryonic stem cells a discrimination of their pluripotent state with sufficient statistical power²⁴. The idea of stretching cells in an extensional flow has been expanded by Dudani *et al.* towards a pinched-flow geometry, where hydraulic circuit design is used to deform cells in multiple stretching modes²⁵. Recently, mechanical cell characterization has also been complemented with real-time data analysis^{26,27} as well as fluorescent labelling to combine the sensitivity of intrinsic material properties with the specificity of molecular markers²⁸.”

In addition, as suggested by the reviewer we emphasize the novelty of our work compared to Dudani *et al.*, which is found in two points. First, cells in our work are measured in steady state, which allows for extraction of material properties. Second, the viscosity mismatch (as correctly pointed out by the reviewer) leads to a stress at the liquid-liquid interface of sufficient magnitude to induce cell deformation. Put together, both points enable to transfer mechanical phenotyping from microfluidic chips made of soft lithography to large microfluidic systems like commercial glass cuvettes. These points are now addressed in the introduction on page 2 as follows:

“While lab-on-a-chip devices are nowadays routinely being used in translational research to assess viscoelastic properties, e.g., for studying the onset of infectious diseases²⁹ and for monitoring patients’ status over time^{30,31}, the high-throughput analysis of this label-free biomarker is currently limited to single cells and excludes multicellular structures. However, a detailed understanding of how mechanics impacts on tissue development and homeostasis would be highly relevant to establish 3D cell culture models that incorporate material properties into the field of tissue engineering^{32,33}. Despite some initial work on spheroids using atomic force microscopy and micropipette aspiration current approaches lack the potential of simultaneous high-throughput characterization of cells and tissues applying a single experimental method^{34,35}.”

A suitable microfluidic system for high-speed mechanical phenotyping of spheroids is a standard glass cuvette. Having an equivalent diameter of several hundred of micrometers these cuvettes are the main entity in fluorescence-based flow cytometers, where translocating cells are analysed by laser excitation and detection of the emitted as well as the scattered light signal³⁶. While providing sufficient confinement for spheroids, commercial flow cytometers are not capable to analyze single cells by their mechanical properties. One reason is found in the large cross-sectional area of the cuvette, which is needed to reduce blocking by debris but also results in low hydrodynamic stress amplitudes. Since determination of elasticity and viscosity requires, first, exposure of cells to sufficient shear as well as normal stresses and, second, detection of the corresponding cellular deformation, the large cuvette diameter effectively inhibits mechanical measurements.

Here, we introduce virtual fluidic channels to create microfluidic constrictions of variable cross-section inside glass cuvettes from commercial flow cytometers. This scalability allows for adjusting the hydrodynamic stress range sufficiently to deform cells and spheroids within one experimental assay. Virtual channels are liquid-bound microfluidic devices and are formed by co-flowing aqueous polymer solutions at low Reynolds numbers where a stabilizing sheath confines a sample flow between immiscible liquid phases. We demonstrate that virtual fluidic channel dimensions can be adjusted within seconds in real-time and enable high-throughput rheological measurements of biological samples across various size ranges.”

and we kept:

“We develop an analytical model and compare the interfacial stress to typical shear and normal stress distributions inside Poiseuille flows of laminar systems. Our results verify that cell deformation of small objects embedded in virtual channels inside large geometries, e.g. single cells in cuvettes, is governed by interfacial stress, while large objects, e.g. spheroids as a tissue model system in glass cuvettes, deform dominantly by shear stress.”

Reviewer:

Related to this issue is that I still do not believe that these solutions are “immiscible”. This question was posed previously and wasn’t answered in the reviewer response. Simply put, these aqueous mixtures have differing viscosities and relatively low diffusion coefficients but do they form droplets at other flow rates?

Author response:

We apologize for not having answered the previous question of the reviewer to full extend and would like to do that now. In fact, aqueous PEG and MC solutions are miscible. There is no droplet formation observed. Yet, the required time scale necessary for a noteworthy intermixing is larger than the observed interaction time in a fluidic chip set-up. The diffusion coefficients of PEG8000 and MC are $D = 2 \cdot 10^{-10} \text{ m}^2/\text{s}$ and $D = 8 \cdot 10^{-12} \text{ m}^2/\text{s}$, respectively (obtained from dynamic light scattering measurements). Thus, diffusive mobility of PEG8000 and MC differ by at least one order of magnitude. Considering the finite resolution of our optical system ($0.34 \text{ } \mu\text{m}/\text{pix}$) we use $\Delta x = 1 \text{ } \mu\text{m}$ as minimal unambiguously resolvable displacement and calculate the minimal intermixing time $t = 2.5 \text{ ms}$ (PEG) and $t = 62 \text{ ms}$ (MC), respectively, by solving $\Delta x = \sqrt{2Dt}$. Thus, time required for observable intermixing is larger than the passage time through the PDMS chip channel depicted in Fig. 2C of the manuscript ($t_{\text{passage}} = \text{channel length}/\text{flow velocity} = 300 \text{ } \mu\text{m}/20 \text{ cm}\cdot\text{s}^{-1} = 1.5 \text{ ms}$).

However, this simplified approach ignores steric hinderance due to polymer entanglement and dynamic effects of flowing polymers. Steric hinderance is likely to play an important role for the PEG solution since PEG concentration approaches PEG saturation level (dissolved concentration is 400 mg/ml while saturation concentration is 630 mg/ml). Correspondingly the average separation between adjacent PEG molecules (3.2 nm) is similar to PEG radius of gyration $R_g = 2.9 \text{ nm}$. Polymer coils are likely to get in contact or even overlap with adjacent coils. Once polymer concentration exceeds the overlap concentration chain entanglement starts to dominate diffusivity and motion of a long polymer chain embedded in a polymer network is restricted along a tube-like region within the network (reptation model)^{1,2}. As a consequence, diffusion is qualitatively slower ($D \sim M_w^{-2}$) as compared to a diluted polymer solution ($D \sim M_w^{-1}$). Moreover, a flowing Gaussian polymer coil is stretched in direction of flow when exposed to acceleration in the extensional flow of the channel / cuvette^{3,4}. A stretched polymer coil possesses a directional diffusivity which is increased in direction of flow (compared to a uniform Gaussian coil) and decreased in direction of the liquid-liquid interface. We can only speculate on the magnitude of this stretching and whether the used PEG concentration is above or below the overlap concentration. However, the considerations suggest that the effective PEG diffusion coefficient is smaller than the measured value.

The correctness of our hypothesis, that the effective PEG diffusion coefficient is smaller than the measured one, is also supported by Supplementary Figure 2. Here, we observed the liquid-liquid interface in a 1.75 mm long constriction. While the passage time is a factor of 5 longer and exceeds the intermixing time of 2.5 ms we still observe a stable interface.

For incorporating the results of these calculations, the manuscript now reads on page 4:

“interface, which is stable and immiscible on the experimental time scale”

instead of:

“stable and immiscible interface”.

Also, the following paragraph is added to the Materials and Methods section on page 17:

“Characterization of interface stability has been done by analyzing the diffusion of both polymers, MC and PEG, in PBS. Diffusion coefficients of MC, PEG8000 and PEG40000 in PBS were measured by dynamic light scattering (DLS, Zetasizer Nano-ZS, Malvern Instruments, Herrenberg, Germany). DLS analyzes the autocorrelation function of back-scattered intensity fluctuations over time. The diffusion coefficients are $D_{MC} = 8 \cdot 10^{-12} \text{ m}^2 \text{ s}^{-1}$, $D_{PEG8000} = 2 \cdot 10^{-10} \text{ m}^2 \text{ s}^{-1}$ and $D_{PEG40000} = 3 \cdot 10^{-11} \text{ m}^2 \text{ s}^{-1}$, respectively. Measurements were carried out at 25°C at a scatter angle of 173°. Average diffusive displacement of dissolved polymers can be calculated using the solution to Fick’s second law $\Delta x = \sqrt{2Dt_{\text{passage}}}$ and is less than 1 μm for each polymer (typical time to traverse a microfluidic chip constriction $t_{\text{passage}} = \text{channel length}/\text{flow velocity} \approx 2 \text{ ms}$).”

References:

- [1] Pierre-Gilles de Gennes. *Scaling concepts in polymer physics*. Cornell university press, 1979.
- [2] Doi, Masao, and Samuel Frederick Edwards. *The theory of polymer dynamics*. Vol. 73. oxford university press, 1988.
- [3] Perkins, Thomas T., Douglas E. Smith, and Steven Chu. "Single polymer dynamics in an elongational flow." *Science* 276.5321 (1997): 2016-2021
- [4] Neumann, Richard M. "Polymer stretching in an elongational flow." *The Journal of chemical physics* 110.15 (1999): 7513-7515.

Reviewer:

If no, the interface evolution could perhaps be modeled with a simple diffusion profile over time down the channel length. This presumably would impact the relationships used to model forces on cells and the measures of elasticity extracted from their experiments.

Author response:

In order to analyze the interface evolution along channel length we turned to the flow cytometer glass-cuvette depicted in Fig. 3 of the manuscript since in PDMS chips diffusive intermixing can be neglected (see answer to previous comment). The channel length of a flow cytometer glass-cuvette is two orders of magnitude larger compared to a PDMS chip and thus passage time $t_{\text{passage}} = 700 \text{ ms}$ ($= 20,000 \mu\text{m}/2.8 \text{ cm}\cdot\text{s}^{-1}$) exceeds the minimal intermixing time. In fact, the width of the virtual channel increases by 3% over the course of the observable channel (corresponding to $\Delta w = 2 \mu\text{m}$ along 15 mm channel course; see Figure R1).

However, an intermixing between PEG sheath flow and MC sample is not resolvable, which might be due to the fact that the high polymer concentration inhibits diffusion across the interface (see answer above). However, an unexpected observation can be made at the far end of the cuvette channel. Small perturbations in the virtual channel lead to the formation of surface waves at the liquid-liquid interface which lead to considerable and periodic fluctuation in channel width (positions G, H and I in Figure R1). In contrast, experimental data reported in the manuscript are measured roughly midway between inlet and channel outlet (positions C and D in Figure R1) where such surface waves are negligible.

Figure R1: Distribution of virtual channel width along channel length.

Comment 2

Reviewer:

2) the approach allows for scale up to commercial cuvettes. Here the innovation is the use of co-flowing aqueous streams in larger scale cuvettes that can be used in commercial devices. Related to this question is the physical origin of how a “3D” channel is created in the cuvette. Is it a wetting issue because these are truly immiscible phases or is it strictly related to shear? Along these lines and while the simulations of Fig 3B are clear, is there experimental verification (say, a confocal image) that this indeed does happen?

Author response:

We gratefully acknowledge the question from the author. The formation of the virtual fluidic channel takes place in the merging funnel localized between sample nozzle and the actual cuvette channel. We have added a new Supplementary Figure (Supplementary Fig. 6) to illustrate the geometry of the cuvette inlet in detail (see below). Even though both fluids are theoretically miscible a noteworthy mixing cannot be observed since (1) the interaction time is too low for diffusive intermixing and (2) both fluids are restricted to laminar flow only. Laminar flows tend to conserve the shape once imposed upon them. Since all wall geometries involved in the virtual channel formation are circular (sample nozzle, merging funnel and the first $600 \mu\text{m}$ of the cuvette channel) a cylindrical “3D” channel embedded in sheath flow is created. Additionally, co-flow system tends to minimize the potential energy stored in shear stress at the liquid-liquid interface by minimizing the total area of the liquid-liquid interface. Hence, a cylindrical sample flow is energetically favorable.

In order to reveal the geometry of the virtual fluidic channel experimentally we added a water-soluble black ink to the sample flow while the sheath flow remained transparent (Supplementary Fig. 6D). According to Beer’s law the resulting light absorption is a measure

for the optical pathlength of the dyed medium (provided a constant dye concentration throughout the sample flow). The observed intensity profile across the fluidic channel is in good agreement with a circular arc as expected for a cylindrical sample flow enclosed by a transparent sheath flow (Supplementary Fig. 6E).

Supplementary Figure 6: Close-up images of flow cytometer glass-cuvette and geometry of the virtual fluidic channel. **A** Microscopy image of the sample inlet nozzle. **B** Stitched microscopy images of the transition from merging funnel to cuvette channel. **C** Technical drawing of flow cytometer glass-cuvette with markings of the pictured regions (red rectangles). **D** Microscopy image of virtual channel formed by co-flowing sample (H_2O , $Q_{\text{sa}} = 500 \text{ nl s}^{-1}$) and sheath (50 mM PEG8000, $Q_{\text{sh}} = 1000 \text{ nl s}^{-1}$) inside the cuvette channel. Water-soluble dye added to the sample flow provides additional contrast due to light absorption. Width of virtual channel $w = 228 \mu\text{m}$. **E** Intensity profile across the fluidic virtual channel. Red circle and crosshairs serve as guide to the eye. Scale bar $100 \mu\text{m}$.

Another way to visualize the curved geometry of virtual fluidic channel is to alter the focus plane by adjusting the focal length. A virtual channel that is “in focus” can be identified by two narrow bright bands localized at each side of the virtual channel as well as a uniform intensity within the virtual channel (Figure R2). However, if the focus plane is moved either upwards or downwards two opposite optical effects occur. Now, the virtual channel appears as a single band of increased or decreased intensity, respectively. These positional shifts in intensity can be attributed to the curvature of the liquid-liquid interface which is convex (i.e. focusing) for light entering the virtual channel and concave (i.e. diverging) for light exiting the virtual channel.

Figure R2: Imaging of virtual fluidic channel in a flow cytometer glass-cuvette under variation of focal length. Left: Schematic front view image of the cylindrical virtual channel (red circle) with light coming from above (black arrows) and objective below (not depicted). Right: **A** Focal plane adjusted to predominantly collect light from upper edge of the virtual fluidic channel. Note the bright band in the center of channel. The convex shaped liquid-liquid interface has a focusing effect on the incoming light (“focus A” in schematic). **B** Focal plane adjusted to predominantly collect light from center of the virtual channel (“focus B” in schematic). **C** Focal plane adjusted to predominantly collect light from slightly below center of the virtual channel (“focus C” in schematic). **D** Focal plane adjusted to predominantly collect light from lower edge of the virtual fluidic channel. Note the dark band in the center of channel. The concave shaped liquid-liquid interface has a diverging effect on the incoming light (“focus D” in schematic). Straight white lines indicate the extend of the virtual channel and dashed white lines the apparent channel width at the given focal length. Virtual channel is formed by co-flowing sample ($57 \mu\text{M MC}$, $Q_{\text{sa}} = 20 \text{ nl s}^{-1}$) and sheath (5 mM PEG40000 , $Q_{\text{sh}} = 1000 \text{ nl s}^{-1}$). Measurements were carried out using 40 x objective (LD Plan Neofluar, Zeiss). Scale bar 20 μm .

Comment 3

Reviewer:

Specific comments:

Pg. 2: “here, the accessible volume for the analyte is defined by the microfluidic chip geometry only”. Relative flow rates, and hence accessible volume, could certainly be altered in the work of Dudani, et al. (ref 24) so I’m sure I understand this statement.

Author response:

We thank the reviewer for this comment. Since we have changed the focus of the introduction from a general perspective to one on cell and tissue mechanics, this sentence is not included anymore.

Comment 4

Reviewer:

References need to be double checked. I didn’t go through these in detail but ref 32 in the Methods section appears to be ref 31 in the References list for example.

Author response:

We gratefully acknowledge the advice from the reviewer and have checked all references for correct integration and citation.

Minor changes

For better readability we replaced w by \tilde{w} to indicate the relative channel width.

Reviewer #3 (Comments to the author)

Comment 1

Reviewer:

In the revised manuscript, the authors articulated the rationale and novelty of their work in a highly improved context. The achievements and limitations of the relevant techniques were discussed properly. While multi-phase flow and single-molecule or single-cell mechanics using single-phase microfluidic channels have been extensively studied, the integration of the methods to probe the cellular mechanics at both single-cell and tissue level is the advance of this work.

Author response:

We thank the reviewer for the positive evaluation of our work.

REVIEWERS' COMMENTS:

Reviewer #2 (Remarks to the Author):

My previous concerns have been addressed in this revised manuscript.

REVIEWERS' COMMENTS:

Reviewer #2 (Remarks to the Author):

Reviewer:

My previous concerns have been addressed in this revised manuscript.

Author response:

We thank the reviewer for the positive evaluation of our revision.